**Characterizing regional oceanography and bottom environmental conditions at two**
**contrasting sponge grounds on the northern Labrador Shelf.**
*Evert de Froe[1,2,3]\*, Igor Yashayaev[4], Christian Mohn[5], Johanne Vad[6], Furu Mienis[1], Gerard*
*Duineveld[1], Ellen Kenchington[4], Erica Head[4], Steve W. Ross[7], Sabena Blackbird[8], George A.*
*Wolff[8], Murray Roberts[6], Barry MacDonald[4], Graham Tulloch[9], Dick van Oevelen[10]*
[1] NIOZ Royal Netherlands Institute for Sea Research, Department of Ocean Systems, PO
Box 59, 1790 AB, Den Burg, the Netherlands
[2] Centre for Fisheries Ecosystems Research, Fisheries and Marine Institute of Memorial
University of Newfoundland and Labrador, St. John's, Newfoundland and Labrador, Canada
[3] Wageningen Marine Research, Wageningen University and Research, PO Box 77, 4400 AB
Yerseke, the Netherlands
[4] Bedford Institute of Oceanography, Department of Fisheries and Oceans, PO Box 1006,
Dartmouth, NS, Canada B2Y 4A2
[5] Department of Ecoscience, Aarhus University, Frederiksborgvej 399, 4000 Roskilde,
Denmark
[6] Changing Oceans Research Group, School of GeoSciences, The University of Edinburgh,
Edinburgh, United Kingdom
[7] Univ. of North Carolina at Wilmington, Center for Marine Science, 5600 Marvin Moss Ln.,
Wilmington, NC, 28409 USA
[8]School of Environmental Sciences, University of Liverpool, 4 Brownlow Street, Liverpool,
L69 3GP, UK.
[9] British Geological Survey, Lyell Centre, Research Avenue South, Edinburgh, EH14 4AP
[10] NIOZ Royal Netherlands Institute for Sea Research, Department of Estuarine and Delta
Systems, PO Box 140, 4400 AC, Yerseke, the Netherlands
**\*Corresponding author:** evert.defroe@wur.nl
Key words: deep-sea sponges, sponge grounds, benthic-pelagic coupling, organic matter
transport, tidal dynamics, nutrients

## Abstract

Deep-sea sponge grounds are distributed globally and are considered hotspots of biological diversity and biogeochemical cycling. To date, little is known about the environmental conditions that allow high sponge biomass to develop in the deep sea. Here, we characterize oceanographic conditions at two contrasting sites off the northern Labrador Shelf with high- and low-sponge-biomass. Data were collected by year-long benthic lander deployments equipped with current meters, a turbidity and chlorophyll-*a* measuring device, and a sediment trap. Additionally, regional oceanography was described by analysing vertical conductivity - temperature-depth (CTD) casts, Argo float profiles, and surface buoy drifter data for the northern Labrador Shelf from 2005 to 2022. Stable isotopic composition of benthic fauna was determined to investigate food web structure at the sponge grounds. Our results revealed strong ($0.26 \pm 0.14$ m s$^{-1}$; mean $\pm$ SD) semidiurnal tidal currents at the high-sponge-biomass site, but twofold weaker currents ($0.14 \pm 0.08$ m s$^{-1}$; mean $\pm$ SD) at the low-sponge-biomass site. Tidal analysis suggests that, at the high-sponge-biomass site, kinetic energy is dissipated from barotropic tide to baroclinic tide/turbulence, which could enhance food availability for benthic organisms. Bottom nutrient concentrations were elevated at the high-sponge-biomass site which would benefits growth in deep-sea sponges. Organic matter flux to the seafloor was increased at the high-sponge-biomass site and consisted of fresher material. Finally, both sponge grounds demonstrated tight benthic-pelagic coupling prior to the onset of stratification. Stable isotope signatures indicated that soft corals (*Primnoa resedaeformis*) fed on suspended particulate organic matter, while massive sponges (*Geodia* spp.) likely utilized additional food sources. Our results imply that benthic fauna at the high-sponge-biomass site benefit from strong tidal currents, which increases food supply, and favourable regional ocean currents that increase nutrient concentration in bottom waters.

# 1  Introduction

Sponges are an ancient group of sessile filter feeders capable of pumping large quantities of water through their bodies (Vogel, 1977; Bergquist, 1978; Leys et al., 2011), thereby exchanging significant amounts of particulate- and dissolved organic matter and nutrients with the water column (e.g., van Duyl et al., 2008; Maldonado et al., 2012; Kahn et al., 2015; Rix et al., 2016). In the deep sea, sponges can form dense aggregations, known as sponge grounds, which are considered hotspots of macrofaunal diversity and abundance (Klitgaard, 1995; Buhl-Mortensen et al., 2010; Beazley et al., 2013; McIntyre et al., 2016), carbon- and nutrient cycling (Kutti et al., 2013; Cathalot et al., 2015; Maldonado et al., 2020a), and benthic-pelagic coupling (Pile and Young, 2006). Sponge grounds form complex habitats that provide breeding grounds and shelter for (commercially important) fish, increasing demersal fish biomass and diversity (Kenchington et al., 2013; Kutti et al., 2015; Meyer et al., 2019; Brodnicke et al., 2023). Finally, they are often classified as Vulnerable Marine Ecosystems (VMEs) as defined by the Food and Agriculture Organization of the United Nations (FAO, 2009).

Deep-sea sponge ecosystems are currently under threat from anthropogenic disturbances such as deep-water bottom trawling, deep-sea mining, and climate change. Pham et al. (2019) found that large quantities of sponges (~4% of total stock) have been removed by bottom trawling from sponge grounds on the Flemish Cap. Deep-sea sponges are especially vulnerable to bottom fishing due their longevity and slow growth (Leys and Lauzon, 1998; Hogg et al., 2010). Benthic trawling reduces the density and diversity of deep-sea sponge grounds (Morrison et al., 2020; Colaço et al., 2022), and recovery of disturbed sponge habitats can take decades to centuries (Vieira et al., 2020). In addition, prolonged exposure to elevated concentrations of suspended sediments, e.g. due to deep-sea mining, could adversely affect deep-sea sponges (Wurz et al., 2021). Recent studies suggest that climate change also impacts deep-sea benthic fauna (Brito-Morales et al., 2020; Jorda et al., 2020). For example, modelling predicted that the suitable area for *Vazella pourtalesii* on the Scotian Shelf would increase four-fold in the coming years due to warming of colder waters around its current habitat (Beazley et al., 2021). Nevertheless, research on the effect of climate change on deep-sea sponges is still in its infancy and to predict its effects on sponge grounds, a better understanding of the environmental conditions that favour their occurrence is needed.

In the past decades, research on deep-sea sponges has focused on their physiology and feeding behaviour (e.g., Leys and Lauzon, 1998; Yahel et al., 2007; Kahn et al., 2015; Robertson et al., 2017; Kazanidis et al., 2018; Maier et al., 2020b; Bart et al., 2021; de Kluijver et al., 2021), and assessing their spatial distributions using habitat suitability models (Knudby et al., 2013;

Howell et al., 2016; Beazley et al., 2018; Murillo et al., 2018). More recently, data on the
environmental conditions where sponge grounds are found have been gathered using long-term
measurements from lander-mounted equipment. These data indicate that sponge grounds are
commonly found in areas with internal waves (Davison et al., 2019) and comparatively strong
tidal currents which flush the seafloor with oxygen and nutrient-rich water, and with a high
suspended particle matter load near the seabed (Roberts et al., 2018; Hanz et al., 2021a, 2021b).
In addition, sponges can alter the hydrodynamic conditions of the benthic boundary layer by
increasing the bottom roughness, creating conditions favourable for larval recruitment and
suspended particle deposition (Abelson and Denny, 1997; Culwick et al., 2020). These studies
show that sponge grounds are found in areas with a variety of environmental conditions, but
little is known of the mechanisms controlling their spatial distribution or what controls their
biomass.
The Canadian Atlantic continental shelf breaks and upper slopes, including the northern
Labrador Shelf, host extensive sponge grounds (Kenchington et al., 2010; Knudby et al., 2013).
Sponge assemblages occur over a large depth range (200 – 2875 m) and are often aligned along
depth contours with presumably similar environmental conditions (Murillo et al., 2012;
Knudby et al., 2013). On the northern Labrador Shelf and upper slope, sponge assemblages
consist mostly of *Geodia* spp. and glass (hexactinellid) sponges (Kenchington et al., 2010) but
with locally variable sponge biomass. Therefore, this region provides a suitable setting to study
which environmental conditions favour high sponge biomass and to provide insight into the
factors that drive the spatial distribution of sponge assemblages on the eastern Canadian Shelf.
Furthermore, research on present environmental conditions on the seafloor is timely as the
Labrador Shelf region is one of the fastest warming large marine ecosystems globally (~1 °C
decade[-1]; Belkin, 2009), and according to ensemble-based climate change prediction, critical
water mass properties there, including temperature, particulate organic carbon, pH, and
aragonite saturation, are likely to change substantially by 2100 (Puerta et al., 2020). Recent
work on the Labrador Sea also shows that Arctic sea-ice melt can impact the hydrographic
conditions in this region (Yashayaev, 2024). Therefore, analysis of the contemporary
conditions provides a baseline or a benchmark for referencing future ocean and ecosystem
conditions. This study presents a valuable reference dataset for the upper slope of the northern
Labrador Shelf against which future changes could be evaluated.
To obtain a better understanding of the environmental conditions and ecosystem functioning
of high- and low-sponge-biomass sites on the upper slope of the northern Labrador Shelf, this
study specifically aimed to examine: (i) differences in ocean dynamics and seawater
properties,, (ii) the annual dynamics of near-bed environmental and hydrodynamic conditions,
and (iii) differences in organic matter flux and isotopic signatures for sponges and associated
macrofauna. To this end, data on regional oceanography of the Northern Labrador shelf was
collected from CTD casts, Argo float profiles, and surface drifter buoys. Bottom
hydrodynamic- and environmental conditions were assessed using two year-long benthic
lander deployments. Organic matter fluxes were measured with sediment traps, and benthic
macrofauna was sampled by two rock dredge deployments. This study is the first to collect
year-long hydrodynamic and environmental data simultaneously at a high- and a low-biomass
sponge ground.

## 2   Material and methods

### 2.1   Oceanographic setting and the study area

The study area comprises the northern Labrador Shelf and upper slope and extends from the
south-eastern Hudson Strait outflow region to the base of the Labrador slope (Figure 1A). This
region is known for intense mixing and water mass transformation (Dunbar, 1951; Kollmeyer
et al., 1967; Griffiths et al., 1981; Drinkwater and Jones, 1987) and four distinct flow
components can be identified (Figure 1A; Smith et al., 1937; Yashayaev, 2007; Straneo and
Saucier, 2008; Curry et al., 2011, 2014): first, the cold and relatively fresh Arctic outflow,
passing through the Davis Strait via the Baffin Island Current (BIC), enters the region from the
north as Arctic Water (AW) and Baffin Bay Water (BBW; Sherwood et al., 2021); second, the
West Greenland Current (WGC) approaches our study site from the northeast; third, Irminger
Water (IW), a warmer and saltier water mass, can often be seen below the WGC, usually >150
m depth; and fourth, Hudson Strait outflow water which enters the region from the west. The
resulting aggregated boundary current joins the Labrador Current (LC) flowing southward
along the Labrador Shelf/slope, effectively forming and maintaining a baroclinic transition
between the less-saline shelf water and the more-saline deep-basin water (Yashayaev, 2007).
The northern Labrador Shelf hosts multiple sponge grounds with contrasting sponge
community composition, density, and biomass (Kenchington et al., 2010; Dinn et al., 2020).
We selected a high-sponge-biomass site (HSB; 410 m depth) in the north and a low-sponge-
biomass site (LSB; 558 m depth) in the south of the study area (Table S1; Figure 1B),
approximately 130 km apart.
The substrate at the HSB lander location consisted mostly of pebbles, cobbles, and boulders
(Figure 2 A & B; Kenchington et al., 2010; Dinn et al., 2020) and a visual assessment of the
sediment type at the LSB lander location suggested the dominance of gravel (Coté et al., 2019).
The seafloor at the HSB lander was characterized by large-sized massive demosponges (e.g.
*Geodia* spp.), glass sponges (e.g. *Asconema* spp.), and large gorgonian corals (*Primnoa*
*resedaeformis*; Figure 2 A & B; Kenchington et al., 2010; Dinn et al., 2020). The benthic
community at LSB consisted mostly of small specimens of corals including *Anthomastus* sp.,
as well sponges as *Polymastia sp, Craniella* sp., *Axinella sp,* and possibly *Mycale sp.* (Figure
2 C & D; Coté et al., 2019).
The HSB lander was located on the shelf on a 2° slope and slope aspect was directed northwest
at 60°. The LSB lander was located on the upper slope, east of the shelf break, on a 7° slope
and aspect was directed southeast at 105° (Figure S1).

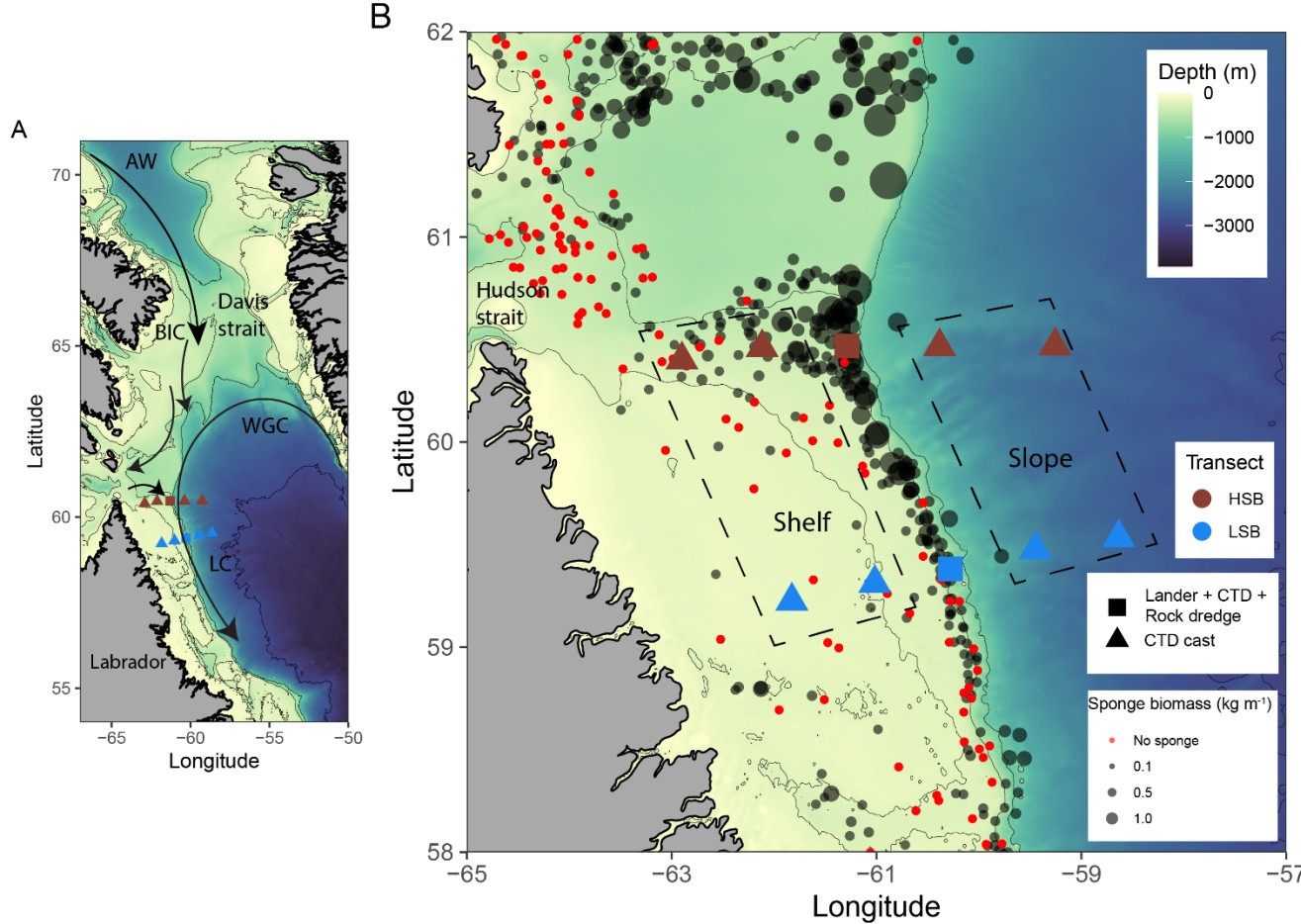


*Figure 1: Map of the study area with (A) the general circulation pattern (Curry et al., 2014). Cold Artic Water (AW) flows*
*southward through the Davis Strait and continues as the surface-intensified Baffin Island Current. The warmer, more saline*
*West Greenland Slope Current (WGC) of North Atlantic origin largely follows the continental slope in the depth range 150 –*
*800 m and is deflected westward at approximately 64° N. Cold and fresh water leaves Hudson Strait and joins the BIC and*
*WGC to form the offshore branch of the Labrador Current (Straneo and Saucier, 2008). (B) Location of lander deployments*
*and CTD-casts, with sponge biomass (in kg m⁻¹) based on Kenchington et al. (2010). Dotted line boxes indicate the shallow*
*shelf and deeper slope stations at both sites. HSB = high-sponge-biomass transect (red symbols), LSB = low-sponge-biomass*
*transect (blue symbols).*

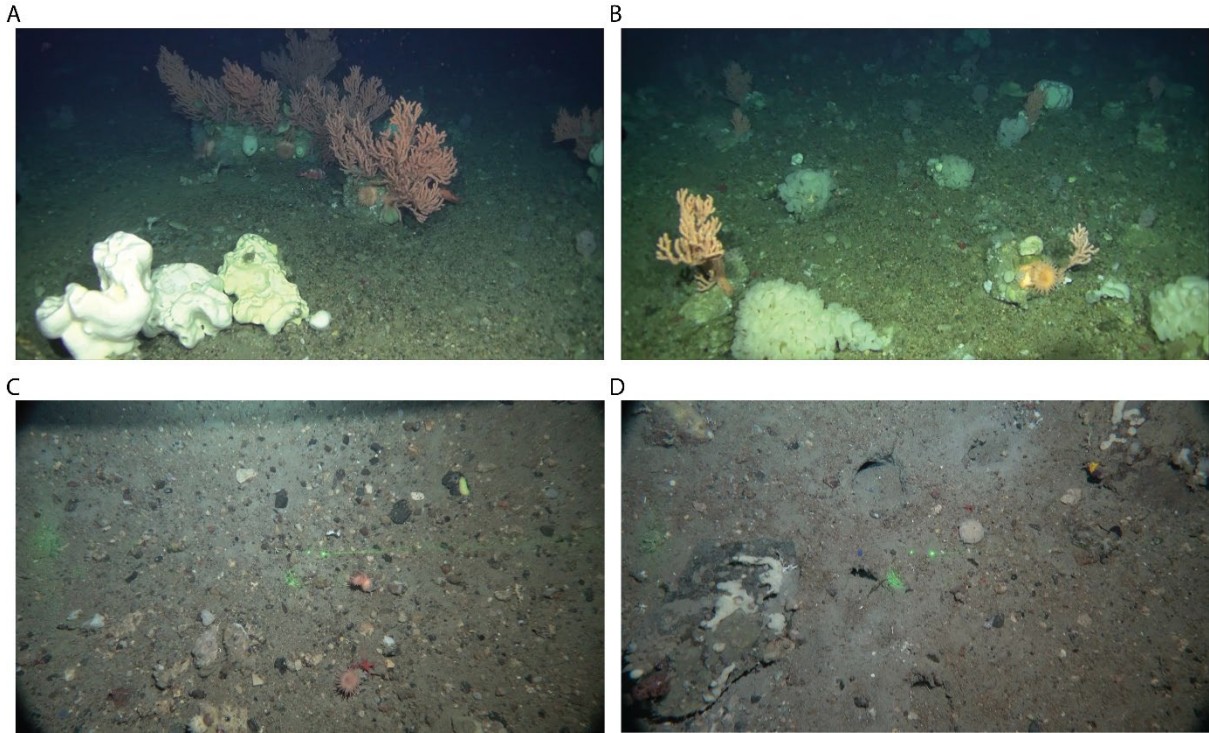

*Figure 2: Images of benthic lander deployment sites, at the high-sponge-biomass lander site (HSB; A,B) and low-sponge-biomass lander site (LSB; C, D). ROV image credits: ArcticNet/Canadian Scientific Submersible Facility (CSSF)/Department of Fisheries and Oceans (DFO). Laser points in panel C & D are 6 cm apart.*

## 2.2 Sampling methodology

### 2.2.1 Near-bed lander deployment

Landers were deployed during research cruise Amundsen 2018 leg 2c (27 July 2018) and retrieved during research cruise Amundsen 2019 leg 1b (1 & 2 July 2019). The landers were each equipped with a 2 MHz single point measurement ADCP (upward-looking, Nortek Aquadopp), a sediment trap, and a combined optical backscatter sensor (OBS) for turbidity and fluorescence (Wetlabs ECO-FLNTU; Table S1).

The ADCPs collected data on pressure, water velocity, echo intensity (ABS; acoustic backscatter signal), and water temperature at a 10 minute interval. Furthermore, the built-in accelerometer and magnetometer in the ADCPs collected data on heading, pitch, and roll. The ADCP was mounted 2 m above the bottom, the blanking distance was 1.14 m. Velocity data were recorded in beam coordinates and transformed in MATLAB to ENU coordinates (East, North, Up) after recovery using the transformation matrix provided by the manufacturer. The 2 MHz ADCP have a lower particle size detection limit of 12 μm in diameter, and a maximum sensitivity for particles of 242 μm diameter (Haalboom et al., 2021, 2023). The combined optical backscatter sensor for turbidity and fluorescence was programmed to

measure every 10 minutes over the one-year period. The sediment trap (PPS 4/3, Technicap
Inc.) with a surface area of 0.05 m$^2$ was equipped with twelve bottles for suspended particulate
matter collection and with the aperture mounted at 2 m above the bottom. Collection started at
15/08/2018 and lasted until the end of the deployment. Different time intervals of bottle rotation
were set to increase sampling resolution during spring and summer months. The bottles rotated
every 15 days from mid-August to mid-September 2018, every 30 days from mid-September
to mid-November 2018, every 60 days from mid-November to mid-March 2019, then every 30
days from mid-March to mid-May 2019, and every 15 days again from mid-May to mid-July
2019. Prior to deployment, a 4% solution of formalin in brined seawater (40 psu) was added to
each bottle.

### 2.2.2 Water column and benthic sampling

Conductivity-Temperature-Depth (CTD) casts were performed over two cross-shelf transects
crossing the LSB and HSB lander sites (Coté et al., 2018; Figure 1B; Table S1). Two CTD
casts were carried out on the continental shelf and three on the continental slope, where the
third or middle cast was performed above each benthic lander deployment. The CTD-Rosette
water column profiling and sampling package was equipped with a Seabird SBE 911*plus*
system, which contained sensors to measure temperature (Seabird SBE 3plus), conductivity
(Seabird SBE 4), pressure (Paroscientific Digiquartz®), dissolved oxygen (Seabird SBE 43),
fluorescence (Seapoint), and a rosette water sampler with 12 Niskin bottles (12L each). CTD
data were processed and "cleaned" with the *Sea-Bird SBE Data Processing* software (Guillot,
2018). Water samples were taken from Niskin bottles at five depths (5 m, 50 m, mid-water,
100 m above bottom, 10 m above bottom) for the determination of nutrients ($NH_4^+$, $NO_2^-$ +
$NO_3^-$, $PO_4^{3-}$, $SiO_2$), and suspended particulate matter (SPM).
Benthic macrofauna samples for stable isotope analysis were collected at the two lander
locations using a rock dredge on retrieval of the benthic landers (Coté et al., 2019; Table S3).
A description of the species found at the two locations can be found in Coté et al. (2019). The
rock dredge (7 mm mesh size) was deployed in "drift" mode at HSB, with a maximum speed
of two knots (~4 km h$^{-1}$) for 10-20 minutes, and "tow" mode at LSB, with the ship moving at
one knot for 10 minutes. During CCGS Amundsen cruise 2019 leg1B, it was the first time that
a rock dredge was operated on this research vessel, and therefore different operational modes
of deployment were tested. At the LSB lander station, the rock dredge collected lots of soft
sediment, and therefore "drift" mode was used. On deck, the dredge was rinsed, and the catch
was subsampled and deposited in fish totes (64 L). The remaining material was sieved through
a 2 mm mesh for analysis of invertebrates and fishes. The total catch was photographed and
preserved for species identification and quantification. Samples for stable isotopes were frozen
(-20 °C) for further analysis at the Netherlands Institute for Sea Research (NIOZ).
### 2.2.3  Regional oceanography, sea-ice cover, and bottom temperature/salinity profiles
To explore the regional oceanography on the northern Labrador Shelf and upper slope,
vertical Argo float profiles collected within the water depth range 330 - 2575 m (Figure S3)
were extracted from the NOAA NODC World Ocean Dataset and profiling Argo float Global
Argo Data Repository archives (Kieke and Yashayaev, 2015; Yashayaev and Loder, 2017)
using the approach of Kenchington et al. (2017). We used Argo float profile data (N = 1472)
collected between 2005 and 2022 to determine the seasonal variability in temperature and
salinity along the northwest Labrador shelf break. Specifically, seawater properties of the
corresponding water layers to the depth of the benthic landers (LSB = 350 – 450 m, HSB = 550
– 650 m depth) were assessed. We report the mean temperature and salinity values binned per
water layer. Argo float profiles below ~59° N latitude were considered LSB and above as HSB.
Temperature and salinity values were detrended for interannual variability using an $8^{th}$ degree
least-square polynomial fit. Time-average surface currents were derived from trajectories of
satellite-tracked surface drifting buoys (drifters) deployed within the NOAA Global Drifter
Program during 2000–2020 (Centurioni et al., 2019). The trajectories were obtained from
delayed-mode hourly data and real-time variable time-step data (Elipot et al., 2016, 2022). The
drifter data were temporally interpolated into 15-min time intervals, binned hourly, and a low-
pass filter was used to remove tidal and inertial oscillations. Then, the surface velocities were
binned into a 1/3° grid. The drifter-derived surface currents reveal well-defined large-scale
cyclonic circulation of the Labrador Sea, recirculation gyres, and mesoscale circulation
features.
Sea-ice cover above the two benthic landers was extracted from weekly ice charts (Canadian
Ice Service, 2022). Slope angle and aspect was estimated for each lander by taking the wider
topography into account (Figure S1; Gille et al., 2004). Along-slope and across-slope bottom
velocities are derived from the bottom current direction, slope aspect, and bottom horizontal
current speed.
### 2.3  Laboratory analysis
Water column nutrient concentrations were analysed with a SEAL QuAATro analyser (Bran +
Luebbe, Norderstedt, Germany) following standard colorimetric procedures. SPM samples
were freeze-dried, weighed, and analysed for organic carbon content and total nitrogen content.
Sediment trap samples were filtered through a 1 mm sieve to remove large particles and
swimmers, then split into five sub-samples using a McLane WSD-10 rotary splitter, rinsed with
demineralized water to remove salts and formalin and subsequently freeze-dried and weighed
(Newton et al., 1994; Mienis et al., 2012). Lipids were extracted and analysed following the
method of Kiriakoulakis et al. (2004). Briefly, samples were spiked with internal standard
(5$\alpha$(H)-cholestane), extracted by sonication in dichloromethane:methanol (9:1; x3). The
solvent was removed and samples were first trans-methylated (Christie, 1982) and then treated
with bis-trimethylsilyltrifluoroacetimide: trimethylsilane (99:1; 30-50 $\mu$L; 60 °C; 1 h) prior to
analysis by gas chromatography-mass spectrometry (GCMS). GCMS analyses were conducted
using a GC Trace 1300 fitted with a split-splitless injector and column DB-5MS (60m x
0.25mm (i.d.), with film thickness 0.1 $\mu$m, non-polar stationary phase of 5% phenyl and 95%
methyl silicone), using helium as a carrier gas (2 mL min$^{-1}$). The GC oven was programmed
after 1 minute to rise from 60°C to 170°C at 6°C min$^{-1}$, then from 170°C to 315°C at 2.5 °C
min$^{-1}$ and was then held at 315 °C for 15 min. The eluent from the GC was transferred directly
*via* a transfer line (320 °C) to the electron impact source of a Thermoquest ISQMS single
quadrupole mass spectrometer. Typical operating conditions were: ionisation potential 70 eV;
source temperature 215°C; trap current 300 $\mu$A. Mass data were collected at a resolution of
600, cycling every second from 50– 600 Daltons and were processed using Xcalibur software.
Compounds were identified either by comparison of their mass spectra and relative retention
indices with those available in the literature and/or by comparison with authentic standards.
Quantitative data were calculated by comparison of peak areas of the internal standard with
those of the compounds of interest, using the total ion current (TIC) chromatogram. The
relative response factors of the analytes were determined individually for 36 representative
fatty acids and sterols using authentic standards. Response factors for analytes where standards
were unavailable were assumed to be identical to those of available compounds of the same
class.
Sponges and other benthic fauna collected using a rock dredge were subsampled on-board the
CCGS Amundsen, as parts of the specimens' bodies were used in separate studies and parts for
isotopic analysis in this study. In the laboratory, the collected fauna was freeze-dried and
homogenized with a pestle mortar/ball mill. Subsamples (*ca.* 10 mg) were transferred into
silver cups and acidified by addition of dilute HCL (2%, 5%, and 30%) to remove carbonates.
Organic carbon and $\delta^{13}$C were analysed on acidified subsamples, and total nitrogen and $\delta^{15}$N
was determined on non-acidified subsamples using an Electron Analyser coupled to an Isotope
Ratio Mass Spectrometer (Thermo flash EA 1112). $\delta^{13}$C and $\delta^{15}$N isotope values are expressed
in parts per thousand (‰) relative to the international standard Vienna Pee Dee Belemnite and
atmospheric $N_2$ for carbon and nitrogen, respectively. Standard deviation of $\delta^{13}C$ and $\delta^{15}N$
measurements was 0.15 ‰.

## 2.4  Data analysis

### 2.4.1  Data processing

The transformation of beam coordinates to ENU coordinates for the ADCP data was
carried out in MATLAB (MATLAB, 2010), and other data processing steps used R. The
following R packages are used during data analysis: oce, ggplot2, RColorBrewer, cowplot,
knitr, reshape2, RNetCDF, readxl, lubridate, xts, ggalt, tibble, dplyr, clifro, mapdata, metR,
patchwork, tibbletime, readr, viridis, biwavelet, signal, astsa, terra, and raster (Wickham, 2007,
2016; Grolemund and Wickham, 2011; Neuwirth, 2014; signal developers, 2014; Michna and
Woods, 2019; Pedersen, 2019; R Core Team, 2019; Wickham and Bryan, 2019; Wilke, 2019;
Kelley and Richards, 2020; Stoffer, 2020; Vaughan and Dancho, 2020; Xie, 2020; Lovelace et
al., 2022). Statistics are presented as means ± standard deviations.

### 2.4.2  Benthic lander analysis

Occasionally, pitch and roll data from the ADCP sensor at HSB were shifted for a small period
of the deployment, implying the lander was occasionally moving slightly (Figure S3).
Pitch/heading/roll was almost identical before and after these disturbances. Furthermore, the
ADCPs correct for the pitch/roll/heading of the respective device when producing the raw beam
data.  Removing datapoints during disturbance did not change the outcome of any of the
analyses, statistical tests, or descriptive statistics and therefore datapoints were retained in the
HSB lander time series.
Chl-$a$ concentration (in µg $L^{-1}$) and turbidity (in Nephelometric Turbidity Unit; NTU) were
calculated from ping counts as described in the manual of the manufacturer.
Spectral analyses of lander data based on a Fourier transformation (Bloomfield, 2004) were
performed to examine recurring patterns or periodicity in the time-series data (e.g. Shumway
et al., 2000; Bloomfield, 2004). Prior to these analyses, time series data were smoothed using
modified lowpass Daniell filters (Bloomfield, 2004), to remove periodicities shorter than 3
hours. The magnitude and direction of ADCP-recorded tidal currents were analysed with least-
squares harmonic analysis.

### 2.4.3   Critical-slope and comparing barotropic with baroclinic tides

Internal tides are generated by the barotropic tide interacting with sloping bottom topography and can have a profound influence on the thermohaline structure and local mixing processes. Internal tides are found at complex deep-sea topographic features such as continental shelves, ridges, seamounts and canyons (e.g., Cacchione et al., 2002). Internal tide – topography interactions can be classified by the slope parameter $\alpha$ / c (St Laurent and Garrett, 2002; Cacchione et al., 2002). The internal wave slope c is calculated from $c = \sqrt{\frac{\omega^2 - f^2}{N^2 - \omega^2}}$, with tidal frequency $\omega$ =1.4053e-4 rad s$^{-1}$ (representing the dominant M2 tidal component) and local inertial frequency $f$ (s$^{-1}$). The Brunt-Väisälä frequency $N^2$ (rad s$^{-2}$) was calculated as the mean value (1.4228 * 10$^{-5}$ rad s$^{-2}$) from all CTD stations and depths below the deep pycnocline at 250 m or from bottom values at shallower profiles. The topographic slope $\alpha$ was calculated from the maximum depth gradients in latitude and longitude based on GEBCO_2023 data (GEBCO Bathymetric Compilation Group, 2023). At critical or near-critical slopes ($\alpha \approx$ c), the internal tide is locally amplified and vertical mixing is intensified. At subcritical slopes ($\alpha <$ c), internal waves pass the topographic slope without being locally modified. At steeper supercritical slopes ($\alpha >$ c), internal waves are reflected into deeper waters.

Bottom currents and direction were compared to model derived barotropic tidal currents, retrieved from the Oregon State University (OSU) Tidal Inversion Software (OTIS; Egbert and Erofeeva, 2002).

## 3   Results

### 3.1   Seawater properties over the northern Labrador Shelf and upper slope

The CTD casts, performed in July 2018, revealed different seawater properties between the two transects (Figure 3; Figure S4). The surface water at the time of survey was relatively warm (2 – 6 °C) and fresh (31.2 to 33.8 psu) showing an  offshore increase in temperature and salinity. From the surface to the depth of 20-70 m, depending on the transect and location, temperature decreased to sub-zero or near-zero at the shelf locations, to 3 °C at the slope locations, and then increased again to 2.8 °C at 250 m depth on the shelf and to 4.3°C at 150 m on the slope. A cold intermediate layer was visible at all profiles between 50 – 150 m depth. Salinity increased nearly monotonically with depth up to the pycnocline across all stations. The stations at LSB were more saline overall than those at the matching water depths on the HSB transect. Buoyancy frequency showed peak values at the upper- and lower boundaries of the above described cold intermediate layer at both transects (Figure S4F).

The oxygen concentration was highest in the surface waters (0 – 50 m) on the shelf and decreased with depth at all CTD stations (Figure 4A). Although oxygen concentrations were still generally high, the bottom oxygen concentrations at the lander stations were, for both transects, relatively depleted compared to the deep water CTD transects at similar depths. Concentrations of nitrate, phosphate, and silicate were lowest above the thermocline and increased with depth, while ammonium and nitrite were higher near the surface than at depth (Figure 4B & C, Figure S5). The HSB station exhibited relatively high nitrate, phosphate, and silicate concentrations at 10 and 100 metres above bottom compared to similar depths at shelf and deep stations (Figure 4B & C, Figure S5). This increased nutrient concentration in the bottom waters was also apparent for silicate at the LSB station (Figure 4C), but not for nitrate (Figure 4B). Chl-*a* profiles showed a deep chlorophyll maximum along both transects at 50 m (Figure 4C), and near-zero concentrations in the bottom waters (Figure S4D). Particulate organic carbon (POC) concentrations were highest in the surface waters (8 – 38 µmol POC L$^{-1}$) and on the shelf (Figure S6:). POC concentrations decreased with depth, and concentrations 10 m above bottom were 1.48 µmol POC L$^{-1}$ at HSB, and 5.95 µmol POC L$^{-1}$ at LSB.

Surface water above the benthic lander locations was partly ice-covered from December to June, but both sites were located at the sea-ice border in the study area and ice cover was highly variable (Figure S11). Only during January ice-coverage was above 70% at both sites. Both locations showed a short ice-free period in February and March. During the spring bloom, between the end of March and early May, sea-ice coverage tended to be higher at HSB than at LSB (Figure S11D).

## 3.2 Regional oceanography and seasonal temperature patterns

Surface buoy drifter data showed that the HSB lander was located in an area where three (surface) currents converge (Figure 5A). Strong surface currents (>0.24 m s$^{-1}$ on average) carry water from the Hudson strait towards the Labrador shelf break, where this current meets two others that, respectively, flowed toward the HSB site from the north and northeast. On convergence, the currents followed the bathymetry of the Labrador shelf break or upper slope southwardly.

The seawater in the region of HSB was warmer and less saline than around LSB for both depth ranges within which the landers were deployed (Figure 5B & C; Figure S7). Bottom water temperature shows a steeper decrease in February at LSB compared to HSB (Figure 5C). Temperature and salinity show higher scatter at HSB than LSB throughout the season, but variability in temperature is highest at HSB in February/March (Figure 5B & C; Figure S7).

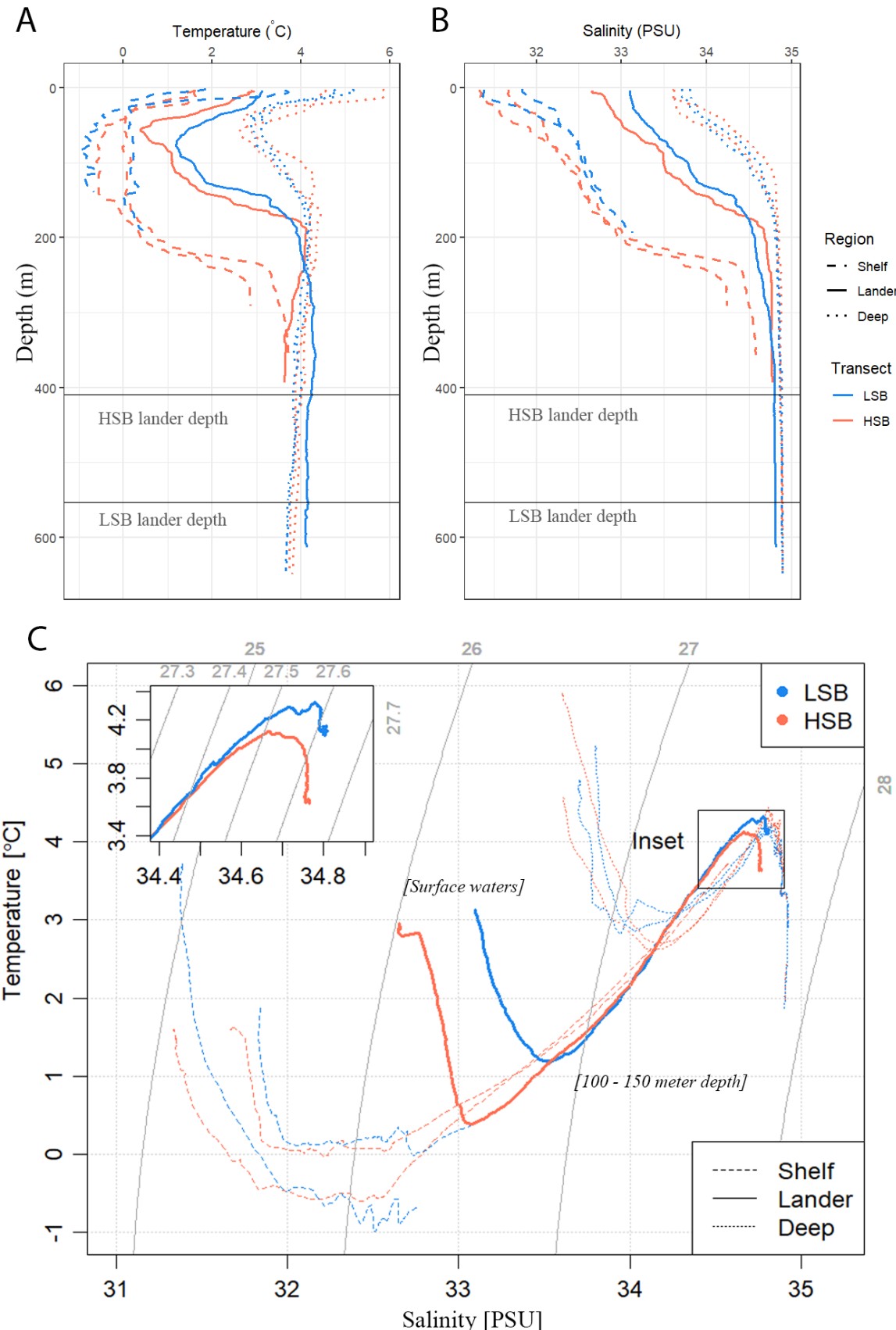

393

Figure 3: Hydrographic conditions in the study area: (A) temperature, (B) salinity and (C) temperature – salinity (TS) plots
for the two transects. LSB = low-sponge-biomass, HSB = high-sponge-biomass. Depths of landers are indicated by the
horizontal grey lines in A and B. Temperature and salinity profiles in A and B only show top 600 m, while TS plots include the
entire water column. The thin grey lines in subplot C resemble isopycnals.



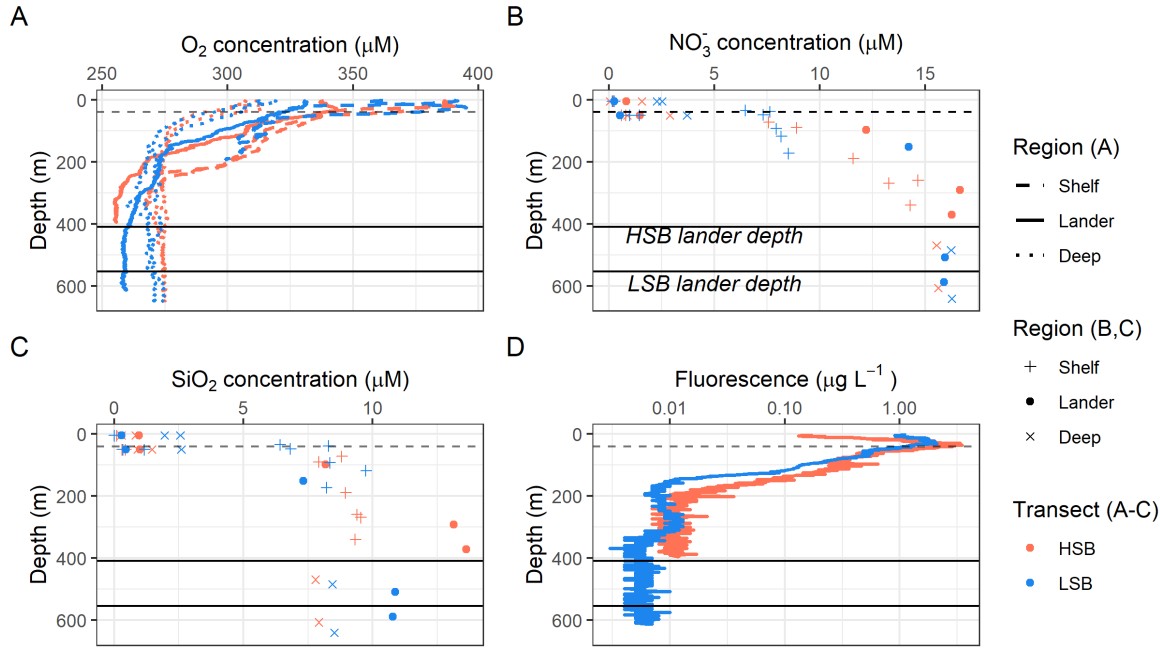


Figure 4: Oxygen (A), nitrate (B), silicate (C) concentration profiles for the two transects, and D) fluorescence profiles for
the two CTD casts above the two lander locations. HSB = high-sponge-biomass site, LSB = low-sponge-biomass site. Black
lines indicate lander depths, dashed line indicates thermocline.



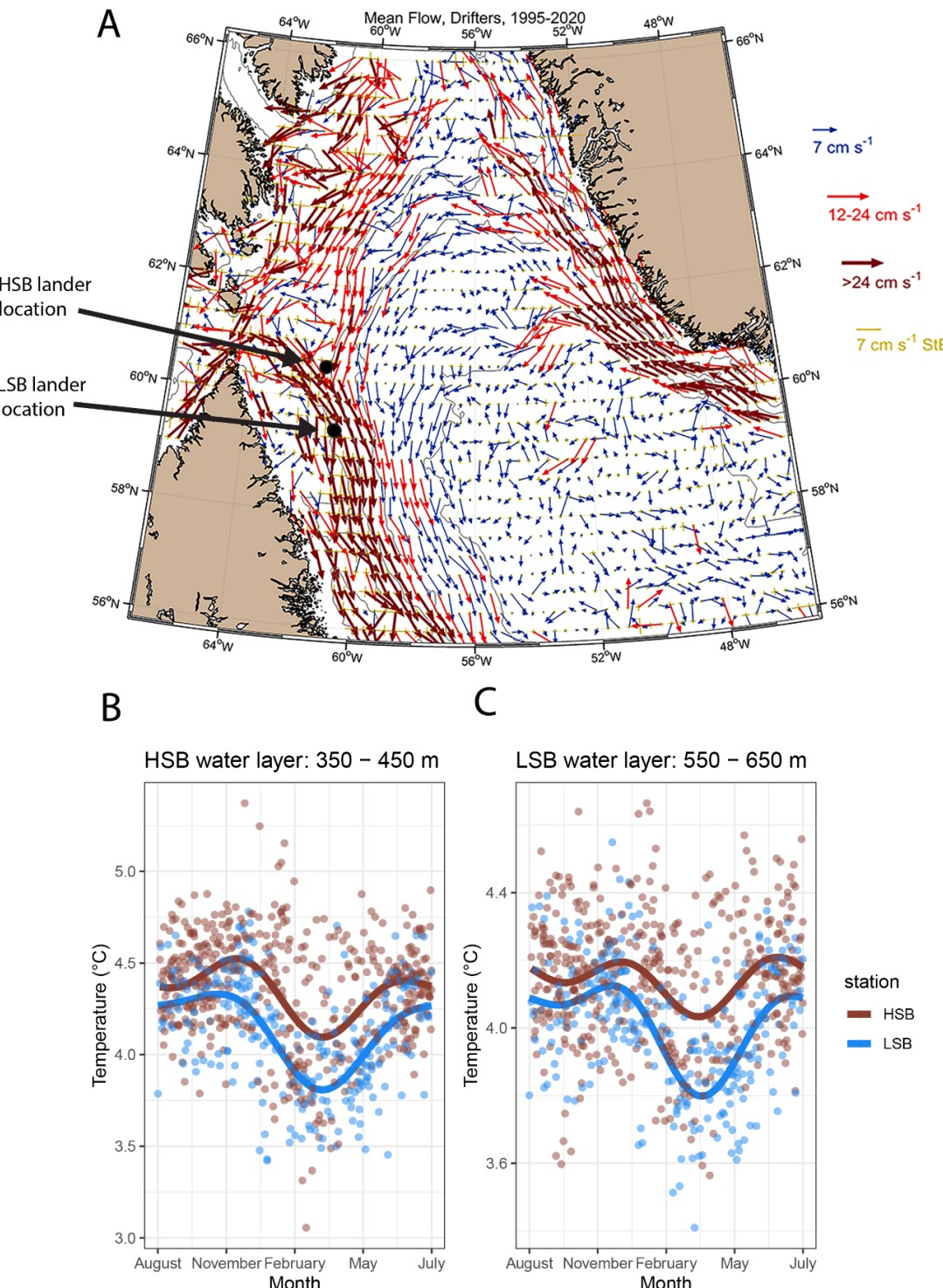


Figure 5: A) general surface circulation pattern in the Labrador Sea based on drifter buoy data spanning from 1995 - 2020.
Arrows indicate mean direction, colours and length of arrow present the strength of the mean flow, the yellow arrows
present the standard error of the flow over 1995 – 2020. The lander locations are indicated by the coloured dots. B & C)
seasonal temperature, from Argo float profiles, of the water layer in which HSB/LSB lander was located. Dots represent
individual water-layer-binned temperature measurements vs. date of the year. The lines are a smoothed fit that show the
seasonal pattern.

## 3.3   Year-long near-bottom measurements

### 3.3.1   Near-bottom current velocities

In general, bottom current speeds were higher at the HSB compared to the LSB station (Table 1; Figure 7). General current direction was south-easterly at HSB and south-south-westerly at LSB (Figure 6). Vertical velocity ($w$) was on average upward and comparable between HSB and LSB, but the range in vertical velocity was higher at HSB (-0.35 to 0.32 m s$^{-1}$) compared to LSB (-0.11 to 0.21 m s$^{-1}$; Figure 7C). Bottom horizontal currents were twice as high at HSB than at the LSB (Table 1), and peak bottom horizontal current speeds were 0.75 m s$^{-1}$ (HSB) and 0.65 m s$^{-1}$ (LSB), with the third quantile at 0.33 m s$^{-1}$ (HSB) and 0.18 m s$^{-1}$ (LSB).

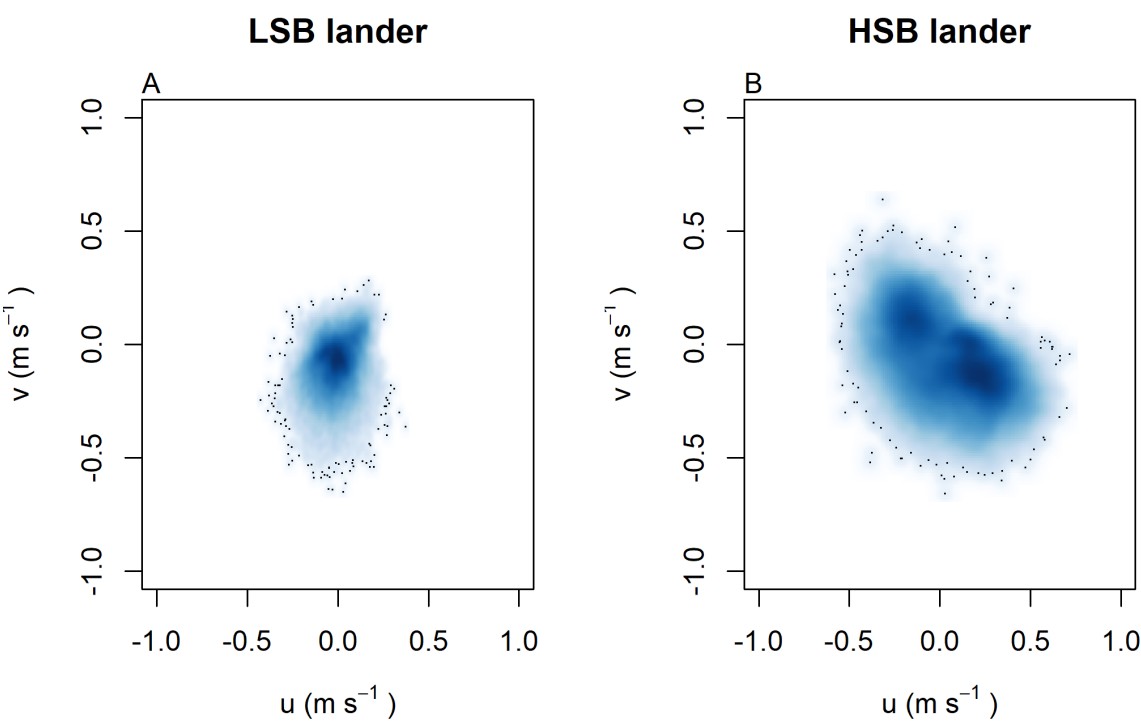

Figure 6: horizontal current velocities at A) LSB lander and B) HSB lander.

### 3.4   Near-bottom environmental conditions

Bottom temperature was slightly warmer at HSB compared to LSB and increased at both sites (0.2 – 0.3 °C) during December and January (Figure 9). The benthic lander temperature aligned well with the seasonal temperature pattern retrieved by Argo float profiles (Figure 5 B & C). Turbidity measured by ABS was similar for the two stations (Table 1; Figure 9 B) and showed higher values in winter months. Chl-$a$ remained low from October to February/March values started to increase for both landers (Figure 9 C). Bottom chl-a concentrations started to increase

after short ice-free period mid February and mid March Figure 9C; Figure S11D).  The HSB
station showed highest chl-a concentrations from mid-March to the end of May, while at the
LSB station increased concentrations were observed from mid-March to early May.
Turbidity measured by OBS was elevated at HSB from February to April, and at LSB from
December to January. The higher variability in chl-$a$ and turbidity at the LSB site over the year
(Table 1) was caused by several peaks in chl-$a$ and turbidity that were an order of magnitude
higher than average values (Figure S8).
During several periods in the year-long time-series, turbidity measured by the ABS increased
at the turning of the tide and at high south-easterly current velocities at HSB (see e.g. Figure
10F). Strong along slope (southerly) bottom currents increased ABS turbidity and OBS
turbidity at LSB (Figure 10F). Cross-and along slope water transport influenced bottom
temperature. At the HSB lander, for example, in the first week of September, temperature
decreased when the current was directed northwest and increased when the current was directed
southeast (Figure 10 A-E).

*Table 1: Benthic lander mean and standard deviations over the year-long deployment period. Values are given as mean ±*
*standard deviation. HSB = high-sponge-biomass lander, LSB = low-sponge-biomass lander. ABS = acoustic backscatter*
*signal. OBS = optical backscatter signal*

| Variable | HSB | LSB |
|---|---|---|
| $u$ (eastward velocity; m s$^{-1}$) | 0.05 ± 0.22 | -0.01 ± 0.09 |
| $v$ (northward velocity; m s$^{-1}$) | -0.07 ± 0.16 | -0.09 ± 0.11 |
| $w$ (vertical velocity; m s$^{-1}$) | 0.03 ± 0.05 | 0.02 ± 0.03 |
| Bottom current speed (m s$^{-1}$) | 0.26 ± 0.14 | 0.14 ± 0.08 |
| Temperature (°C) | 3.70 ± 0.17 | 3.58 ± 0.17 |
| Daily temperature variability (Δ°C d$^{-1}$) | 0.25 ± 0.16 | 0.17 ± 0.1 |
| Turbidity by ABS (counts) | 98.1 ± 9.8 | 96.6 ± 11.0 |
| Chl-$a$ concentration (µg L$^{-1}$) | 0.11 ± 0.03 | 0.08 ± 0.10 |
| Turbidity by OBS (NTU) | 0.20 ± 0.10 | 0.21 ± 0.27 |
| Across slope velocity (m s$^{-1}$) | 0.01 ± 0.13 | -0.01 ± 0.01 |
| Along slope velocity (m s$^{-1}$) | -0.08 ± 0.23 | -0.09 ± 0.11 |


3.4.1   Tidal analysis of bottom currents and environmental conditions
Bottom current speeds showed semi-diurnal and spring-neap tidal patterns, with a peak every
fortnight for both sites (Figure 7 C; Figure 8 B; Figure 10). The major axes of the semidiurnal
tidal ellipses were directed in a northwest-southeast direction at HSB and a north-south
direction at LSB (Figure 7D). The tidal analysis presented in Table 2 and Figure 8 shows
notable differences in tidal characteristics between the LSB and HSB lander locations. While
semidiurnal tidal harmonics predominate at both locations, the semi-major axis at the HSB site

 is approximately four times larger than the corresponding value at the LSB site. Moreover,

there is a significant discrepancy between the modelled and observed main semidiurnal tidal

harmonics (M2) at the HSB site, particularly in terms of magnitude and tidal ellipse

eccentricity. This indicates that the dominant barotropic semidiurnal tide (M2) is altered at the

HSB site, leading to strongly rectified near-bottom baroclinic tidal currents. There are no

substantial differences between the modelled (barotropic) and observed S2 tidal currents,

except for the tidal ellipse eccentricity at the LB site, likely due to the depth difference between

the model and observations at this location. Furthermore, spectral density for the HSB bottom

current components also peaked at shorter frequencies (3-6 h) and at the fourteen-day spring-

neap tide (Figure 8B). In addition, a superimposed seasonal pattern can be seen at both sites,

where the bottom current speed gradually increased from July 2018 to March 2019 and

decreased again from March 2019 to July 2019.

*Table 2: Tidal analysis of velocity time series from the HSB and LSB lander sites based on ADCP measurements and OTIS tidal model analysis. $A_{maj}$ and $a_{min}$ are the semi-major and semi-minor axes of the tidal ellipse and $\varepsilon$ is the eccentricity ($a_{min}/a_{maj}$). OTIS model data represent the barotropic tidal signal, whereas ADCP data show the near-bottom tidal characteristics.*

| LSB – lander data | $a_{maj}$ (cm s$^{-1}$) | $a_{min}$ (cm s$^{-1}$) | $\varepsilon$ ($a_{min}/a_{maj}$) | Water depth (m) |
|---|---|---|---|---|
| M2 | 5.73 | 2.17 | 0.38 | 558 |
| S2 | 1.74 | 0.51 | 0.30 | |
| K1 | 0.65 | 0.05 | 0.08 | |
| O1 | 0.10 | 0.03 | 0.25 | |
| HSB – lander data | | | | |
| M2 | 27.77 | 7.26 | 0.26 | 410 |
| S2 | 9.61 | 2.88 | 0.30 | |
| K1 | 0.88 | 0.44 | 0.51 | |
| O1 | 0.36 | 0.21 | 0.58 | |
| LSB – OTIS tidal model | | | | |
| M2 | 6.08 | 1.48 | 0.24 | 629 |
| S2 | 1.58 | 0.57 | 0.36 | |
| K1 | 0.49 | 0.06 | 0.11 | |
| O1 | 0.18 | 0.01 | 0.04 | |
| HSB – OTIS tidal model | | | | |
| M2 | 40.67 | 19.23 | 0.47 | 425 |
| S2 | 10.45 | 4.47 | 0.43 | |
| K1 | 1.35 | 0.53 | 0.39 | |
| O1 | 0.80 | 0.38 | 0.48 | |

Temperature, chl-*a*, turbidity measured by ABS and OBS, all showed a reoccurring tidal peak,

with higher peaks in spectral density for the semidiurnal periodicity at HSB than at LSB (Figure

8C). Daily temperature fluctuations were higher at HSB than at LSB. During the spring bloom,

bottom chl-*a* concentration increased during strong south-easterly current velocities at HSB

(Figure S10) and showed a periodic reoccurring peak (Figure S11A).





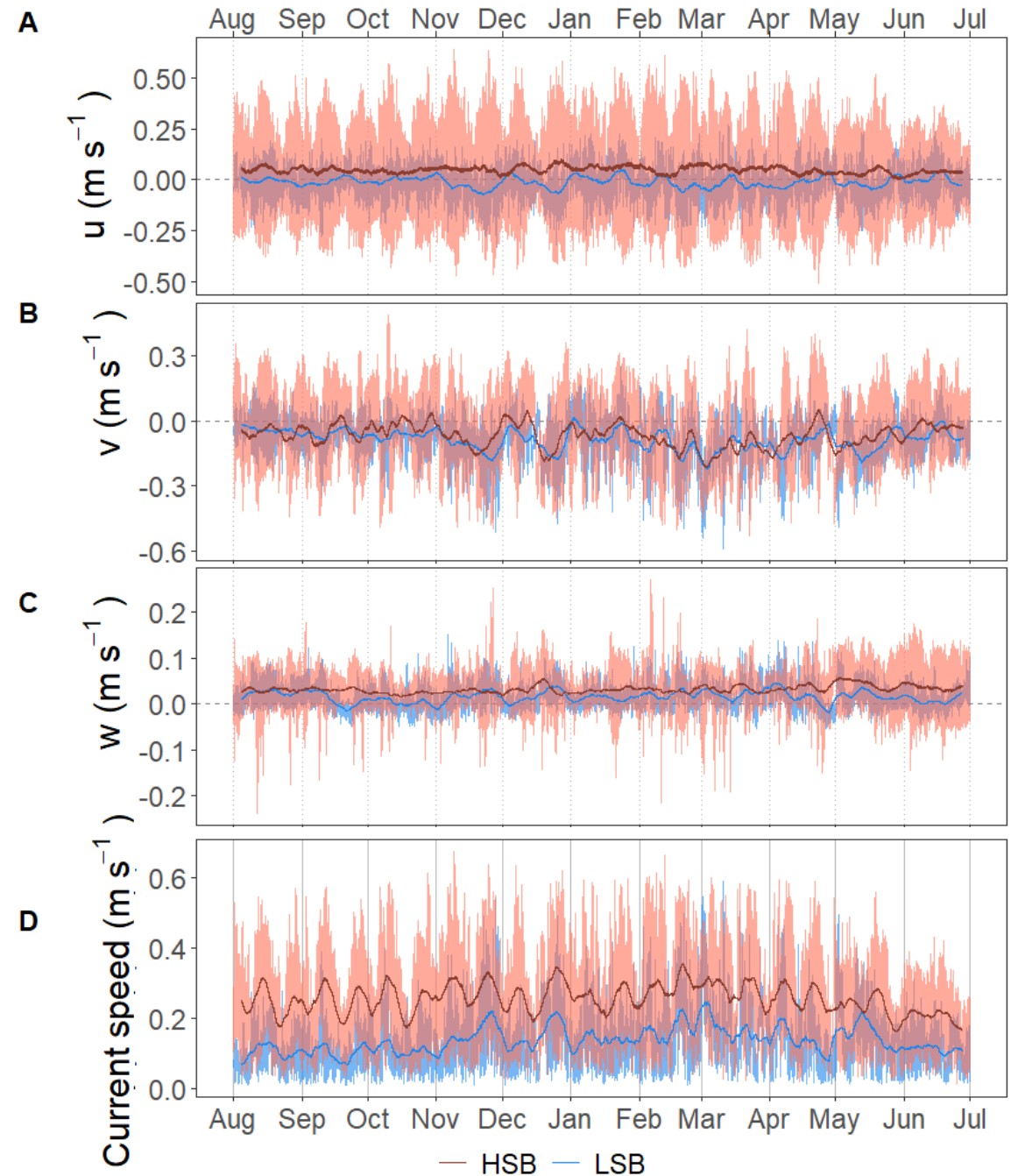


Figure 7: Time series of the flow velocities with eastward u velocity (A), northward v velocity (B), vertical w velocity (C), and bottom current speed (D). Plots show the hourly averaged data as transparent lines and the seven-day rolling means as solid lines.


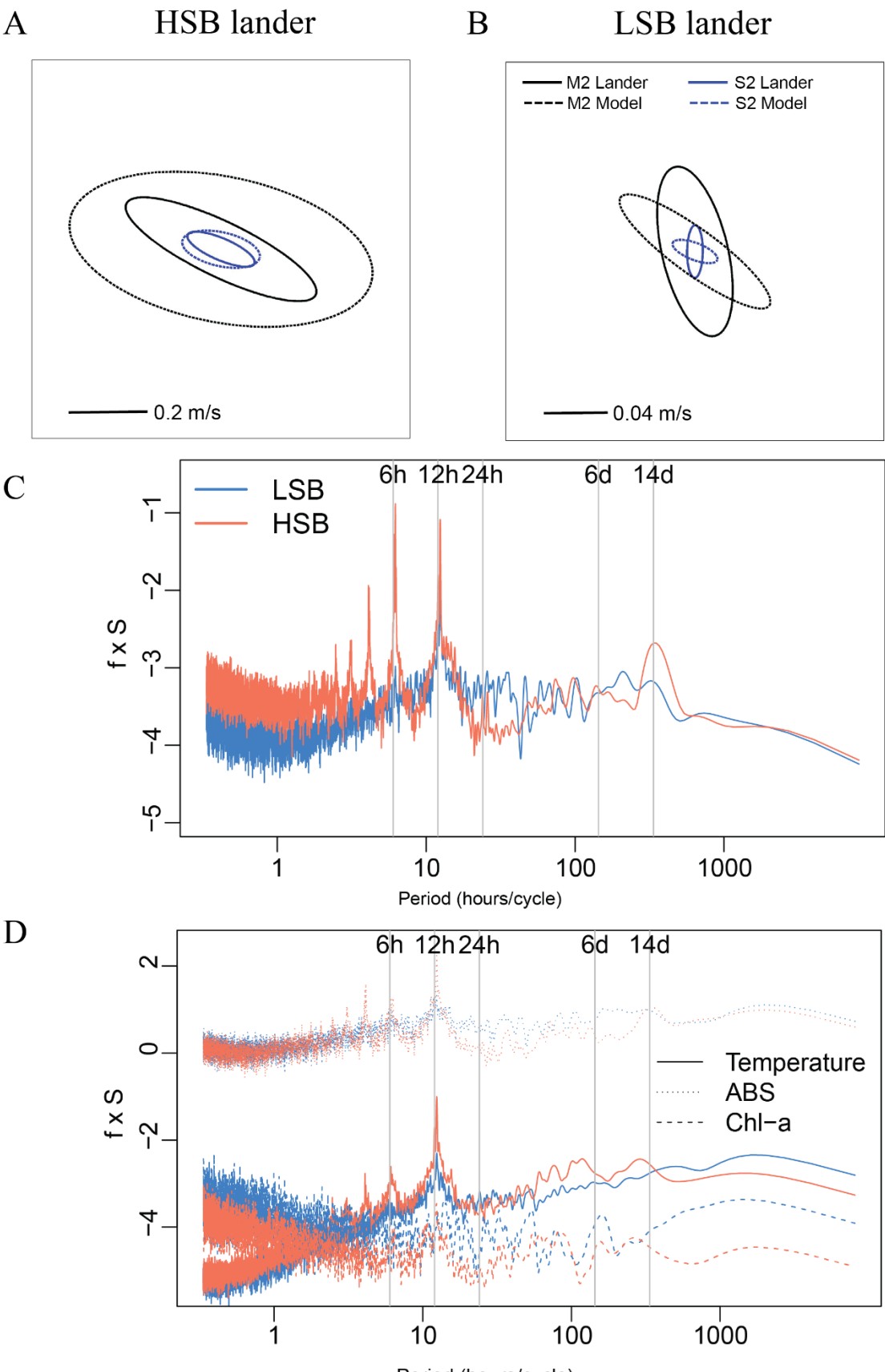

 *Figure 8: A & B) Tidal current ellipses at the HSB and LSB lander sites for the two dominant semidiurnal tidal harmonics*

 *M2 (black lines)and S2 (blue lines) derived from the unfiltered ADCP velocities (solid lines) and the OTIS inverse tidal*

 *model (dashed lines) respectively. Variance preserving spectra for C) bottom current speed, (D) temperature, turbidity by*

 *acoustic backscatter signal (ABS), and chl-a.*

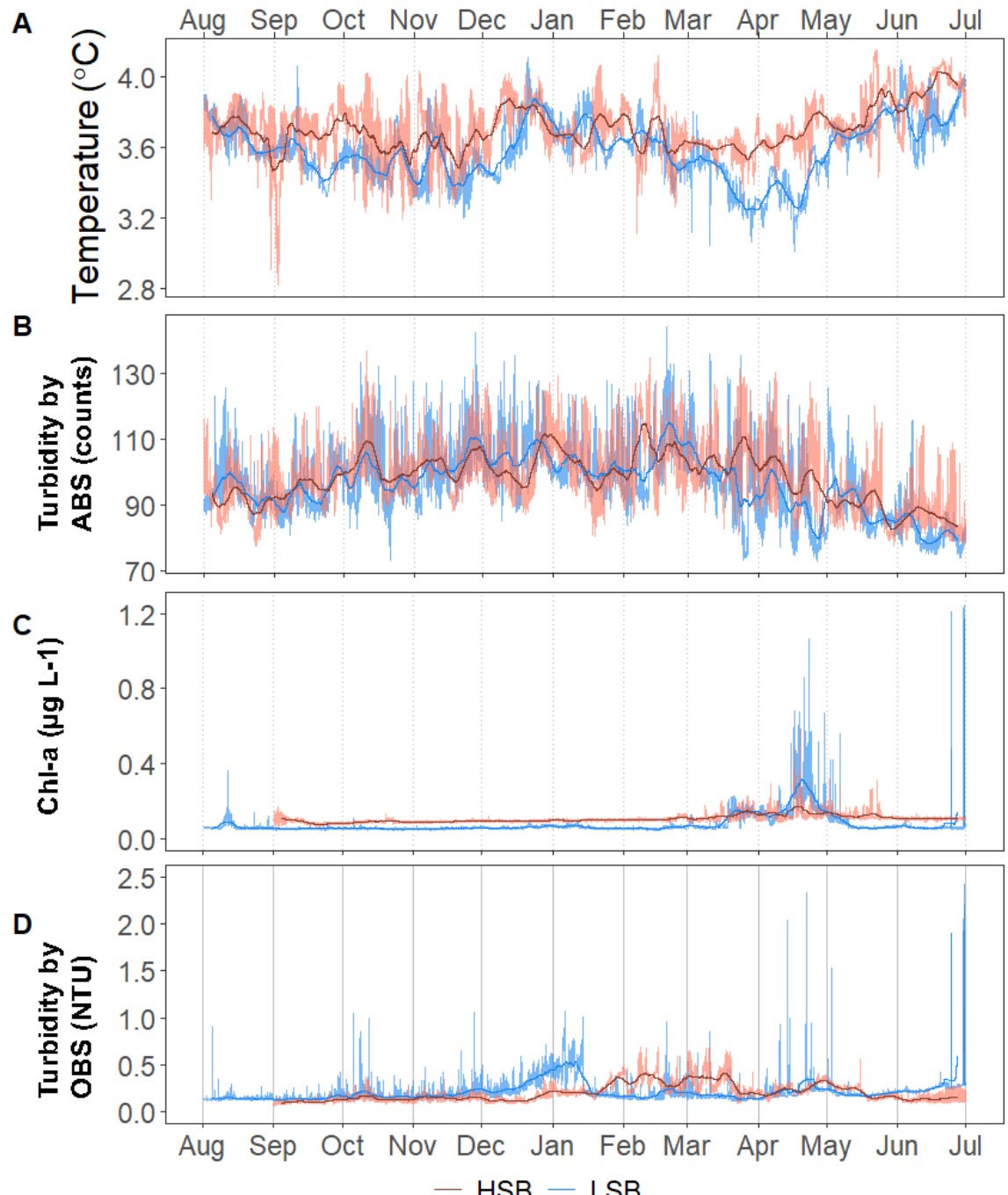


*Figure 9: Time series for temperature in °C (A), Turbidity by acoustic backscatter (ABS; in counts) (B), Chl-a concentration*
*in µg L⁻¹ (C), and turbidity by optical backscatter (OBS) in NTU (D). Plots C and D are limited on the y-axis to 1.25 µg L⁻¹*
*and 2.5 NTU, respectively, for clarity. Chl-a and turbidity by OBS data without the Y-axis cut-offs are plotted in Figure S10.*

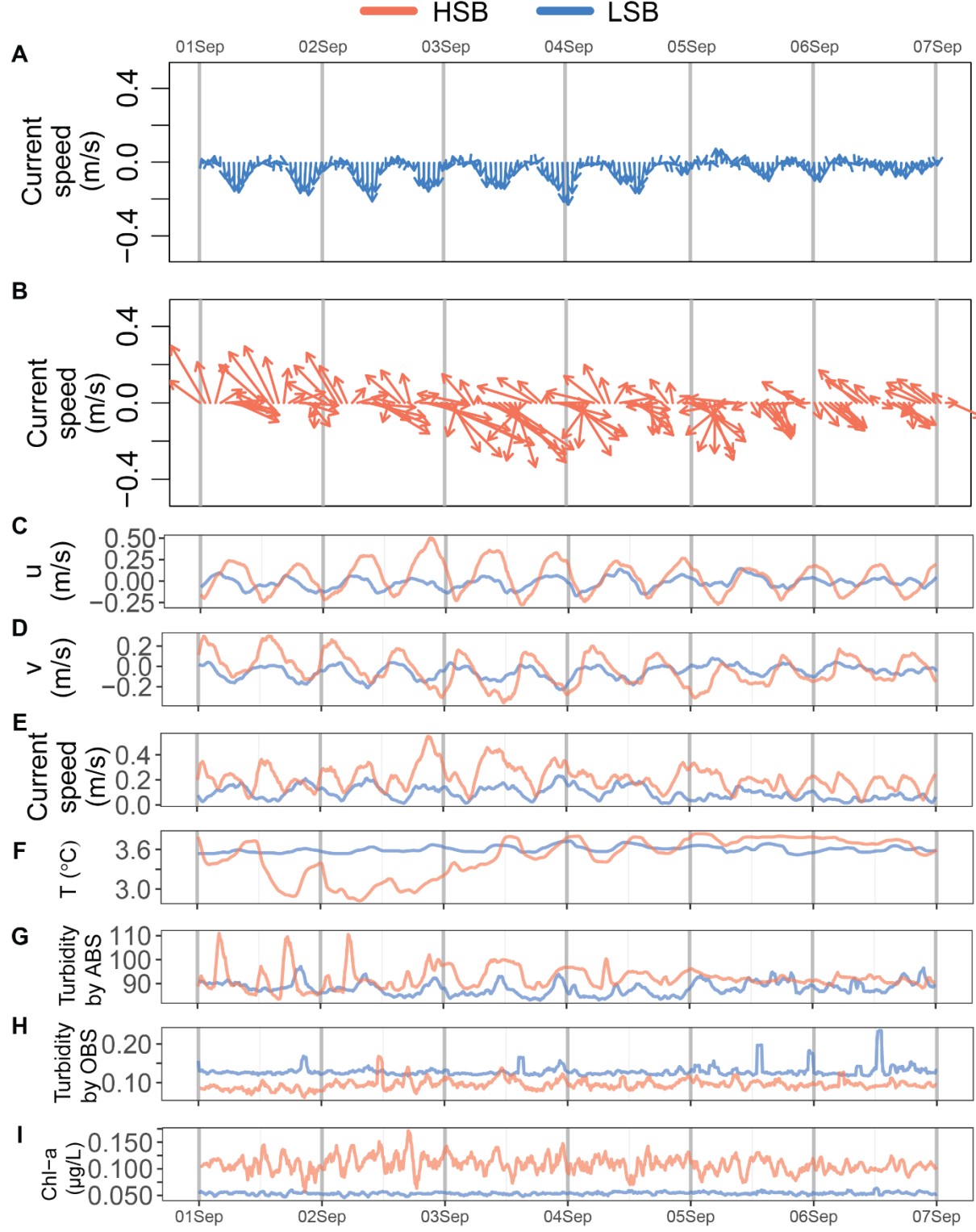

Figure 10: Expanded detail for the first week of September for the current direction at LSB (A), current direction at HSB (B), eastward velocity (C), northward velocity v (D), bottom current speed (E), temperature (F), turbidity by acoustic backscatter (ABS; G), turbidity (H), and chl-a concentration (I).

## 3.5 Mass deposition and organic carbon fluxes

The average mass fluxes were higher at HSB ($2.46 \pm 1.76$ g m$^{-2}$ day$^{-1}$) than at LSB ($1.43 \pm 0.93$ g m$^{-2}$ day$^{-1}$), with highest fluxes in winter (October to April) at both sites, which corresponds well with the superimposed seasonal patterns seen in ABS turbidity and bottom current speed. Average POC fluxes were higher at HSB ($3.07 \pm 1.91$ mmol C m$^{-2}$ d$^{-1}$) than at LSB ($1.91 \pm$

0.71 mmol C m$^{-2}$ d$^{-1}$). Organic carbon content at HSB was highest in autumn/summer months
(~2 %) and highest at LSB in autumn (2-4%; data not shown). Average C:N ratios were lower
at HSB (8.6 ± 3.2) than at LSB (10.8 ± 2.7) and were higher in winter and also in May 2018
(Figure 11C). The δ$^{13}$C ratios of trapped material were higher in winter at HSB compared to
LSB, and were higher in summer at LSB than at HSB (Figure 11D). The δ$^{15}$N of trapped
material was comparable between sites, although slightly higher at LSB. Winter δ$^{15}$N values
were highest compared to the rest of the year for both landers (Figure 11E). The lipid flux was
slightly higher at LSB, with low values in winter and peak values during the spring bloom
(Figure 11F). Unsaturated alcohols comprised the largest fraction of lipids at LSB, especially
in autumn and winter (Figure S12B). Peak lipid flux in April consisted of 25% polyunsaturated
fatty acids (PUFAs) at HSB (Figure S12C). Sterols made up the largest fraction of total lipids
at HSB and LSB in May (Figure S12D). The sterol fraction was lower in spring at both sites.
Swimmers were found in the sediment trap bottles, especially in the autumn months at LSB.
These consisted mostly of copepods (e.g., *Calanus* sp.), mysids (e.g., *Boreomysis* sp.),
amphipods (e.g., Eusiridae) and chaetognaths (i.e., arrow worms). Numbers of trapped
swimmers were lowest during winter at both sites. In addition, several large sponge spicules
were found in the bottles at HSB, but not at LSB.

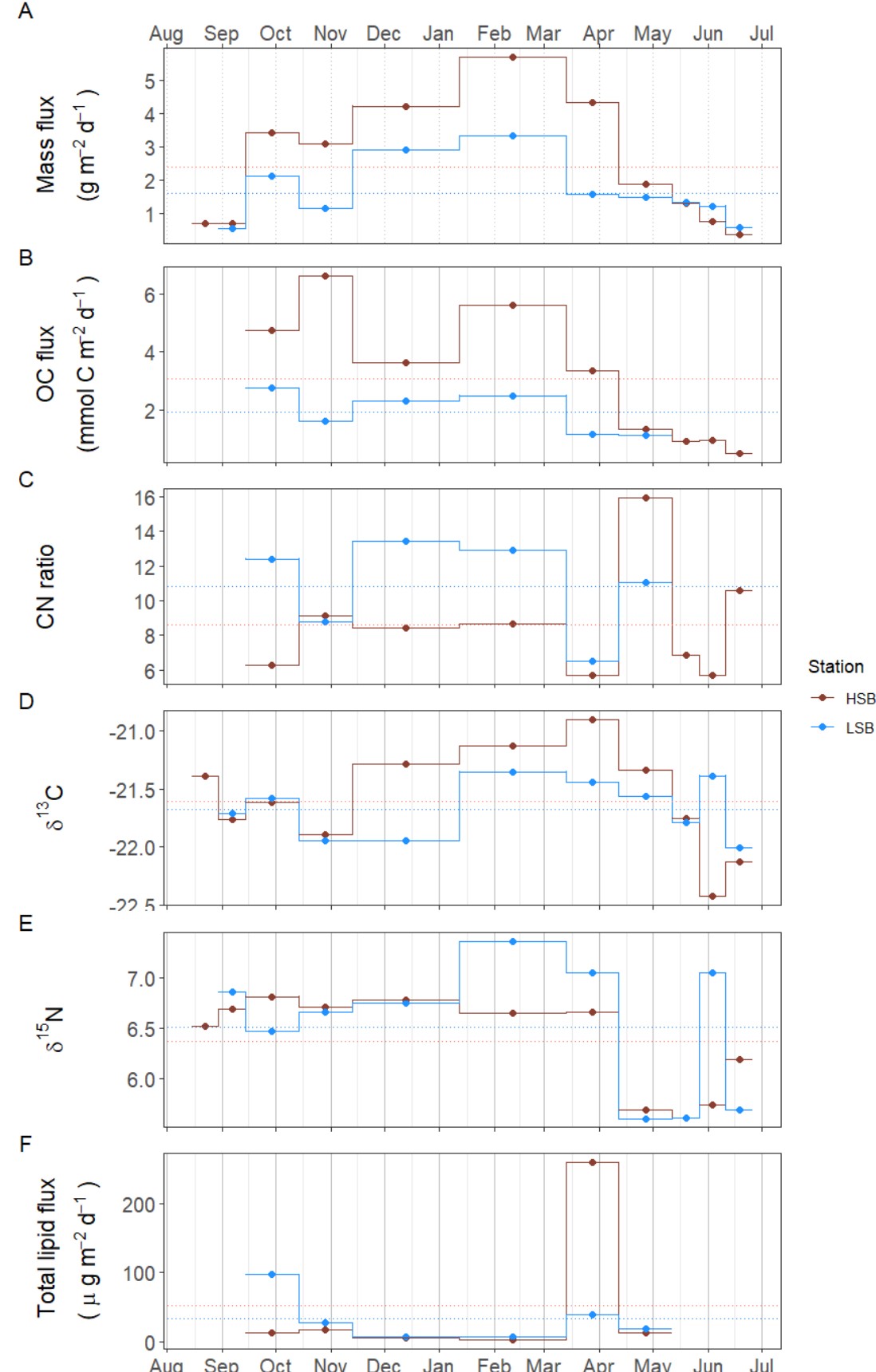


*Figure 11: Sediment trap content from the two benthic landers. HSB = high-sponge-biomass lander, LSB = low-sponge-*
*biomass lander. A) mass flux in g m⁻²d⁻¹, B) organic carbon flux in mmol C m⁻² d⁻¹, C) molar C:N ratio of trapped material,*
*D) δ¹³C of trapped material, E) δ¹⁵N of trapped material, F) total lipid flux in μg m⁻² d⁻¹.*

 ## 3.6   $\delta^{13}$C and $\delta^{15}$N isotopic ratios of benthic fauna and trapped material

The massive sponge *Geodia* spp. sampled at HSB showed a distinct isotopic signature
compared to the other benthic organisms, with a relatively enriched $\delta^{13}$C (-18.55 ± 0.17 ‰)
and a low $\delta^{15}$N (8.24 ± 0.16 ‰; Figure 12). The gorgonian coral *Primnoa resedaeformis* had
$\delta^{13}$C of -21.19 ± 0.59 ‰ and $\delta^{15}$N of 10.54 ± 0.33 ‰. Compared to *P. resedaeformis*, Decapoda
sp. showed slightly enriched $\delta^{13}$C (-20.48 ± 0.31 ‰), and $\delta^{15}$N (11.97 ± 0.43 ‰) values. The
glass sponge *Asconema* sp., sampled at HSB, also had relatively enriched isotopic values ($\delta^{13}$C:
-20.27 ± 0.36 ‰, and $\delta^{15}$N: 12.57 ± 0.31 ‰) while the sponge *Mycale* sp., sampled at LSB,
had a high $\delta^{15}$N isotopic ratio (13.05 ± 0.41 ‰), and a $\delta^{13}$C ratio of -19.47 ± 0.06 ‰. Sediment
trap samples had the lowest $\delta^{15}$N and $\delta^{13}$C isotopic ratios, with only small differences between
HSB and LSB (Figure 11 D & E; Figure 12).

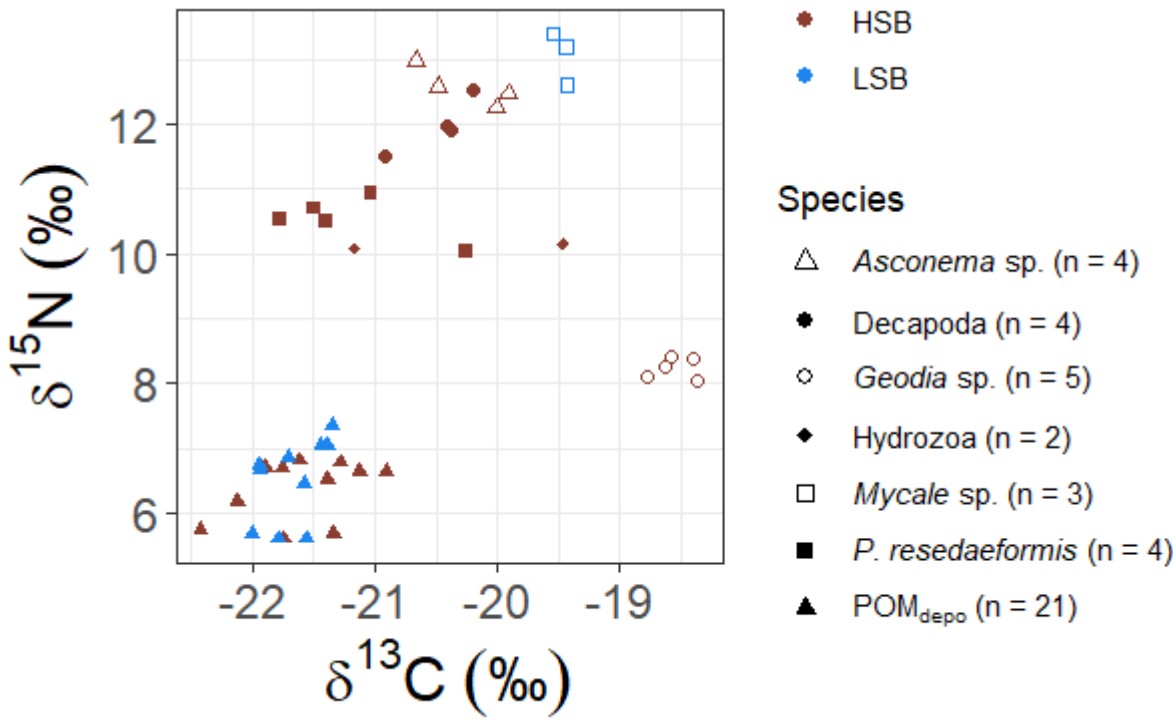

*Figure 12: Carbon and nitrogen stable isotopes plots of megafauna and sediment trap samples. HSB = high-sponge-*
*biomass, LSB = low-sponge-biomass.*

## 4   Discussion

Hydrodynamic- and environmental conditions were compared at two contrasting high- and
low-sponge-biomass sites along the northern Labrador shelf break. The aim was to compare
differences between the two sites in terms of (i) seawater properties and regional hydrography
(section 4.1, 4.2), (ii) bottom currents and environmental conditions, including seasonal
variations over the course of a year (section 4.3, 4.4), and (iii) benthic-pelagic coupling, organic
matter supply and isotopic signatures of benthic macrofauna (sections 4.6, 4.6, and 4.7).

## 4.1 Regional oceanography on the northern Labrador Shelf and Slope

The northern Labrador Shelf and Labrador Slope are known to be subject to strong tidal forcing
which causes vertical mixing, high bottom current speeds (Griffiths et al., 1981; Drinkwater
and Jones, 1987), and reduced stratification compared to the more northerly Baffin Island Shelf
(Lazier 1982; Sutcliffe et al. 1983; Drinkwater and Harding 2001). The results of our drifter
analysis confirm that around the HSB area three currents converge: the Hudson Strait Outflow,
the Baffin Intermediate Current, and the West Greenland Current (Figure 5A; Smith et al.,
1937; Yashayaev, 2007; Straneo and Saucier, 2008; Curry et al., 2011, 2014). These three
currents transport, respectively, Hudson Strait Outflow Water, Arctic Water and/or Baffin Bay
(intermediate) Water, and Irminger Water towards the northern Labrador Shelf and upper
slope. Our CTD transects show the characteristics of these water masses, and are similar to
earlier observations (Petrie et al., 1988; Fissel and Lemon, 1991; Drinkwater and Harding,
2001). The warmer and saltier water at HSB ($\Theta \sim 4.5$ ºC and S $\sim 34.9$) compared to LSB is
likely caused by Irminger Water (Figure 5 B & C), which follows the Labrador slope in
cyclonic direction beneath the cold water of the West Greenland Current and above the upper
slope (Lazier et al., 2002). Our findings concur with previous work which showed that Irminger
Water is gradually cooled while moving southward by mixing with the Baffin Island Current
(Cuny et al., 2002). However, the Argo float temperature profiles indicate that the area around
HSB might play an important role in transforming Irminger Water. For example, the 350-450
m depth layer in the HSB area regularly showed presence of Irminger Water (>4.5 ºC), while
it was only sporadically measured at LSB (Figure 5B). Irminger Water might therefore be
cooled and freshened in the area around HSB due to convergence and consequently mixing
occurs with the Hudson Outflow and Baffin Island Current. Our results support earlier findings
that identified a connection between the Hudson Strait outflow strength and the southern
Labrador Shelf water based on salinity measurements (Sutcliffe et al., 1983; Myers et al.,
575 1990).

## 4.2 Increased bottom nutrient concentrations

Both the LSB and HSB lander sites show higher nutrient concentrations in the bottom water
compared with the other shelf/deep CTD stations, and this difference was more pronounced at
the HSB lander location (Figure 4). Here we discuss two possible explanations for this
observation: large scale advection of nutrient-rich water from Baffin Bay and sediment efflux
of silicic acid. Intermediate water flows from Baffin Bay via the Davis Strait southward along
the continental slope (Curry et al., 2014). This water mass, referred to as Baffin Bay Water
(BBW), contains high nutrient concentrations (e.g., $41.6 \pm 25.5$ µM $Si(OH)_4$, $18.5 \pm 2.6$ µM
$NO_3^-$; Sherwood et al., 2021) due to *in situ* remineralization of organic matter to deep water
circulating in the Baffin Bay basin (Jones et al., 1984; Tremblay et al., 2002; Lehmann et al.,
2019). Furthermore, BBW shows relatively high concentrations of silicate and phosphate
compared to nitrate, due to denitrification at depth in Baffin Bay (Lehmann et al., 2019;
Sherwood et al., 2021). Secondly, high efflux of silicic acid (nutrients) from the sediment could
enhance bottom water silicate (nutrient) concentrations. Research on glass-sponge grounds on
the Scotian shelf has shown that the biogenic silica efflux from sediments lead to higher bottom
silicate concentrations (Maldonado et al., 2020a). This would also be possible for our study
area. Given that the silicate concentration was elevated by ~2-3 µM up to 100 meters above
the bottom (Figure 4), assuming that the length of the sponge ground was ~120 km (Figure 1),
and thereby estimating the retention time of a water parcel on the sponge grounds is about 33
days (length sponge ground divided by residual current speed), this would mean that, under the
assumption that the bottom 100m is well mixed, a sediment efflux of $6 - 9$ mmol Si $m^{-2}$ $d^{-1}$
would be required. While this would be a substantial sediment efflux, silicate effluxes of 2.4
mmol Si $m^{-2}$ $d^{-1}$ have been measured on the Scotian Shelf (Andrews and Hargrave, 1984;
Maldonado et al., 2020a), and of up to 14.1 mmol Si $m^{-2}$ $d^{-1}$ in the Laurentian Channel (East
Canada; similar depth and temperature; Miatta and Snelgrove, 2021). Nonetheless, the higher
silicate concentrations at HSB lander than at LSB lander imply that the source is located closer
to HSB. The fact that phosphate was also enhanced in bottom waters at HSB, suggests that
advection of nutrient-rich water from upstream is the more probable explanation. However,
further work on bottom silicate concentrations in relation to sponge grounds in this area is
needed to unravel the source of this excess silicate and investigate if and how sponge grounds
benefit from this.
The elevated nutrient concentrations could be beneficial for benthic organisms, specifically,
deep-sea sponges, which require silicic acid for spicule formation and skeletal growth (Whitney
et al., 2005; Maldonado et al., 2011, 2020b; López-Acosta et al., 2016). Published kinetic
uptake curves, describing silicic acid uptake rate *versus* concentration, suggests the
concentration at the HSB lander (13.6 µM) compared to LSB shelf (9.3 µM) lead to a higher
silicic acid uptake rates at the HSB site of 39% for *Axinella* spp. and 40% for *V. pourtalesii*
(Maldonado et al., 2011, 2020b). Furthermore, elevated silicic acid concentrations on a spatial
scale of kilometres are thought to allow the persistence of sponge grounds and build-up of
(glass) sponge biomass over long timescales (Whitney et al., 2005; Maldonado et al., 2020a).

## 4.3 Tidal dynamics and bottom current speed

This study, to our knowledge, is the first to report year-long hydrodynamic- and environmental conditions measured simultaneously at a high- and low-sponge-biomass ground. Our measurements show high bottom currents at both sites with distinct differences in tidal dynamics. While semidiurnal tidal harmonics predominate at both sites, tidally driven horizontal current speeds were around five times higher at HSB than at LSB. At the HSB site, barotropic and near-bottom M2 tidal currents are oriented across-slope, but the near-bottom M2 tidal ellipse is smaller in magnitude and strongly indicating enhanced local near-bottom energy dissipation of the barotropic tide through tide-topography interaction (Table 2; Figure 8). At the LSB site, near-bottom M2 and S2 tidal ellipses from the ADCP are oriented along-slope with a small across-slope component. In contrast, modelled barotropic semi-diurnal tidal harmonics were of similar magnitude, but mainly oriented across- interaction (Table 2; Figure 8). This discrepancy is likely due to local changes in bathymetry (Figure S1), which are not resolved in the OTIS tidal model. The outcome of strongly enhanced current speeds at the HSB site is contrary to White (2003) who measured high current speeds in areas where no sponges were recorded, and vice versa, at the Porcupine Sea Bight. Caution should be applied comparing these areas, as the sponge fields in the Porcupine Sea Bight mostly consist of glass sponges, and here we see a mixture of glass sponges and massive demosponges. Bottom current speeds are higher at HSB than at LSB (Table 1), but bottom currents at LSB are still comparable with current speeds found at other sponge grounds on the Scotian Shelf (mean: 0.12 m s$^{-1}$; Hanz et al., 2021a) and on the Arctic mid-Atlantic ridge (mean: 0.14 m s$^{-1}$; Hanz et al., 2021b). The conversion of kinetic energy from barotropic to baroclinic tides and to turbulence over rough topography shapes the distribution of benthic filter feeding communities in many areas throughout the global ocean (van der Kaaden et al., 2024). At the northern Labrador shelf break, larger aggregations of sponges are mainly found on topographic slopes, where near-critical and super-critical reflection of internal waves are predicted (Figure 13).

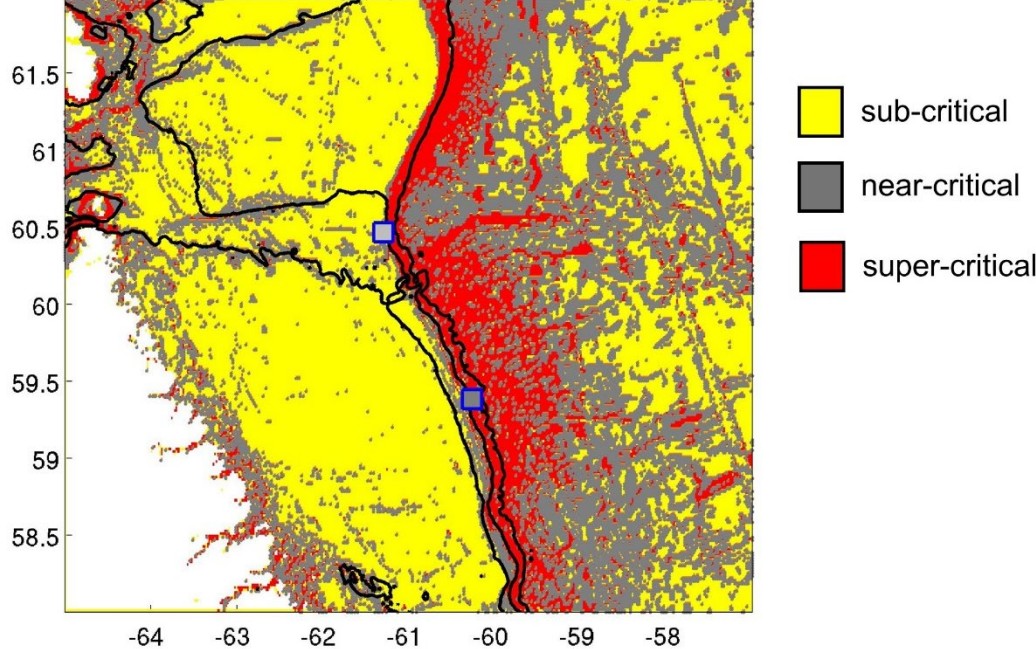


*Figure 13: The internal wave slope parameter indicates sub-critical conditions across most of the Labrador Shelf and in the*
*deep Northwest Atlantic. Near-critical and super-critical conditions are primarily observed along the continental margin.*
*This analysis suggests that the HSB lander (northern point) was situated in near-critical conditions for the M2*
*tide, while the LSB lander (southern point) experienced supercritical bottom slopes for M2.*

## 4.4 How can strong bottom currents benefit the benthic community?

Strong tidally-induced bottom currents can benefit the benthic community at the HSB site in
various ways. First, passive suspension feeders as the gorgonian *P. resedaeformis* benefit from
high horizontal currents through an increased particulate organic matter flux (Shimeta and
Jumars, 1991) and sponges (specifically glass sponges) could benefit from an increased water
flow rate through their body plan (Vogel, 1977; Leys et al., 2011), thereby increasing food
availability. Second, resuspension caused by oscillating tidal bottom currents enhance organic
matter and inorganic nutrient availability in the benthic boundary layer and enhance food
supply to the sponges (Roberts et al., 2018). In this study, high along-slope bottom currents at
both sites were associated with increased turbidity (both ABS and OBS), indicative of
resuspension. However, the beneficial effect of resuspension for sponge biomass is not yet fully
understood, as reoccurring strong turbidity flows (at LSB) could also prevent high sponge
biomass from developing by smothering young sponges when particles settle out (Klitgaard
and Tendal, 2004).
The substrate at HSB consisted mostly of pebbles, cobbles, and boulders (Dinn et al., 2020)
and a qualitative assessment of the sediment type at LSB suggested the dominance of muddy
soft sediment (Coté et al., 2019; J. Vad, *pers. com.*). As higher bottom currents would increase

bed shear stress and thereby enhance resuspension (Lesht, 1979; Jones et al., 1998), we argue that fine material is resuspended at HSB before its accumulation on the seafloor. This increases availability of organic matter to benthic suspension feeders in the benthic boundary layer and prevent smothering from sedimentation. Resuspension has also been linked to high sponge biomass (Davison et al., 2019), as potential food sources such as organic matter and bacteria can bind to suspended particles in the water column. The interaction of high bottom currents with rough topography causes turbulence and mixing of bottom waters (Witte et al., 1997; Leys et al., 2011; Culwick et al., 2020). As the substrate is likely rougher and bottom currents are higher at HSB than at LSB, the bottom water probably experiences more intense mixing and turbulence at HSB. Finally, periodic supply of fresh phytoplankton derived material during the spring bloom (Figure S10, Figure S11) increases the food availability of passive suspension feeders living on the sponge grounds. In short, the stronger tidal currents at HSB enhance bottom water mixing which replenishes oxygen, dissolved organic matter, POM, and (inorganic) nutrients in the benthic boundary layer, and thereby increases food supply to benthic fauna (Davison et al., 2019; Hanz et al., 2021b, 2021a).

## 4.5 Surface productivity and benthic-pelagic coupling

The Hudson Strait outflow water is known to increase nutrient concentrations in the surface waters on the northern Labrador Shelf (Kollmeyer et al., 1967; Sutcliffe et al., 1983; Drinkwater and Harding, 2001). The increased nutrient supply supports high primary productivity in an area extending from the Hudson Strait to the southern Labrador Shelf, bounded by the thermal front associated with the 1,000 m isobath (Frajka-Williams et al., 2009; Frajka-Williams and Rhines, 2010; Cyr and Larouche, 2015). Previous studies show that surface chl-a concentrations are comparable between the two sponge grounds (see Figure 2A in Frajka-Williams and Rhines, 2010), suggesting that differences in surface productivity alone are insufficient to explain the differences sponge biomass between regions. Furthermore, studies elsewhere in the Canadian Arctic have shown that benthic biomass is explained not only by surface productivity but also by local hydrodynamics and benthic-pelagic coupling (Thomson, 1982; Grebmeier and Barry, 1991; Roy et al., 2014).

Our year-long recordings of bottom water chl-a concentrations provide evidence for strong benthic-pelagic coupling during spring in this region. The benthic landers showed early arrival of fresh phytodetritus in early March, a peak in chl-a mid April, and chl-a concentration was close to background values again from early May at LSB and from mid May at HSB (Figure 9C). Studies on the onset of the phytoplankton bloom on the Labrador shelf show that blooms usually initiate around mid April and peak around mid June (Fuentes-Yaco et al., 2007; Frajka-

Williams and Rhines, 2010; Cyr et al., 2023). The study of Cyr et al. (2023) estimates that the
standard deviation in timing of the initiation of the phytoplankton bloom is around 21 days. As
environmental conditions of the Northern Labrador shelf were close to average during 2019
(Cyr and Galbraith, 2021), we think its acceptable to assume phytoplankton bloom timing was
similar to values found in literature. Therefore, arrival of phytodetritus at our benthic landers
was then three months earlier to normal phytoplankton bloom timing. Earlier research has
shown that chl-a starts to increase on the northern Labrador shelf from early March onwards
(Harrison et al., 2013). During this time the water column is still relatively cold and poorly
stratified, allowing for relatively high export of phytoplankton to the seafloor. Additionally,
the short periods of low ice-cover mid February and mid March (Figure S11D) match the
subsequent increase in bottom chl-a concentration seen for both landers (Figure 9C). The onset
of the phytoplankton bloom for the northern Labrador shelf is around mid April, and related to
the onset of stratification (Cyr et al., 2023) and sea-ice cover (Wu et al., 2007). The timing of
peak bottom chl-a concentrations (mid April) and consequential decline compare well with the
timing phytoplankton bloom initiation proposed by Cyr et al. (2023). They show there is a
south-to-north progression of the phytoplankton bloom over the Labrador shelf, which matches
with our data that shows chl-a concentrations stay elevated around three weeks longer at the
more northern HSB lander. Furthermore, assuming surface chl-a concentration peaks in June,
we can infer that there appears to be a decoupling between pelagic productivity and bottom
chl-a concentration in summer, likely due to enhanced stratification and intense zooplankton
grazing (Rivkin et al., 1996; Turner, 2015).
Our findings suggest strong benthic-pelagic coupling started weeks before the peak of the
phytoplankton bloom, supplying fresh fluorescent material to the seafloor in spring for a period
of weeks to months. Since the timing of phytoplankton bloom for high-latitude seas is shifting
to earlier in the year due to rising temperatures and earlier sea-ice retreat (Edwards and
Richardson, 2004; Wu et al., 2007; Hunter-Cevera et al., 2016), and since deep-sea sessile
organisms, such as cold-water corals and deep-sea sponges demonstrate seasonality in their
phenology (Leys and Lauzon, 1998; Maldonado, 2011; Maier et al., 2020a), the early arrival
of phytoplankton-derived material could have consequences for their overall fitness and
survival. Nevertheless, the effect of a shift in spring bloom timing for benthic suspension
feeders, including deep-sea sponges, remains unknown.
Recent ABS measurements reveal a layer of increased 300 kHz backscatter along the northern
Labrador Shelf, indictive of high abundance of micronekton and macrozooplankton
(Chawarski et al., 2022). Earlier studies showed a high zooplankton biomass on the
Newfoundland Shelf from July onwards (Head et al., 2003, 2013). In our traps the highest flux
of unsaturated alcohols, a biomarker for zooplankton (specifically copepods; Dalsgaard et al.,
2003), and the highest numbers of swimmers were in summer and autumn. During the spring
bloom, trapped material at LSB had the highest relative amount of unsaturated alcohols while
at HSB the level of PUFAs, markers for phytoplankton derived-material, was highest
(Dalsgaard et al., 2003). Furthermore, our observations suggest that the number of trapped
swimmers was higher at LSB than at HSB. These results are consistent with the hypothesis that
zooplankton biomass is high over the northern Labrador Shelf (Saglek Bank) and that
zooplankton is transported by the southerly current along the Labrador Shelf together with the
high phytoplankton biomass plume (Sutcliffe et al., 1983; Drinkwater and Harding, 2001).
Overall, there was a larger fraction of zooplankton marker lipids in trapped material at LSB,
which implies that zooplankton play a more important role in benthic-pelagic coupling at LSB
than at HSB.

## 4.6 Organic matter fluxes to the seafloor

Organic matter deposition was higher at the HSB lander than at the LSB lander. Overall,
deposition was highest during the winter months and consisted of more degraded material than
during summer, indicated by high C:N ratios and high $\delta^{15}N$ values. This increased deposition
in winter is likely resuspended material as shown by peaks in ABS turbidity in the bottom
boundary layer and relate to higher current speeds. The C:N ratio of deposited matter was
higher at LSB (~13) compared to HSB (~8), indicating the material was more degraded at LSB.
Hanz et al. (2021a, 2021b) also found higher mass and carbon fluxes during winter months and
low carbon fluxes when the spring/summer phytoplankton bloom arrived. They attributed this
to the presence of more degraded and resuspended material in winter. Data concerning mass
fluxes from sponge grounds remain scarce, but the fluxes measured here (HSB 2.46 ± 1.76 g
$m^{-2}$ day$^{-1}$, LSB: 1.43 ± 0.93 g $m^{-2}$ day$^{-1}$) were comparable to those of a *Vazella pourtalesii*
sponge ground on the Scotian Shelf (3.17 ± 3.42 g $m^{-2}$ day$^{-1}$; Hanz et al., 2021a) but
substantially higher than those of a sponge ground on the Arctic mid-Atlantic ridge (0.03 –
0.30 g $m^{-2}$ day$^{-1}$; Hanz et al., 2021b). Overall, our data suggest organic matter deposition fluxes
are higher at HSB compared to LSB, and that the organic matter is of higher quality. The
organic carbon fluxes (HSB: 3.07 ± 1.91 mmol C $m^{-2}$ d$^{-1}$; LSB: 1.91 ± 0.71 mmol C $m^{-2}$ d$^{-1}$)
reported in our study are considerably lower than those of a more shallow (150 – 250 m depth)
*V. pourtalesii* sponge ground on the Scotian Shelf (8.3 mmol C $m^{-2}$ d$^{-1}$; Hanz et al., 2021a), but
high compared to an Arctic mid-Atlantic ridge sponge ground (peak of 1.6 mmol C $m^{-2}$ d$^{-1}$;
Hanz et al., 2021b). The higher organic matter deposition rate and relative fresher material at
HSB compared to LSB are likely related to its shallower position on the shelf and the more
dynamic water column.

 ## 4.7 Isotopic signatures of benthic macrofauna at two contrasting sponge grounds

Although the sample size was limited, the stable isotope data revealed interesting patterns of
organic matter utilization by the benthic community. The gorgonian coral *P. resedaeformis* is
found one trophic level (Fry, 2006) above the sediment trap material and therefore likely feeds
on sinking organic matter, confirming previous observations (Sherwood et al., 2005, 2008).
Sponges can generally be classified into two groups based on their associated microbial fauna,
those with high microbial abundance (HMA) or those with low microbial abundance (LMA;
Vacelet and Donadey, 1977). *Geodia* spp. can occur in high abundance and biomass on sponge
grounds (Kutti et al., 2013). These sponges are considered HMA (Radax et al., 2012) and feed
mostly on dissolved organic matter with additional particulate sources such as bacterioplankton
(Bart et al., 2021). Many hexactinellidae that can form sponge grounds, for instance *Vazella*
*pourtalessii* and *Aphrocallistes vastus*, are considered LMA sponges and feed mostly on
bacterioplankton (Kahn et al., 2015). The high $\delta^{15}N$ isotopic ratios for the sponges *Asconema*
spp. ($12.6 \pm 0.3$ ‰ $\delta^{15}N$) and *Mycale* spp. ($13.1 \pm 0.4$ ‰ $\delta^{15}N$), have been observed previously
for LMA sponges (Iken et al., 2001; Polunin;, 2001; Kahn et al., 2018). Deep-sea LMA sponges
typically have elevated $\delta^{15}N$ values in the benthic food web (Kahn et al., 2018), a phenomenon
that is still poorly understood. Possible explanations could be selective feeding on [15]N enriched
bacteria (Wilkinson et al., 1984), feeding on resuspended benthic bacteria (Kahn et al., 2018),
or nitrogen (re)cycling within the sponge holobiont (Rooks et al., 2020; Hanz et al., 2022).
Interestingly, the HMA massive sponge *Geodia* sp. has distinct $\delta^{13}C$ *and* $\delta^{15}N$ values, which
was also observed in Hanz et al. (2022), indicating different feeding or metabolic strategies.
Recent research on *Geodia baretti* has indeed demonstrated that these sponges rely in large
part on DOM for their metabolic requirements (Bart et al., 2021; de Kluijver et al., 2021). In
this study, *Geodia* spp. ($8.2 \pm 0.2$ ‰ $\delta^{15}N$) was one trophic level higher than oceanic DOM
$\delta^{15}N$ (~5 ‰; Benner et al., 2005; Sigman et al., 2009) and $\delta^{15}N$-$NO_3^-$ (~5‰; Sigman et al.,
2009; Sherwood et al., 2021), limiting our ability to distinguish between DOM and $NO_3^-$ (by
i.e., denitrification; Hoffmann et al., 2009) as potential nitrogen sources. The $\delta^{13}C$ value of
*Geodia* spp. ($-18.4 \pm 0.17$ ‰ $\delta^{13}C$) is $\pm 3.5$‰ higher than bottom water $\delta^{13}C$-DOC values on
the Labrador Shelf (Barber et al., 2017), i.e. more than four times higher than the expected
0.8‰ $\delta^{13}C$ step per trophic level (Vander Zanden and Rasmussen, 2001). Alternatively, *Geodia*
spp. could capitalize on DIC via their symbionts (de Kluijver et al., 2021), as recently observed
in Arctic *Geodia* spp. assemblages (Morganti et al., 2022) and other deep-sea sponges (van
Duyl et al., 2020). Even limited chemoautotrophic assimilation of high $\delta^{13}C$-DIC (~0 ‰ $\delta^{13}C$)
could explain the high $\delta^{13}C$ values of *Geodia* spp. These results indicate that passive suspension
feeders benefit from high tidal currents through an increased particulate organic matter flux
(Shimeta and Jumars, 1991), whereas sponges likely benefit from replenishment of nutrients,
oxygen, and dissolved organic matter (Schläppy et al., 2010).

## 5  Conclusion

The aim this research was to obtain a better understanding of the environmental conditions in
which sponge grounds occur and investigate the conditions in which high-sponge-biomass
could develop. This study identified that the high-biomass sponge ground on the northern
Labrador Shelf differ from the low-biomass sponge ground in the following ways: a more
dynamic water column with strong tidal bottom currents and near-bottom energy dissipation
by tide-topography interactions, increased bottom inorganic nutrient concentrations, and higher
organic matter flux to the seafloor. Furthermore, both sponge grounds experienced strong
benthic-pelagic coupling during spring and a decoupling during summer months. The elevated
bottom nutrient concentrations at the high-sponge-biomass ground could be related to large
scale circulation or sediment effluxes, and future work is needed to asses this. Our findings
suggest a relation between slope-criticality and sponge biomass on the northern Labrador Shelf
which could be interesting to investigate in future work. The deep-sea sponges and corals
benefit from the dynamic water column in the high-biomass sponge ground by increased
availability of food sources and nutrients.

## 6  Funding statement

This research was supported by the European Union's Horizon 2020 Research and Innovation
Programme under grant agreement nos. 678760 (ATLAS) and 818123 (iAtlantic). This output
reflects only the authors' view, and the European Union cannot be held responsible for any use
that may be made of the information contained therein. Department of Fisheries and Oceans
contributions were funded through the departmental International Governance Strategy
programme awarded to EK. DvO was supported by the Innovational Research Incentives
Scheme of the Netherlands Organisation for Scientific Research (NWO), respectively, under
grant agreement 864.13.007. EdF was partly supported by ArcticNet Network of Centres of
Excellence, Glacier troughs as biodiversity and abundance hotspots in Arctic and subarctic
regions project, ArcticNet Phase V (Geoffroy et al.). The data presented herein were collected
by the Canadian research icebreaker CCGS Amundsen and made available by the Amundsen
Science program, which was supported by the Canada Foundation for Innovation and Natural
Sciences and Engineering Research Council of Canada. The views expressed in this publication
do not necessarily represent the views of Amundsen Science or that of its partners. Ship-time
on the CCGS Amundsen was also funded by an NSERC ship-time grant (Edinger et al., grant
nr.: RGPST-515528-2018), ArcticNet Network of Centers of Excellence Canada, and the
Department of Fisheries and Oceans Canada (DFO; Coté et al.). The funders had no role in
study design, data collection, and analysis, decision to publish, or preparation of the
manuscript.

## 7  Author statement

EDF: sample analysis, data analysis, and writing; IY: data collection, data analysis, and writing.
CM: conceptualization, data analysis and writing; JV: data collection and data analysis; FM:
conceptualization, sample analysis and data analysis; GD: conceptualization, data analysis;
EK, EH, IY, SWR, MR: conceptualization and site selection; SWR, MR, EK, BM, GT: site
contribution and preparation of benthic landers; GW: conceptualization, sample analysis, data
analysis, and writing; SB: data collection and sample analysis; DvO: conceptualization, data
analysis, writing. All authors contributed to the article and approved the submitted version.

## 8  Acknowledgements

We would like to thank the skillful crew and technicians on board CCGS Amundsen for their
support during the fieldwork. Specifically, we thank Dr. Paul Snelgrove (Memorial University
of Newfoundland), Dr. David Cote (DFO) and Shawn Meredyk (Amundsen Science) for their
assistance in facilitating our field programme. Cam Lirette (DFO) assisted in preparing various
data layers to assist in site selection. We would also like to thank Jan Peene for nutrient
analysis, Peter van Breugel and Jurian Brasser for help in measuring
macrofauna/POM/sediment trap stable isotopes, and Pascal Guillot for quality assurance of the
CTD profiles. Finally, we thank Kevin MacIsaac and Marc Ringuette for their help in
identifying the sediment trap swimmers.

## 9  Data availability

Raw data will be stored on zenodo and URL will be provided upon acceptance of the
manuscript.

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

# 11 Supplementary material

## 11.1 Tables

*Table S1: Overview of lander deployment locations and CTD cast locations.*

| station | instrument | date/period | latitude | longitude | depth |
|---|---|---|---|---|---|
| HSB_bl | benthic_lander | 27-7-2018 to 2-7-2019 | 60.47 | -61.29 | 410 |
| LSB_bl | benthic_lander | 27-7-2018 to 1-7-2019 | 59.38 | -60.28 | 558 |
| HSB_ctd1 | CTD | 2018-08-03 07:37:08 | 60.47 | -59.26 | 2428 |
| HSB_ctd2 | CTD | 2018-08-02 17:21:58 | 60.47 | -60.38 | 1877 |
| HSB_ctd3 | CTD | 2018-07-30 15:27:05 | 60.47 | -61.30 | 391 |
| HSB_ctd4 | CTD | 2018-07-30 07:31:07 | 60.46 | -62.12 | 359 |
| HSB_ctd5 | CTD | 2018-07-27 19:41:58 | 60.40 | -62.90 | 289 |
| LSB_ctd1 | CTD | 2018-07-29 04:30:19 | 59.53 | -58.64 | 2563 |
| LSB_ctd2 | CTD | 2018-07-28 23:25:52 | 59.48 | -59.45 | 1938 |
| LSB_ctd3 | CTD | 2018-07-28 09:52:11 | 59.38 | -60.27 | 608 |
| LSB_ctd4 | CTD | 2018-07-28 06:12:07 | 59.31 | -61.02 | 192 |
| LSB_ctd5 | CTD | 2018-07-28 03:10:24 | 59.22 | -61.83 | 138 |

*Table S3: Overview of rock dredge transects. HSB = high-sponge-biomass site, LSB = low-sponge-biomass site, (Coté et al., 2019).*

| Station Name | Start Lat | Start Long | End Lat | End Long | Logged bottom depth (m) | Time at bottom (min) | Length of cable out (m) | Max vessel speed (knots) | Comments |
|---|---|---|---|---|---|---|---|---|---|
| LSB_rd | 59.38 | -60.27 | 59.37 | -60.29 | 552 | 10 | 1500 | 1 | NA |
| HSB_rd | 60.47 | -61.28 | 60.48 | -61.30 | 404 | 20 | 507 | 2 | Small catch |

    ## 11.2  Figures

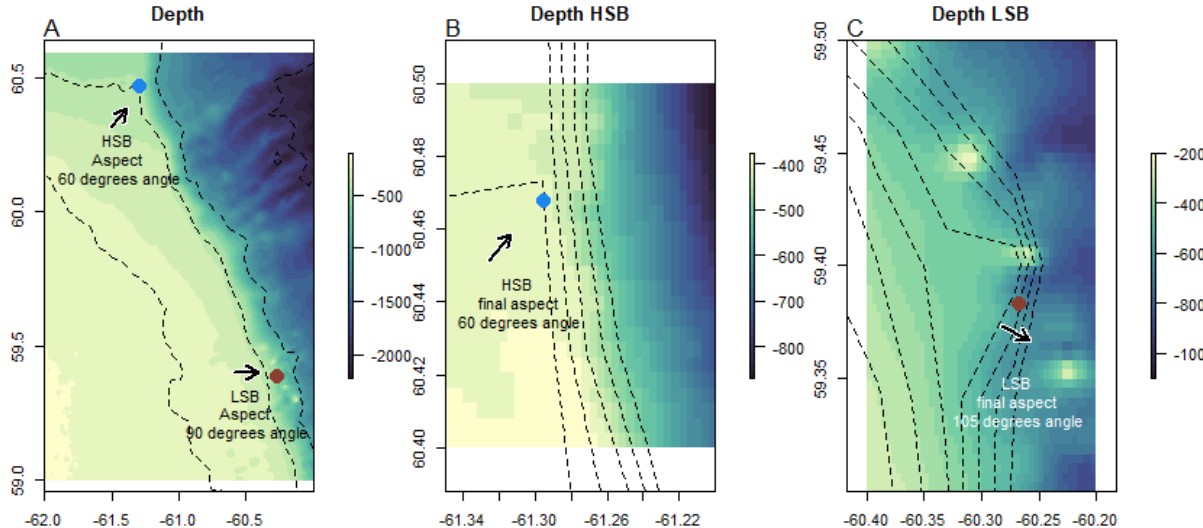

*Figure S1: slope direction or aspect estimation for HSB and LSB. A) map of study area with estimated slope aspects of 60°
and 90° angle for HSB and LSB, respectively. Contour lines at 200, 400, and 1000 metre is shown. B) expanded detail on
HSB shows angle of 60° is a good estimate. Contour lines at 400, 425, 475, 500 are shown. C) expanded detail on LSB site
shows angle of 105° is better estimate. Contour lines at 450, 475, 500, 525, 550, 575, 600 metre depth are shown. Note the
different colour scales for depth. Locations of lander is indicated by coloured dots, with HSB = blue, and LSB = brown/red.*

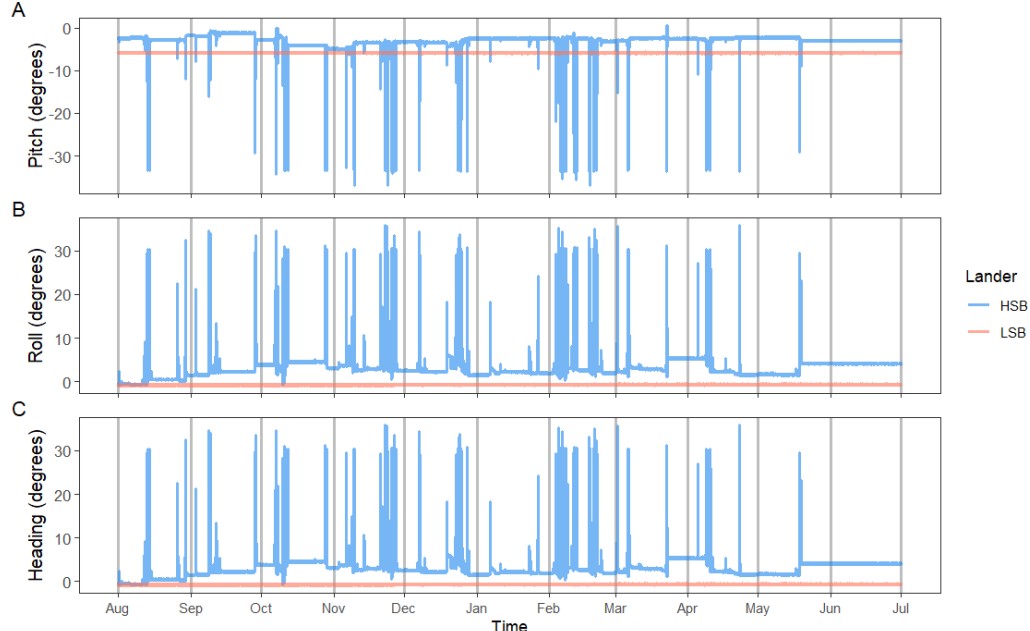

*Figure S2: Pitch (A), Roll (B), and Heading (C) data of the ADCPs from  both benthic landers.*

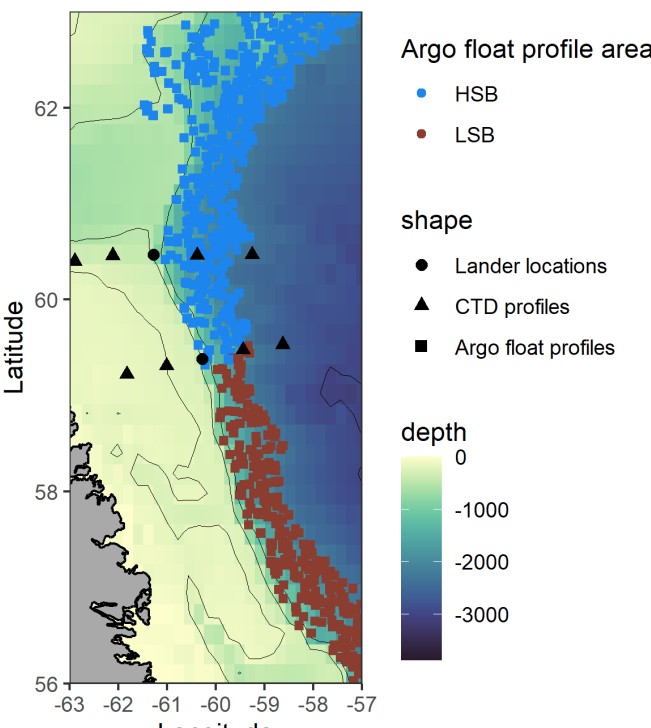


*Figure S3: Locations of Argo float profiles used for assessing the regional oceanography. Coloured squares indicate Argo*
*float profiles, and black triangles/dots the location of CTD profiles/benthic lander location.*

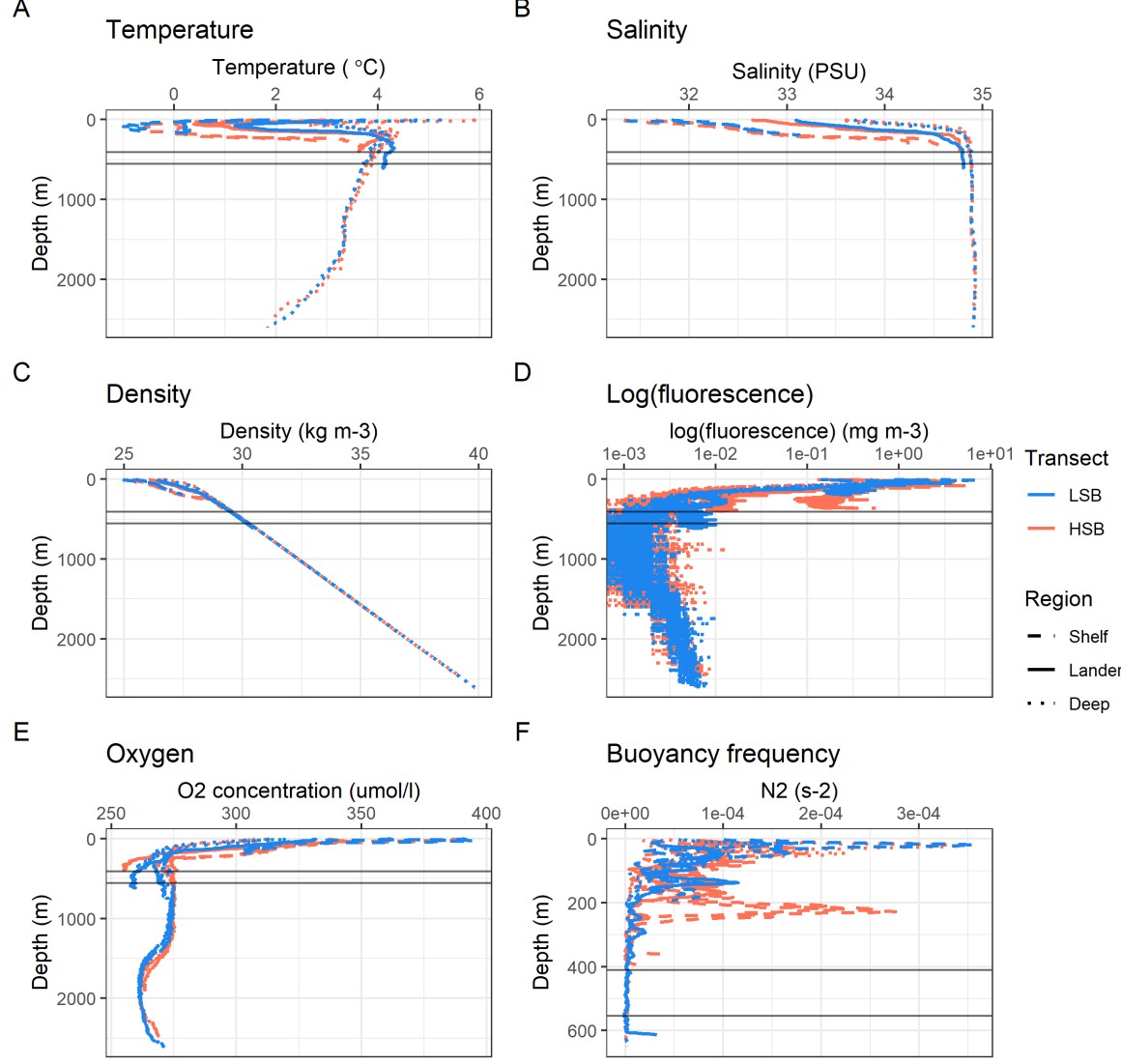


*Figure S4: CTD profiles with temperature (A), salinity (B), density (C), Fluorescence (D), Oxygen (E), Buoyancy frequency*
*(F). LSB = Low-sponge-biomass transect, HSB = High-sponge-biomass transect. Buoyancy frequency is smoothed over 15*
*m for visibility, and the plot only shows top 650 m of the water column, as deeper waters have values close to zero.*

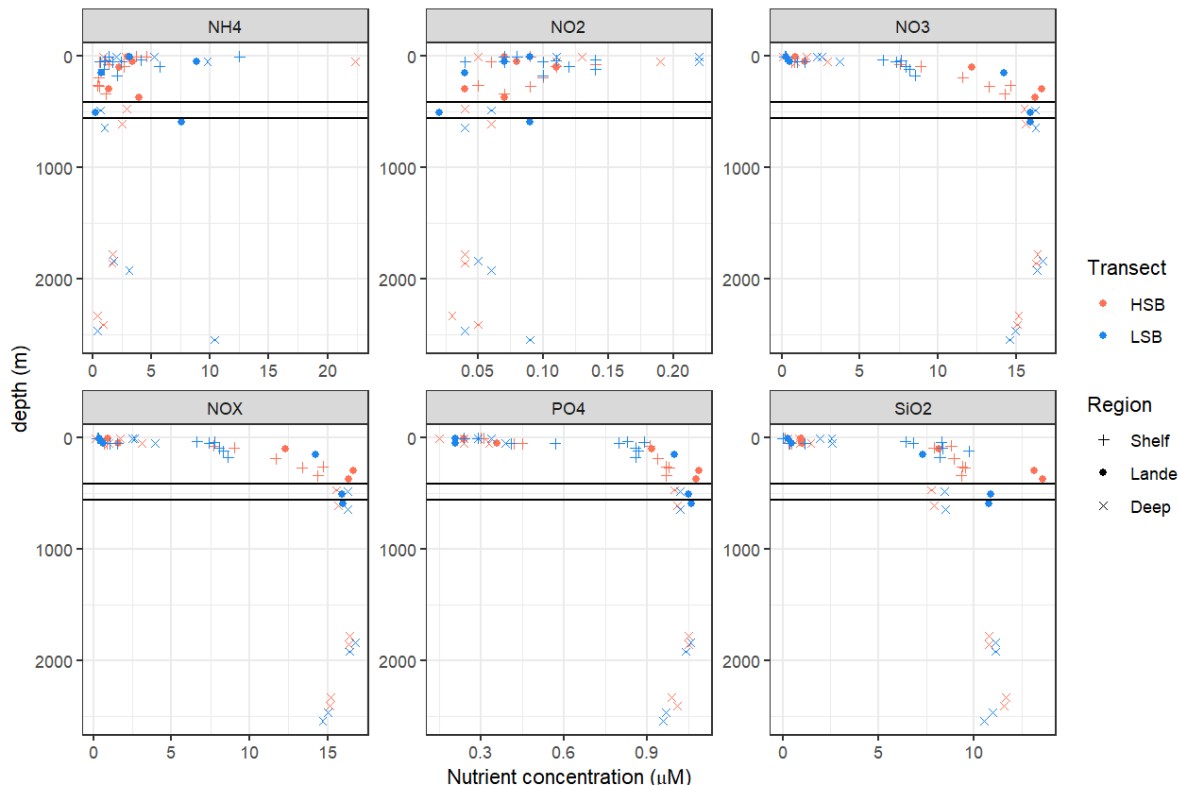


*Figure S5: nutrient profiles for the two transects over the complete depth. HSB = high-sponge-biomass, LSB = low-sponge-biomass.*

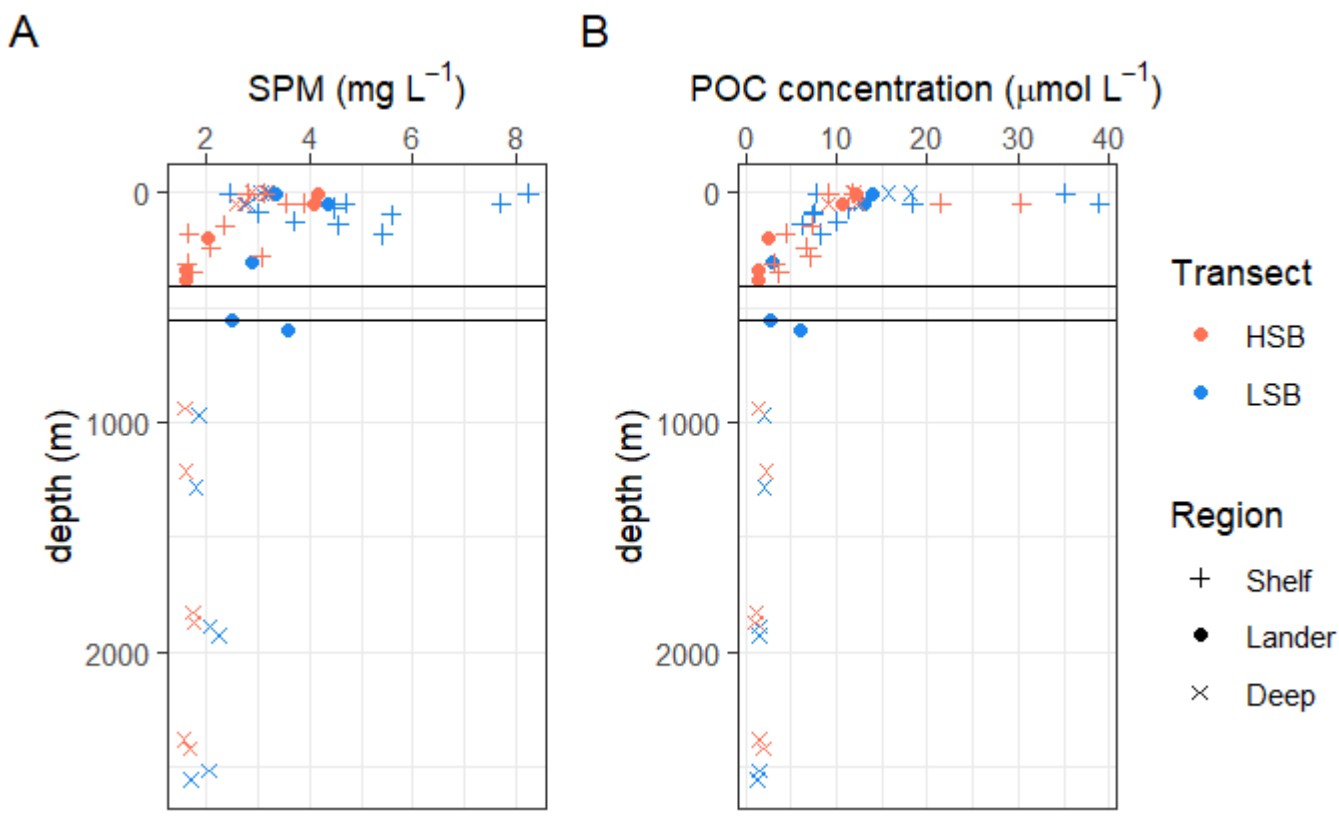


*Figure S6: A) Suspended particulate matter (SPM) concentration and B) particulate organic carbon concentration of the CTD the two transects. HSB = high-sponge-biomass transect, LSB = low-sponge-biomass transect. The horizontal lines resemble depth of benthic landers, where the top line is the HSB lander depth, and lowest line resembles LSB lander depth.*

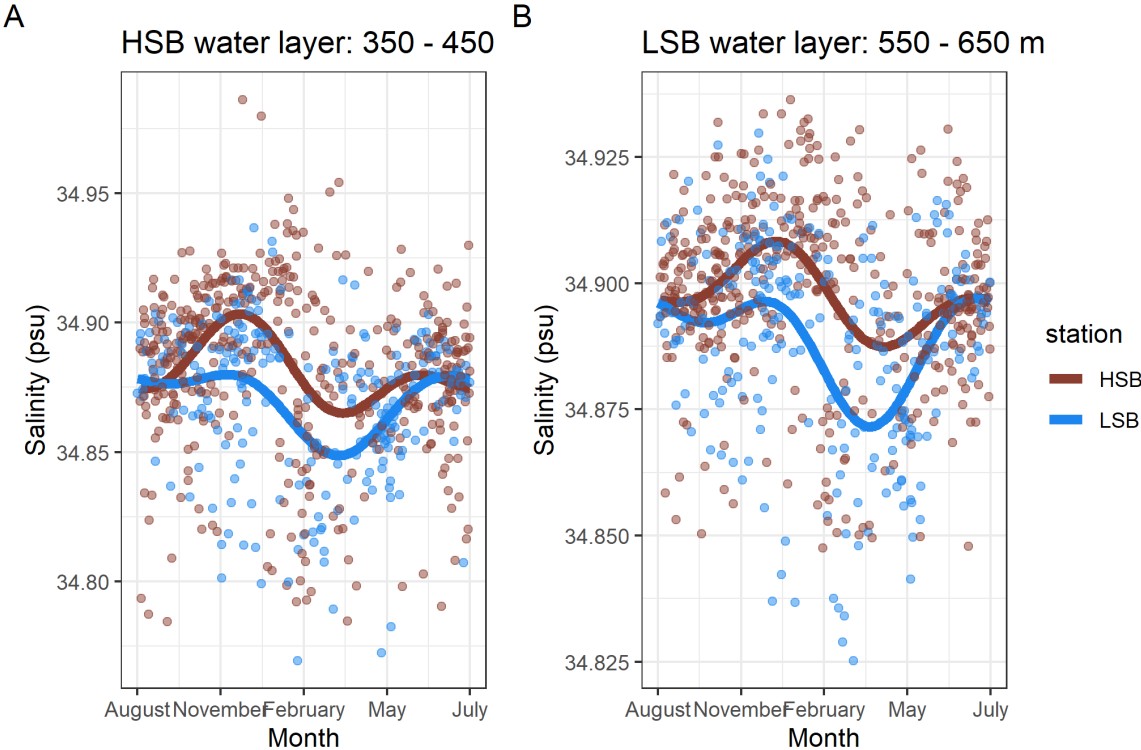


*Figure S7: A) seasonal salinity, from Argo float data, of the water layer in which HSB lander is located. B) seasonal salinity*
*of the water layer in which LSB is located.*

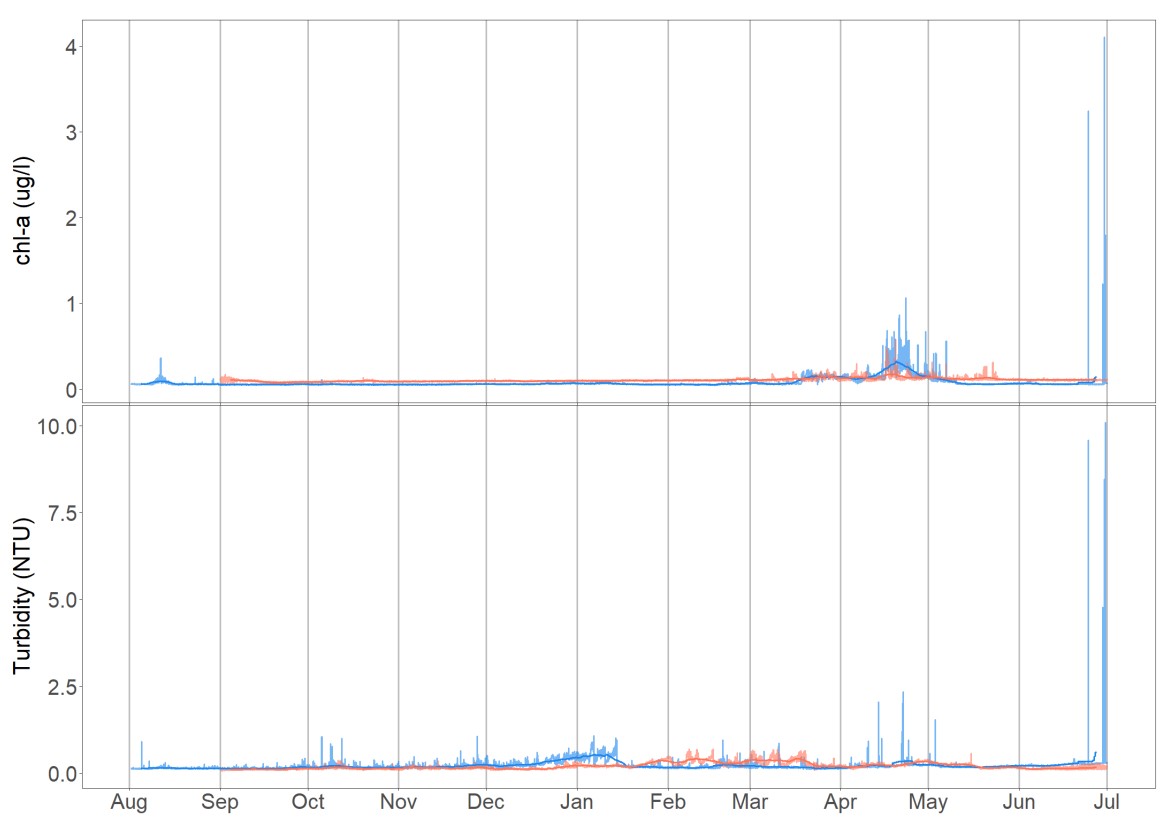


*Figure S8: Chlorophyll-a and turbidity data without cutting the y-axis at 1.25 µg L⁻¹, and 2.5 NTU, respectively.*


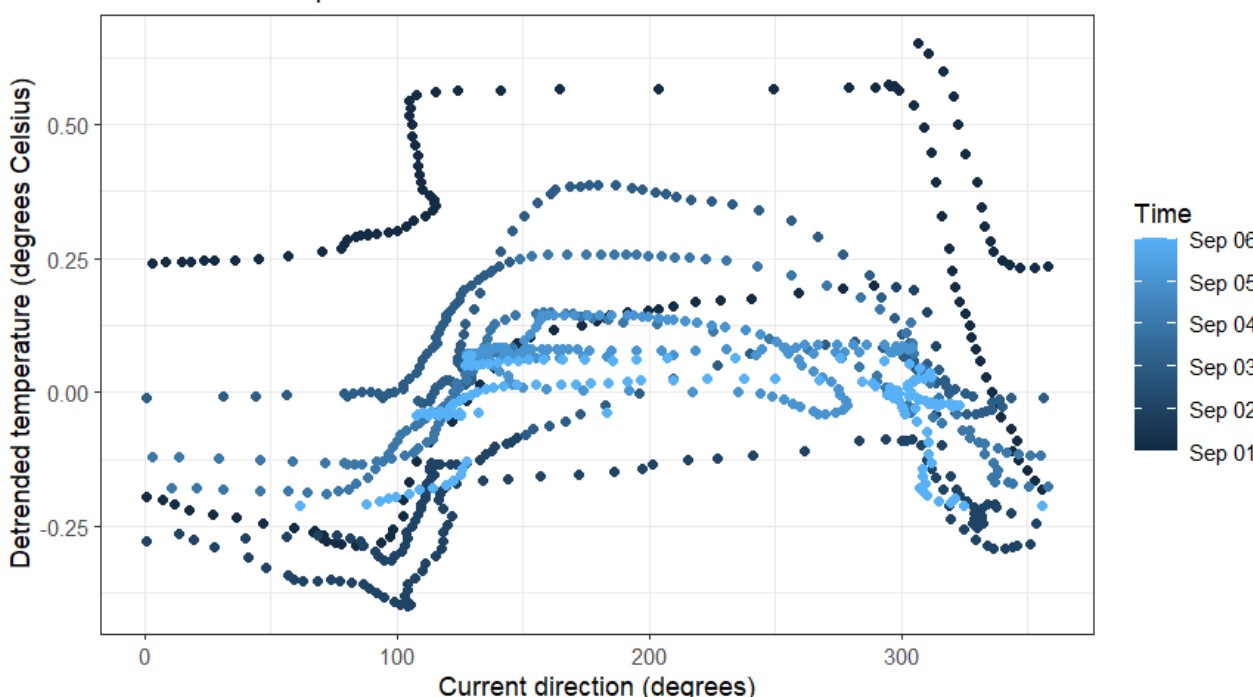


*Figure S9: bottom current direction and (detrended temperature at the HSB lander with data from 1 - 7 September 2018.*

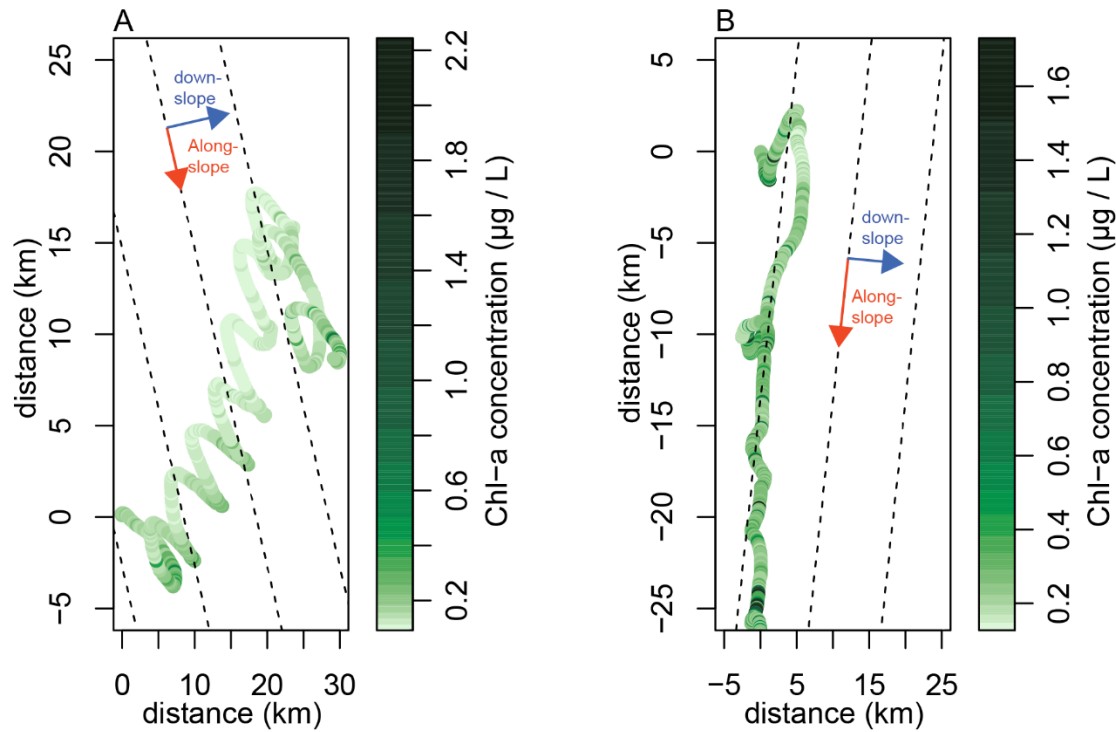


*Figure S10: progressive vector plots with chlorophyll-a as colour variable from 19 to 24 April 2019. With A) the high-*
*sponge-biomass (HSB) lander and b) the low-sponge-biomass (LSB) lander. Dotted lines represent the along slope direction*
*at the respective sites. Note colour is in log-scale.*

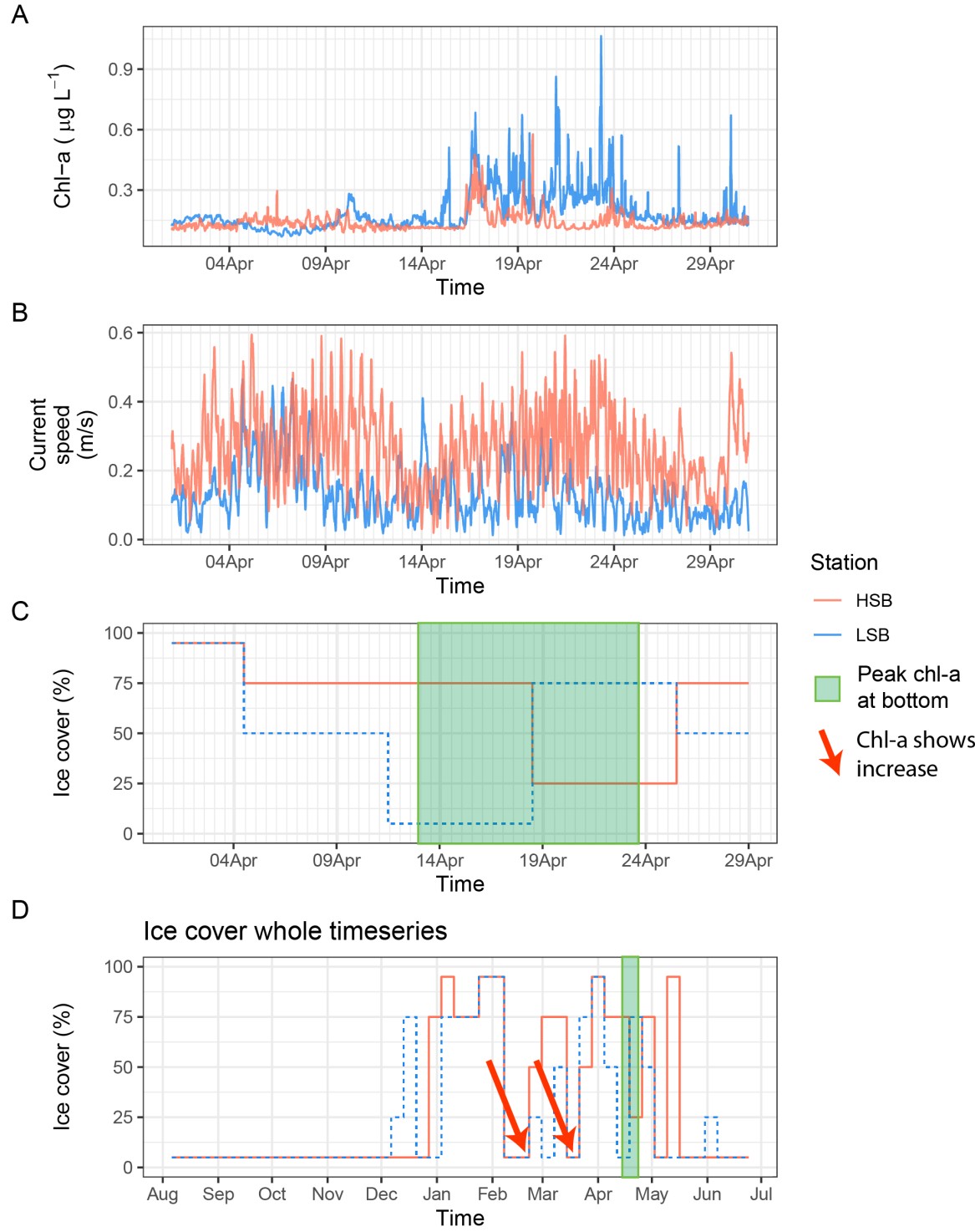


*Figure S11: Spring Chlorophyll-a (A), bottom current speed (B), ice cover (C),during the spring bloom period (1 April-1*
*May, 2019), and ice cover for the whole deployment length (D). Green squares indicate peak bottom chl-a concentrations*
*measured (Figure 9 in the paper), red arrows indicate moment after which chl-a increases at both landers (Figure 9 in*
*paper).*

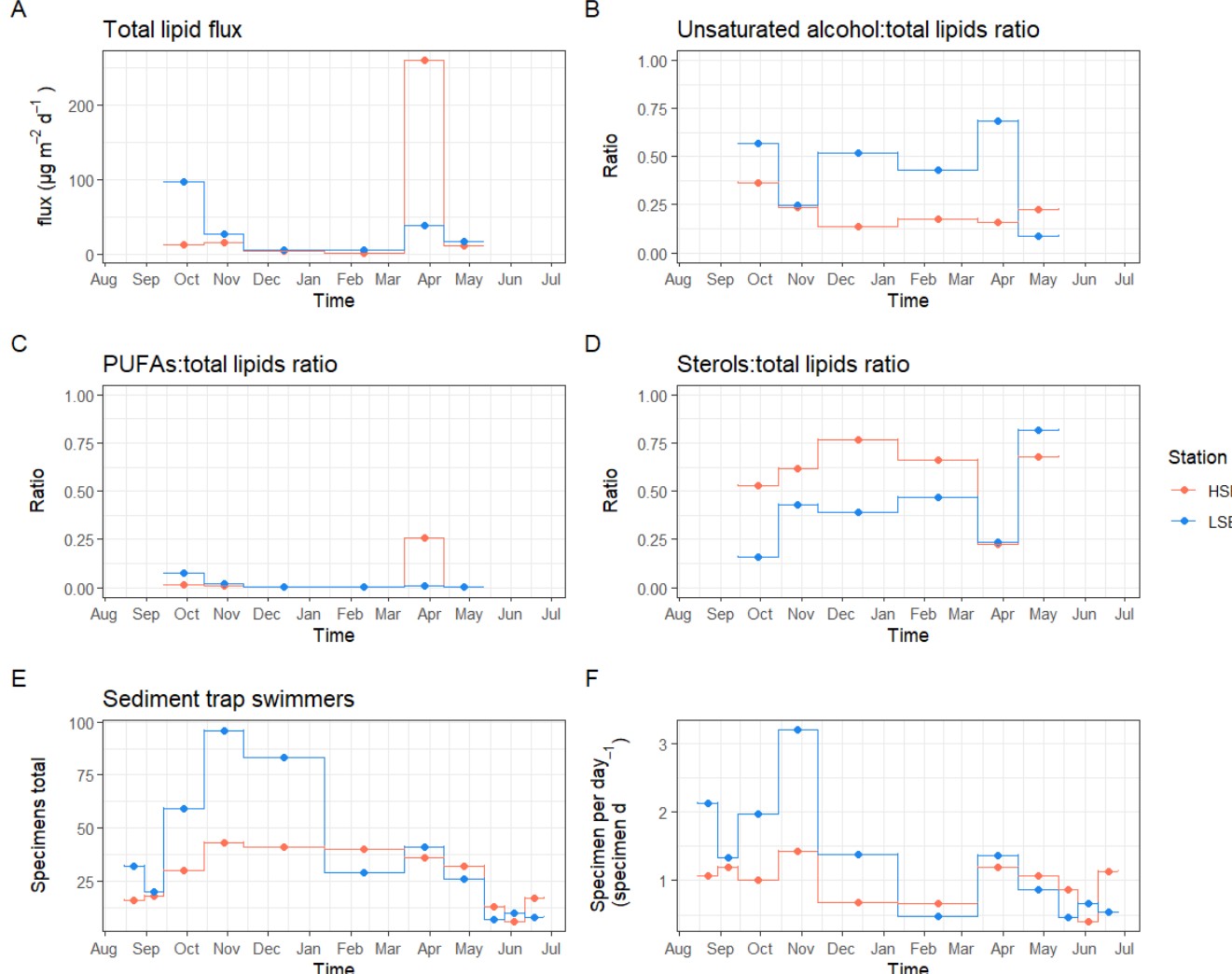


*Figure S12: Sediment trap lipid fluxes. A) Total lipid flux, B) unsaturated alcohol:total lipids ratio, C) poly-unsaturated fatty*
*acid:total lipids ratio, D) sterol:total lipids ratio. E) Swimmers inside sediment trap bottles F) Swimmmers per bottle divided*
*by days that bottle was open.*