# Peer review of "Characterizing regional oceanography and bottom environmental conditions at two"

_EGUsphere, 2024_

## Author Response (AR1)

Friday 6 September 2024

Author's response to the review of the manuscript: "Year-long benthic measurements of environmental conditions indicate high sponge biomass is related to strong bottom currents over the Northern Labrador shelf."

Dear associate editor Andrew Thurber,

Thank you for the opportunity to resubmit our manuscript after major revisions, and my apologies for the delayed response. We would like to thank the two reviewers for their constructive feedback and suggestions. We provided a response to all the reviewer's comments. In short, we provided a new tidal analysis, we revised the discussion in depth, and we rewrote the title, abstract, and conclusion section.

Below we respond (in black) to all the reviewer's specific comments (in grey).

Thank you for considering our resubmitted manuscript for publication in Biogeosciences.

On behalf of the co-authors,
With kind regards,

Evert de Froe

Reply to reviewer 1:

This study presents a long-term time series of environmental conditions within a deep-sea sponge ground. It shows that the food delivery is tightly coupled to the hydrodynamic regime, which defines the occurrence of sponge grounds. I enjoyed reading the manuscript since the study was nicely set-up and many factors that are important for sponge growth were considered. This study will help to entangle the question why sponge grounds establish in certain areas in the deep sea and presents new insight into the hydrodynamics of the seafloor. Year-long measurements of environmental conditions on the seafloor in the deep sea are sparse but can additionally be used to improve hydrodynamical models and even help to assess future changes caused by a changing climate.

There are some major points to consider. This manuscript and many of the figures are in parts already published as a deliverable from the ATLAS project (Wolff et al. 2020). Especially Figure 12, 13, S5, S6, S11 are completely or partly copied from the deliverable. These should be removed from this manuscript and should just be referenced if presented in the same context. I would suggest removing the isotope and respiration part of this manuscript, since there is no new data presented in this manuscript compared to the deliverable from Wolff et al.. Especially the part of carbon utilization assessment by the trawl and ROV transect (L708 ff) is unnecessary because there is no explanation of the methods explained or data shown, which removes a lot of credibility for the rest of the manuscript. The time-series of environmental conditions is anyways the focus of this study, which was (to my knowledge) not presented before. I would have liked if the author would put a bit more effort on the hydrodynamic of this region, including potential occurrence of internal waves, which are important for deep-sea communities in other areas. There is a relatively extensive description of the slope angle which should make a calculation of internal waves (or other hydrodynamic phenomena) possible. Some important data/figures were missing in my opinion: A table of the tidal constituents, the complete ADCP data set, the actual CTD transects or ice cover data (except in the supplements).

The abbreviation of LSB and HSB is used wrongly throughout the complete manuscript, which is hindering the flow of reading. LSB/HSB = "Low/High sponge biomass" (or is it "Low/High sponge biomass site"?) but was almost in every sentence used as "the low/high sponge biomass site". Please use a better abbreviation and check every occurrence if it makes sense.

In general, there are many mistakes that should not appear in a submitted manuscript. For example, figure 1A is missing the complete description (probably wrong figure added?) and some references to figures are wrong.

Dear reviewer,

Thank you for have taken the time to review our manuscript and your elaborate feedback. From reading your feedback, we identify four major concerns: 1. Presenting presumed published data. 2. Speculation on benthic respiration. 3. Not elaborating on internal waves and tidal constituents. 4. Missing ADCP data. We will address these points one by one, after which we reply to your specific comments.

1. Presenting presumed published data. The reviewer states that we use data that was already published as a deliverable from the EU-Horizon project that this study was (partly) funded by. Although we get the point of the reviewer, we do not think we present published data, as the project deliverables are not scientifically peer reviewed. In addition, the deliverable that the review refers to is inaccessible for the

general public and scientist that do not have retrieved access by moderators. Therefore we strongly suggest to keep this data (sediment traps, isotopes) in this paper.

2. Speculation on benthic respiration. We agree with the reviewer that this paragraph contained too much speculation, therefore we removed this paragraph from the text.

3. Not elaborating on internal waves and tidal constituents. We performed an additional analysis on the critical slope at the study sites, where we compared the critical slope at the 10 CTD profile locations.

4. Missing ADCP data. We mistakenly did not report that the ADCP devices used in this study were single point measurement devices. See our reply at L190 for how we addressed this.

Reply to specific comments:

L2-3 The title should be reconsidered. The strong bottom currents are not the only important factor for the sponge ground in this manuscript but also the nutrient and food availability. That all is excluded with this title. All other than the benthic measurements are also excluded with this title (CTD and ARGO float are not benthic).

**Thank you, we altered the title to:**

**Characterizing regional oceanography and bottom environmental conditions at two contrasting sponge grounds on the Northern Labrador shelf**

L38 Please remove "In Canadian waters". The Labrador Shelf is already a unique area, and "Canadian waters" makes it sound political. Is your study region still in Canadas Exclusive Economic Zone?

Thank you, removed.

L40 The sensor most likely measures fluorescence and not chlorophyll-a. The lander is equipped with one sediment trap, not multiple.

Thank you, will make sediment traps singular. We would suggest to keep chlorophyll-a in, as that is the variable that we are reporting in the results.

L41 Check the spaces, looks like there is too much space between some words (whole paragraph).

Thank you, I've checked the text for double spaces and it seems that it looks like this due to justification of the text (formatting issue).

L42 I am confused about the ARGO floats. Were they deployed for this study? Where is the explanation? If not, why is there such a big focus on these floats? Presenting the data so extensively is not necessary and gives no additional information since the general flow pattern are known already.

Thank you for this comment. We see that we did not explain the ARGO float data well enough to prevent confusion in the MS. Argo float data were retrieved from existing datasets, and not deployed for this study. Data from Argo floats were used to compare the seasonal dynamics in temperature at the two benthic landers with the surrounding areas. We think that by showing that the seasonal trend in bottom temperature measured by the lander compares well with Argo float data, we verify that our

measurements are valid and reliable. The Argo float data also serves as material to explain differences in temperature between the two benthic landers.

L44 It seems like the study was not designed to identify potential food sources, since no food source except SPM was considered. Bacteria, dissolved OM and zooplankton are missing for example.

Thank you for this comment, we see that our choice of words is confusing here. We propose to rephrase this to:

Benthic fauna stable isotopes were analysed to investigate food web structure at the sponge grounds.

Extra note (Friday 6 september): after adapting this, we changed the abstract significantly, so response might be different, this goes for all comments on the abstract.

L47ff It was never really shown that tidal currents cause these things. Where are the correlations?

Figure 9 and Figure 11 (old MS numbering) showed that temperature fluctuates with changing current directions. (Linear) Correlations with current direction remains difficult due as current direction is expressed in degrees and therefor a non-linear variable.

We removed the progressive vector plots from the manuscript.

Instead, we added current direction to figure 11, which now shows that from 1-9 September temperature increases when currents are directed 120 degrees at HSB. In addition, we added a plot in the supplements that shows current direction vs. detrended temperature. When current direction approached 120 degrees, the temperature increased.

L51 How could this lead to downwelling?

Downwelling can occur in areas where multiple currents converge and an obstacle, like the Labrador continental shelf obstructs these current from flowing any further. This could result in water being pushed down in the area of convergence.

We leave this statement out of the abstract, as it is not reported as a main finding.

L53 It's not clear for the reader of the abstract why silicate is important.

To clarify, we will add " , which could benefit growth in deep-sea sponges." behind that sentence.

L58…compared to the low-sponge-biomass site.

Adjusted accordingly.

L65-77 The line of thoughts is a bit confusing.

We see what you mean, we will switch the final sentences of the paragraph around, so that we finish with the statement of the VMEs. The last two sentences will now look like this:

Sponge grounds form complex habitats that provide breeding grounds and shelter for (commercially important) fish species, increasing demersal fish biomass and diversity (Kenchington et al., 2013; Kutti et al., 2015; Meyer et al., 2019; Brodnicke et al., 2023). Finally, they are often classified as Vulnerable

Marine Ecosystems (VMEs) as defined by the Food and Agriculture Organization of the United Nations (FAO, 2009).

L75 Not limited to commercially important fish (also see Brodnicke et al. 2023)

Thank you, we will put commercially important fish between brackets, and add the reference to the citations at the end of the sentence.

L79 Maybe also mention deep-sea mining (Wurz et al. 2021).

We will add deep-sea mining in the first sentence of the paragraph, and add the following sentence in the middle of the paragraph (line 89-90):

"In addition, prolonged exposure to elevated concentrations of suspended sediments, i.e. due to deep-sea mining, could adversely affect deep-sea sponges (Wurz et al., 2021)."

L84 Maybe also mention reduced variation due to trawling (Morrison et al. 2020).

Thank you, we have added the reference to " (Colaço et al., 2022)," after the first part of the sentences. As Morrison also describes reduced density and diversity at deep-sea sponge grounds due to benthic trawling.

L102 Food availability was explained in the previous sentence. Gamete dispersal is also connected to currents (maybe connect the sentences).

Thank you for this comment. We adapted these sentences to the following:

*These data indicate that sponge grounds are commonly found in areas with internal waves (Davison et al., 2019), in comparatively strong tidal currents which flush the seafloor with oxygen and nutrient-rich water, and in areas with a high suspended particle matter load near the seabed (Roberts et al., 2018; Hanz et al., 2021a, 2021b).*

The following sentence is removed:

*The spatial distribution of sponge grounds is also linked to gamete/larval dispersal and food availability (Abelson and Denny, 1997; Robertson et al., 2017).*

But the references are, respectively, added to L 94, and L 106 (old numbering).

L123 remove "any"

Done.

L126 I would be a bit careful with extending this too much. The Canadians have very good environmental datasets of their areas.

Thank you for the suggestion. Indeed DFO and Canadian universities and research institutes have good environmental datasets, but year-long recordings of environmental conditions at the seafloor are relatively scarce. Therefore we think this statement remains relevant. Changed the word can for "could" in the last part of the sentence.

L136ff Please mark these areas on the map (with their name) otherwise it is impossible to follow.

Done.

Figure 1 There are no general circulation patterns marked in the map. Make the map bigger and add currents and names.

Done.

L154 remove ":"

Removed.

L157 Rock boulders are no animal. It doesn't make sense to list them in the same sentence with the fauna. The sediment plays a critical role for the occurrence for sponges and should be explained further. Please check the description of the sediment. It is not coherent within the manuscript.

Thank you for pointing this out. We see our writing was unclear on the substrate composition at both benthic lander sites. Because we qualitatively assessed the substrate composition in two ways: video images and the rock dredge, the observations got mixed up in the writing.

The videos show more pebbles and coarse sediment at LSB, but with the rock dredge we collected 5 totes of 64L of soft sediment at LSB. We will elaborate on this, and distinguish the methods, in this paragraph. In addition we will first characterize substrate at each site, and afterwards the biology. We adapted this section to the following:

The substrate at the HSB lander location consisted mostly of pebbles, cobbles, and boulders (Figure 2 A & B; Kenchington et al., 2010; Dinn et al., 2020) and a qualitative assessment of the sediment type at the LSB lander location suggested the dominance of gravel (Coté et al., 2019; Vad et al, submitted.).

The seafloor at HSB was characterized by large-sized massive demosponges (e.g. *Geodia* spp.), glass sponges (e.g. *Asconema* spp.), and large gorgonian corals (*Primnoa resedaeformis*; Figure 2 A & B; Kenchington et al., 2010; Dinn et al., 2020; Vad et al., *submitted*). The benthic community at LSB consisted mostly of small specimens of corals as *Anthomastus* sp., and sponges as *Polymastia sp, Axinella sp, and Mycale sp.* (Figure 2 C & D; Vad et al., *submitted*).

The HSB lander was located on the shelf on a 2° slope and slope aspect was directed northwest at 60°. The LSB lander was located on the upper slope, east of the shelf break, on a 7° slope and aspect was directed southeast at 105° (Figure S1). The west-to-east slope angle was directed downhill (Figure S2 D & H), and north-to-south slope angle was directed uphill at both lander sites (Figure S2 B & F).

L158 It is mentioned later (L613f) that at the LSB site consists of soft, muddy sediment. Figure 2 C&D also shows the opposite.

See comment at L157.

L163 You do not see the slope on the pictures.

Thank you for your comment. In this line we refer to Figure S2 in the supplements. To further clarify, we will adapt this sentence by writing the following:

The west-to-east slope angle was directed downhill (Figure S2 D & H), and north-to-south slope angle was directed uphill at both lander sites (Figure S2 B & F).

Figure 2 Here you define HSB again as "high-sponge-biomass" without the "the" or "site". Do not abbreviate words you don't use further (DFO or CSSF). Pictures are difficult to compare because one is vertically downward with a drop camera and the other one more horizontally. Picture C and D could be the same area when you would take a vertical downward picture for example in the right corner of picture A. Maybe try to find other pictures from the same camera system. What distance are the lasers?

Thank you for your suggestions. The abbreviations refer to the locations where the two benthic landers were deployed. One lander was deployed at a high-sponge-biomass site (HSB), and one lander was deployed at a low-sponge-biomass site (LSB). See also our comments on use of abbreviations above. The lasers are 6 cm apart. We will adapt the caption in the following way:

Figure 1: Images of benthic lander deployment sites, at the high-sponge-biomass lander site (HSB; A,B) and low-sponge-biomass lander site (LSB; C, D). ROV image credits: ArcticNet/Canadian Scientific Submersible Facility (CSSF)/Department of Fisheries and Oceans (DFO). Laser points in panel C & D are 6 cm apart.

L184 "MHz" (make a capital M). Maybe state which particle size classes are observed with a 2 MHz ADCP. These settings will only detect really small particles.

MHz is adjusted accordingly. We added the following sentence on particle size class (in a paragraph below):

*The 2 MHz ADCP have a lower particle size detection limit for particles 12 μm in diameter, and a maximum sensitivity for particles of 242 μm diameter (Haalboom et al., 2021, 2023).*

L185 FLNTU abbreviation is not explained. Sensor measures fluorescence not directly Chlorophyll-a.

Thank you for your comment. We couldn't find a definition of the FLNTU on the website of the manufacturer nor found a definition of FLNTU in the literature. We therefore believe it is probably a brand name of the product (likely meanings something like Fluorescence and Nephelometric Turbidity Units). However, as we are not sure about the precise abbreviation we just mention the name of the product, and adjusted the sentence into the following:

*The landers were each equipped with a 2 MHz ADCP (upward-looking, Nortek Aquadopp), a sediment trap, and a combined optical backscatter sensor for turbidity and fluorescence (Wetlabs ECO-FLNTU; Table S1).*

L186 I do not understand. The ADCP does not measure a 3D velocity field.

Thank you for this comment, we see that our description has lead to some confusion. In this study we used two single point measurement ADCPs (see reply in L190). We replaced this sentence by the following:

*The ADCPs collected every 600 seconds data on pressure, water velocity, echo intensity (acoustic backscatter signal), and water temperature. Furthermore, the built-in accelerometer and magnetometer in the ADCPs collected data on heading, pitch, and roll.*

L188 Are you sure about altitude?

This was a type, and we meant Attitude. Replaced altitude by accelerometer and magnetometer.

L190 Where is the complete data? There is no plot of the ADCP data. Are you sure the first bin is okay? Normally you have to discard (at least) the first two bins (~2-3m distance) since the lander is an obstacle in the flow field and it is highly influencing the (close) currents around it which influences the measurements.

Thank you, we caused confusion by not mentioning that the ADCPs used in this study were single point measurement devices. The ADCPs were set to measure at single bin at 1.14 m distance, this single bin had a cell size of 0.75m. As our current velocity (u,v,w) measurements were consistent throughout the deployment, with, as figure 8 shows, a clear tidal component, we conclude that the benthic lander did not interfere with the velocity measurements.

We added "single point measurement ADCP" in line 199.

L192 Later you state that data was transformed using the program MATLAB. Please clarify.

Thank you, we added "in MATLAB" in this sentence.

L192 Is the FLNTU not one sensor?

Indeed, but we deployed a FLNTU in each benthic lander. We adjusted this by naming it:

combined optical backscatter sensor for turbidity and fluorescence

L193 The sediment trap is one device.

Thank you, adapted.

L195 Remove second dot.

Thank you, done.

L195 What's the end of the collection? Did the last bottle last for the complete cycle? Was the last bottle closed before the lander was retrieved? If it is pulled through the water column while being open, data is not useable.

Thank you for pointing this out. Lander collection happened 1-2 july 2019, and the bottle scheme showed that last sediment trap bottle remained open until 15 july 2019. Therefore, the last bottle closed after retrieval of the lander, therefore the data is not usable and the last datapoint is discarded from the analysis. This has no impact on conclusions of the manuscript.

In addition, date of retrieval mentioned at L182-183 is adapted accordingly to 1-2 july 2019.

L204 Where are the plots of the CTD transects? Was there no sensor for turbidity? It would be interesting if turbidity around the high sponge biomass area is increased.

The CTD transects are plotted in figure 1B. We try to clarify this by adapting the sentences to the following:

*Conductivity-Temperature-Depth (CTD) casts were performed over two cross-shelf transects crossing the LSB and HSB lander sites (Coté et al., 2018; Figure 1B; Table S1). Two CTD casts were carried out on the continental shelf and three on the continental slope, where the third or middle cast was performed above each benthic lander deployment.*

During the Amundsen 2019 research cruise leg 1b the CTD frame was not equipped with a sensor for turbidity.

L214 You probably measured SPM and not sPOM. Where is the data?

We indeed sampled particulate matter, which was later analysed for POC and PN (POM). Therefore we refer to sPOM. We adapt the text by referring to SPM here and at L240.

SPM concentration is added to figure S6.

L217 Cote et al. 2019 is not accessible (internet page with pdf is not existing anymore). Please give another reference (or a way to access it -DOI?)

Thank you for this comment, we checked the URL (https://zenodo.org/records/3862120) that is stated in our reference to Coté et al., 2019, and it seems to work properly.

L219 Explain why two different methods.

The CCGS Amundsen cruise in 2019 leg 1B was the first time that a rock dredge was used on the Amundsen. Therefore the crew and scientists were testing out different modes of deployment of the Rock dredge. At the LSB lander station, the rock dredge collected a lot of material: 5 totes (volume 64L) of very soft sediment were sieved. At both sites (HSB and LSB) substantial amounts of animals were collected. The difference in fauna between the two sites is not caused by the different modes in using the rock dredge.

To accommodate this comment, we added the following sentence:

*During CCGS Amundsen cruise 2019 leg1B, it was the first time that a rock dredge was operated on this research vessel, and therefore different op modes of deployment were tested. At the LSB lander station, the rock dredge collected lots of soft sediment, and therefore "drift" mode was used.*

L240 POM or SPM?

See comment L214.

L244 Please be more specific about the isotopic measurements. How reliable were the measurements? This data was already published before. Consider removing this part.

See our reply in major comments section on presumed published data.

This paragraph explains the laboratory procedures for water column data. We had the POM analysed for $\delta^{13}C$ and $\delta^{15}N$, but we found the results not clear and a procedural error occurred with the $\delta^{15}N$ measurements. Therefore we decided to take out the POM $\delta^{13}C$ and $\delta^{15}N$ results. Mistakenly, I have forgotten to take out the method description. I have deleted the POM isotopic part now, so the paragraph looks as follows:

*Water column nutrient concentrations were analysed with a SEAL QuAATro analyser (Bran + Luebbe, Norderstedt, Germany) following standard colorimetric procedures. SPM samples were freeze-dried, weighed, and analysed for organic carbon content and total nitrogen content.*

The isotopic measurements for benthic fauna and sediment trap data are not excluded, I added a sentence on reliability in the respective paragraph (see reply on comment L276).

L245 Were these particles and swimmer also analyzed?

Thank you for your comment, we counted the number of swimmers from the sediment traps, and identified swimmers into broad taxonomic groups. In L473 – 476 and L676-680 we mention and discuss the results of this analysis. Number of swimmers is added to figure S11 (old numbering, lipid figure).

As discussed, last bottle of both deployments (number 12) is deleted as it was opened when the lander ascended from the bottom.

Data on the lipid flux can be found in figure 12F, and additional data on unsaturated alcohol/PUFAs/Sterols are represented in Figure S11.

L271 Were specific parts of the fauna sampled or only whole animals?

The fauna was mostly subsampled on board of the CCGS Amundsen, as other labs/researchers used the samples for identification purposes. Therefore only parts of the bodies of the fauna were analysed in isotopic composition. We clarify this in the text by adapting the first sentence of the paragraph to the following:

Sponges and other benthic fauna collected using a rock dredge were subsampled on-board the CCGS Amundsen, as parts of the specimens' bodies were used in separate studies and parts for isotopic analysis in this study. In the laboratory, the collected fauna were freeze-dried and homogenized with a pestle mortar/ball mill.

L276 Was this the same as for the water column samples?

Isotopic data on water column samples are not presented (anymore) in the paper.

For comment on reliability (see L244):

We checked with the analytical lab what the margin of error is of the measurement devices for the isotopic measurements, which was ± 0.15‰. The isotopic measurements of the same species were close to each other (i.e., Geodia), therefore we think measurements were accurate.

We added the following sentence:

Standard deviation of $\delta^{13}C$ and $\delta^{15}N$ measurements was 0.15 ‰.

We mean the Vienna Pee Dee Belemnite (δ13C) and air (δ15N) standards. We added this information by changing this sentence to the following:

$\delta^{13}C$ and $\delta^{15}N$ isotope values are expressed in parts per thousand (‰) relative to the international standard Vienna Pee Dee Belemnite and atmospheric air for carbon and nitrogen, respectively.

See L192 comment for answer.

MATLAB and R were both used because of the collaboration between two authors of which one does not know how to do MATLAB and the other does not know how to do R. We added this information by the following:

The transformation of beam coordinates to ENU coordinates for the ADCP data was carried out in MATLAB (MATLAB, 2010), and other data processing steps used R. The following R packages are used during data analysis: *oce, ggplot2, RColorBrewer, cowplot, knitr, reshape2, RNetCDF, readxl, lubridate, xts, ggalt, tibble, dplyr, clifro, mapdata, metR, patchwork, tibbletime, readr, viridis, biwavelet, signal, astsa, terra, raster* (Wickham, 2007, 2016; Grolemund and Wickham, 2011; Neuwirth, 2014; signal developers, 2014; Michna and Woods, 2019; Pedersen, 2019; R Core Team, 2019; Wickham and Bryan, 2019; Wilke, 2019; Kelley and Richards, 2020; Stoffer, 2020; Vaughan and Dancho, 2020; Xie, 2020; Lovelace et al., 2022).

L286 What is the reason for a temporary shift? Did the lander move back again? That doesn't make a lot of sense in my opinion. Where is the pitch and roll data?

We believe the temporary disturbance of the pitch/roll/heading data was due to moving of the lander, and the lander likely moved back again in place as pitch/roll/heading data was close to identical before and after this disturbance. In addition, raw velocity data retrieved from the ADCP, is already corrected for heading/pitch/roll, and therefore we do not expect this to have an effect on our data.

We added the following sentence, and add timeseries of pitch/roll/heading data in the supplements as figure S3:

Pitch/heading/roll was almost identical before and after this disturbance. Furthermore, the ADCPs correct for pitch/roll/heading of the device when producing the raw beam data. Removing datapoints during disturbance did not change the outcome of any of the analyses, statistical tests, or descriptive statistics and therefore datapoints were retained in the HSB time series.

L289 Explain NTU. Turbidity in NTU is not a concentration.

Adapted sentence as follows:

Chl-*a* concentration (in µg L$^{-1}$) and turbidity (in Nephelometric Turbidity Unit; NTU) were calculated from ping counts as described in the manual of the manufacturer.

L293ff Which time-lag and how was the time leg assessed? Where is the outcome of these tests?

Thank you for your comment. Correlation analysis with time lag was done to see at which variables correlated with each other, and at which time lag this correlation was largest. For example, correlating bottom temperature between the two landers showed that highest correlation was found with a 5 day time lag (L411), but without time lag, correlation was low, and therefore not reported. Results are stated in section 3.3.2, L411 – 419, we added a figure in the supplements that shows the cross-correlation between temperatures at the benthic landers to make it more clear, and refer to this graph in results section 3.3.2.

L297f First it was stated that unfiltered data was used and then it was stated that data was filtered before. What is right?

Thank you for your comment, time series data were indeed smoothed prior to spectral and coherence analyses. Removed "on unfiltered data".

L300 Before it was stated that all analyses were performed in R. Please show the results of the harmonic analysis.

Analyses were done in R and Matlab, we will remove statement that all analyses were done in R. A table with results of harmonic analysis will be given. In addition, Figure 8C shows also the results of the harmonic analysis.

L303 Ice data not shown. Please add to supplements.

Sea ice data are shown in Figure S10 D, but we see we made a mistake with the cross-reference. Should be correct now.

L312 remove "yet". If significant show statistics.

We removed the words "yet" and "significant".

L308ff It would be much nicer to show the transects.

Thank you for your comment. We think the reviewer means a coloured chart in which transect length is on the X-axis, depth is on the y-axis, and the variables are given with a color code (oxygen, temperature, etc). However, as we only have 5 CTD measurements per transect, and the transects are ~200 km long, a large area would need to be interpolated. Therefore we chose to combine the transects into one figure.

L316 Sentence is more discussion.

Thank you, we removed this sentence. And Added the following:

*A cold intermediate layer was visible at all profiles between 50 – 150 m depth.*

Furthermore, we adapted the subsequent sentence to the following:

*Salinity increased nearly monotonically with depth up to the pycnocline across all stations.*

L324 It makes no sense to give a range for a mixture of nutrients. The thermocline is not in the Figure.

Range of mixture of nutrients will be removed. We added an indication of the thermocline in the figure.

L328 Remove "to a lesser degree" or give a more specific indication.

Thank you, we rephrased the sentence as follows:

This increased nutrient concentration in the bottom waters was also apparent for silicate at the LSB station (Figure 4C), but not for nitrate (Figure 4B).

L334- 338 This seems weird here. It is not your results.

We understand the confusion. These paragraphs describe the results from the surface buoy and ARGO float data. To accommodate this, we created as separate subheading called:

3.2    Regional oceanography and seasonal temperature

The first sentence is adjusted as follows:

Surface buoy drifter data showed that the HSB lander was located in an area where three (surface) currents converge (Figure 5A).

Figure 3 In A: Why pressure and not depth? All other graphs have depth as an y axis. In general, use the same units (Temperature or potential temperature/ salinity (PSU) or practical salinity). Both graphs have the same legend, combine and make uniform. Explain grey lines in figure C.

Thank you, we replaced pressure with Depth (m), we replaced potential temperature and potential salinity with, respectively, temperature and salinity.  will be replaced with depth. Figure will be adapted that the same units are used. Legends are difficult to combine, as the linetype of plots A and B differ slightly from C. We prefer to keep legends therefore in place. Grey lines are explained in caption.

Figure 4 X-axis description is cut off. Make uniform capital or not (e.g. "Depth (m)" and not "depth(m)"). Maybe combine the legend for region. Its rather confusing. In general: Why not show the data as a transect from west to east?

Figure is adapted, issures are resolved. Added thermocline indication, added data on fluorescence, as was wish of RV2. The legend for region is not combined because figure 4A is a line (from oxygen sensor), and figures 4B and C are point measurements (from Niskin bottles). For comment on transect see reply at L308.

Figure 5 A: Is this really important to show? I think the general currents could be shown in Figure 1. This figure is overloaded with data that is not further used and has some explanation issues. From which depths are the currents? The yellow arrows are not visible. And it is not clear what the difference with the blue arrows is (StE not explained). The dots are above the arrows, which makes it hard to see, which currents are at these positions.

Thank you for this comment. Indeed the general currents are roughly known, but we think that providing a description of the currents in the area that is backed up by data shows a more powerful message. We agree that we haven't explained this figure well enough in the text.

We start 3.2 now with: "Surface buoy drifter data showed that the…"

We adapt the caption to figure 5 as follows:

*general surface circulation pattern in the Labrador Sea based on drifter buoy data spanning from 1995-2020. Arrows indicate mean direction, colours and length of arrow present the strength of the mean flow, the yellow arrows present the standard error of the flow over 1995– 2020. The lander locations are indicated by the coloured dots. B & C) seasonal temperature, from Argo float profiles, of the water layer in which HSB/LSB lander was located. Dots represent individual water-layer-binned temperature measurements vs. date of the year. The lines are a smoothed fit that show the seasonal pattern.*

In addition we provided the following in the methods section:

*Time-average surface currents were derived from trajectories of satellite-tracked surface drifting buoys (drifters) deployed within the NOAA Global Drifter Program during 1995–2020 (Centurioni et al., 2019).*

*The trajectories were obtained from delayed-mode hourly data and real-time variable time-step data (Elipot et al., 2016, 2022). The drifter data were temporally interpolated into 15-min time intervals, binned into hourly bins, and low-pass filtered to remove tidal and inertial oscillations. Then, the surface velocities were binned into a 1/3° grid. The drifter-derived surface currents reveal well-defined large-scale cyclonic circulation of the Labrador Sea, recirculation gyres, and mesoscale circulation features.*

We adapted the figure by making the location of the landers more clear, dots smaller, moved text in the margins.

B: I don't really understand why this data is important to show. On the ARGO float locations from Figure S3 it seems that the ARGO float "HSB" is covering both stations and "LSB" is south of both stations. Make the y axis the same, so it is better comparable.

See major comment section on why we used ARGO float data in our study. We think that showing that the seasonal cycle in temperature over the year matches between the ARGO float data and our lander data, shows that our lander data is valid.

L361 Northward velocity was directed southward sounds wrong; I would describe it in a different way.

Thank you for this comment, we see it might be confusing, but Northward velocity (v) is an international renowned parameter unit ([Parameter Database (ecmwf.int)](Parameter Database (ecmwf.int))). When northward velocity has a negative value, then the direction of the current is southward.

L363 is upward possible? Where is the water coming from?

The vertical velocity was slightly upward, due to the slope angle at which the lander was positioned. The north-to-south slope angle was slightly upwards at both lander locations. Since Northward Velocity (v) was mostly directed southward, vertical velocity was directed slightly upwards.

L365 What is meant with bottom currents? Horizontal currents or the separate velocities? If not, horizontal currents please calculate them.

Yes horizontal currents. We emphasized this by adding word horizontal between bottom and currents.

L367 Remove "signal" and talk about pressure (or water depth).

Replaced pressure signal by pressure.

L371 Give table with tidal analysis.

Thank you for your comment. We now split section 3.3 into three sections:

1. Near-bottom current velocities
2. Near-bottom environmental conditions
3. Tidal analysis of bottom currents and environmental conditions

L383 – 390 are moved to section 3.3.3.

L406 – 427 are also moeved to section 3.3.3.

In section 3.3.3 we show how tides influence bottom hydrodynamics (paragraph 1) and environmental conditions (paragraph 2) at both sites. We added a table with harmonics. Text on cross-correlation between landers and variables is now moved to section 3.1 and 3.2.

L395 Why not recalculate the ABS to decibel sound pressure level to make it comparable to other data? Please show the complete data set of all bins (at least in supplements).

We here only aim to make a qualitative comparison of ABS between the landers. Recalculating ABS to decibel sound pressure level is not straightforward with the set-up for the ADCPs that we used. Therefore, we suggest to keep ABS in echo intensity (as is also used in many other benthic lander studies).

See reply in L190 for reply on the bins.

L400 Which turbidity? From ADCP or FLNTU?

Thank you, here we refer to turbidity measured by the optical backscatter sensor or FLNTU. We added the following information:

"…measured by OBS.."

L405 Reference to Figure S7 is wrong.

Thank you for pointing this out, should be correct now.

Figure F8: cut off the period when the lander was lifted, then you would very likely remove the peak in fluorescence and turbidity.

We think that this comment refers to figure 10. The data in this figure was cut-off at 00:00 $1^{st}$ of July 2019, while the LSB lander was retrieved at 08:35 $1^{st}$ of July 2019, we therefore presume peak in fluorescence and turbidity was not caused by lifting of the lander.

L412 Please give the complete correlation, not only $r^2$ and show the correlation plot.

Thank you for this comment, we removed cross-correlations from our analysis.

L413 ABS was explained before à ABS is also a measure for turbidity. It's confusing if ABS and turbidity are used but describing data from different devices. In general, I would not talk about "signal" but rather what the signal stands for. Every sensor gives just a "signal". In L420 another "signal" (in the data) is mentioned. Very confusing.

Thank you for this comment. Modified the sentence as follows:

Turbidity measured by the ABS increased often at the turning of the tide and at high south-easterly current velocities at HSB (Figure 11F; Figure 9C & G). Strong along slope bottom currents, which are slightly directed downslope, increased ABS turbidity and OBS turbidity at LSB (Figure 11 F; Figure 9C & G)..

Furthermore, we've adapted:

- Replaced signal by peak at L467.
- Replaced ABS with "Turbidity by ABS" throughout the text.
- Replaced signal with signature at start of 3.5
- Replaced "Temperature showed a reoccurring tidal signal" with "Temperature showed a tidal periodicity"
- Removed signal after the word temperature throughout the text.

L417 Show and give complete correlations.

Thank you, we removed cross-correlations analysis from our paper.

L421 Figure S9 does not show ice cover. If S10 is meant I still don't see how the ice cover is influencing Chl-a. Why is the data for this paragraph only showed in the supplements?

Our apologies, we indeed mean S10. This data was put in the supplements, as it should only be seen as a rough estimate of ice cover in the area directly above the landers. In the text we mean that during the spring peak of chl-a at both landers (start April), ice cover was higher at HSB, which could reduce the chl-a quantity in the surface waters and therefore reduce chl-a signal also in bottom waters. However, we see that this is more a point for discussion. We will clarify this in the text, and move part of it to the discussion. We adapted L421-427 to the following, and placed it at the end of section 3.1

*Surface water above the benthic lander locations was partly ice-covered from December to June, but both sites were located at the sea-ice border in the study area and ice cover was highly variable (Figure S13). Only during January ice-coverage was above 70% at both sites. The Hudson Strait froze up in early December and opened again in early June. During the spring bloom, between the end of March and early May, sea-ice coverage tended to be higher at HSB than at LSB (Figure S13D).*

Table 1: What statistics are shown? It seems like these are just mean values and their SD. HSB and LSB are again defined differently than before. Bottom current speed= horizontal current speed? ABS is also a measure for turbidity. How is along and across slope velocity calculated?

We indeed only report mean values and SD. We will review definition of HSB/LSB here. Bottom current speed = indeed horizontal current speed, we will adjust text accordingly. We see we didn't explain cross and along slope velocity. We adapted the text as follows:

- Replaced caption of table 1 by: Benthic lander mean and standard deviations over the year-long deployment period. Values are given as mean ± standard deviation. HSB = high-sponge-biomass lander, LSB = low-sponge-biomass lander. ABS = acoustic backscatter signal.
- Added word "horizontal" to the table.
- Replaced ABS (counts) with Turbidity by ABS (counts)
- Replaced turbidity (NTU) with Turbidity by OBS (NTU)
- At the end of the method section we added the following text: Slope angle and aspect was estimated for each lander by taking the wider topography into account (Gille et al., 2004).

Along-slope and across-slope bottom velocities are derived from the bottom current direction, slope aspect, and bottom horizontal current speed.

Figure 7 I do not think that this is the best way of showing the data. First of all, progressive vector plots might be useful to show transport for short time periods, but after more than some days these plots are invalid. At every new position a parcel of water experiences different forcing, that are not comparable to the initial position. Especially factors like temperature are very different at each position because of the general water mass distributions. I do not believe that you can extract any useful information from this plot. Additionally, it was shown before that tidal currents are influencing temperature which are not even included in this figure. Figure 7C&D are on the other hand a nice way of showing the current direction distribution, whereas it is not clear why not the same way of visualization like for example current speed was used (maybe not useful).

We see your point. We decided to remove this plot altogether from the manuscript, also to meet suggestion from other reviewer to reduce number of panels. Information on residual current direction and magnitude will now be incorporated in a the figure on tidal ellipses.

Figure 8 C is difficult to see the difference between temperature, ABS and Chl-a. I would suggest different colors and to move the legend to the right side.

Subpanel A, and D are removed from the manuscript.

Figure is now build as follows:

- subpanel A: tidal ellipses + comparison with barotropic model
- Subpanel B: spectral frequencies of bottom current speed
- subpanel B: spectral frequencies of environmental conditions: temperature, turbidity measured by ABS, chl-a.

Figure 9 This is in principle a more acceptable that Figure 7. Turbidity (counts) for example is depending on the current speed which is not at all visible in this graph. Turbidity is not differing on small scale spatial patterns as it seems in this plot. The same for temperature. Temperature depends on the current direction of the water mass with a certain temperature. This graph suggests that temperature varies on a small spatial scale. Data needs to be shown differently.

Thank you, we decided to remove the progressive vector plots from the manuscript. Current direction is added to figure 11. In figure 8 a sub figure is added that shows increase of temperature and turbidity is related to current direction at HSB, and therefore to the tide.

Figure 10 Nice way of showing the data. ABS should also be called turbidity or any other coherent way (acoustic vs. optical backscatter). Chl-a needs to be with a capital C. The two different turbidity signals should be used to describe the particle sizes of the SPM (peaks in spring= bigger particles because not seen in the ADCP data). The data of the last days before the retrieval of the lander can probably be cut off, since it is likely showing the time period when the lander was lifted, or other bottom touching activities were nearby.

We adjusted y-axis labels The time period in when the lander was lifted was already cut-off from the data.

The 2MHZ ADCP has a maximum sensitivity for particles of with 242 µm diameter (see comment L184 and section 2.2.1 in the new text). As these are not small particles, and OBS tends to work less well with larger particles, we do not think that peaks in spring automatically mean bigger particles. A more in-depth analysis would be needed for this, but we think that falls outside the scope of the current study.

Figure 11 This makes Figure 9 obsolete. Much nicer way of showing the data. Same comments for the y axis description like before. Again, comparing the turbidity sensors here would help with the particle description à Fine sediments are dependent on tidal currents and are peaking with a high horizontal velocity when currents are in a certain direction (I assume when currents are coming from the shelf, current direction is missing here). A correlation matrix would help here as well.

We added current direction here to this figure, and remove figure 9. This reduces the number of panels/figures in the MS, which was also a wish of the other reviewer.

L457-495 Remove everything that is redundant from the already published data (Wolff et al. 2020). Or mark it better as a repetition if needed to be included.

See our reply in major comments section.

L486 "indicated a lower trophic level.." is for the discussion.

Thank you, adjusted the text.

Figure 12 Most of this figure was published already.

See our reply in major comments section on presumed published data.

Figure 13 This is not a bi-plot (includes sample data and variable data). The plot was already published before and is therefore redundant. Other food sources are missing (zooplankton, dissolved food).

See our reply in major comments section on presumed published data.

In this study we did not take samples on zooplankton or dissolved food, we rephrase our words in a way that we do not refer to the data as food sources, but more to isotopic signatures or food web structure.

We adapt the title of 4.5 to: Isotopic signatures of benthic macrofauna at two contrasting sponge grounds.

Replaced food sources with; "isotopic signatures" in the introduction, as well as in the first paragraph of the discussion.

L499 remove "more specifically."

Done.

L521 "Our ARGO float profiles". The ARGO data is not from this study, this is misleading.

We will adapt this in the text by referring to Argo float profiles/data, and remove the word "our".

L529 How was this calculated?

We decided to remove this part.

L532 Why is the water temperature explaining salinity?

In our study we only measured temperature on the benthic landers, and we do not have info on the salinity. In our study we therefore used temperature to investigate connectivity between the two sites, and in Sutcliff et al, and Myers et al, they looked at salinity to look at connectivity between Hudson strait outflow and Labrador Shelf.

We adapted the final sentence to the following:

This supports earlier findings that found a connection between the Hudson Strait outflow strength and the southern Labrador Shelf water based on salinity measurements (Sutcliffe et al., 1983; Myers et al., 1990).

L540 What is in-situ remineralization supposed to be? Remineralization in the Baffin Bay? Why is there more organic matter in the deep-water that is demineralized? Is it not primary production or terrestrial OM that is demineralized? Normally deep water has higher nutrients but not higher OM.

Thank you, we see that there are some words missing in this sentence. It is indeed OM which is remineralized, we will adjust this sentence.

L567 The tidal amplitude? Or tidally driven horizontal currents?

Thank you, we indeed mean tidally driven horizontal currents. Adjusted.

L568 This is highly dependent on the sponge species. Are these the same sponges?

White et al, 2003 investigated glass sponges, and here we looked at sponge ground what was a mix of glass sponges and demosponges. So partly the same sponge species. We added the following text to the discussion:

Although caution should be applied comparing these areas, as the sponge fields at the Porcupine Sea Bight consists mostly of glass sponges, and here we see a mixture of glass sponges and massive demosponges.

L578 Why is the ARGO float data important here?

We use Argo float data to confirm our benthic lander data. See comment L42.

L584 I would expect current direction to have an effect but not tidal currents.

We indeed mean current direction. The sentence is adjusted as follows:

Bottom current direction has a distinct effect on bottom temperature at both sites, and this effect depends on the season.

L587ff Currents at the shelf break are likely very different than on the shelf and this is likely not a valid argument even though the time frame would maybe allow assumptions like this.

Thank you for your comment. We see it might be a stretch to assume transport of 5 km. Reviewer 2 also found this part to speculative, so we decided to remove L 643 to L660 from the discussion.

From "The temperature fluctuations…. …. mixed with bottom water"

L606 Logically not completely right. Higher resuspension is clogging the sponge due to more particles in the water column. The particles are also retained within the sponge and are not removed from currents.

An important point is also that resuspension is delivering food since bacteria and organic substances are binding to particles, which will be eaten. Higher turbidity is associated to higher amounts of bacteria.

Thank you, we agree higher turbidity could enhance food availability, but could also cause clogging of the sponges at high turbidity values. We think that the second part of this sentence has caused for confusion. We adapted this sentence as follows:

Second, resuspension caused by oscillating tidal bottom currents enhance organic matter availability in the benthic boundary layer and enhance food supply to the sponges (Roberts et al., 2018).

L614 contrary to what was mentioned before and also pictures show something else.

See comment L157.

L616 Check your data. Is there a time lag between high currents and high turbidity? Then SPM is likely coming from somewhere else. Otherwise, OM might be collected between the sponge (and other fauna), like mentioned in the introduction.

Thank you, looking at Figure 11 (in the old MS), peak turbidity measured by ABS, occurs right after, or nearly simultaneously with peak bottom current speed. We therefor conclude that our statement here is correct.

L618 I don't agree that it prevents smothering.

We here mean smothering from a layer of sediment, or sedimentation. If currents would be low, sedimentation increases and sponges could, at one point, become covered by a layer of sediment. We added "from sedimentation" at the end of this sentence.

L619 It is known that OM/bacteria are binding to particles and can act as a food source. Third? What is first and second?

Thank you, we adapted this sentence to the following:

*Resuspension has also been linked to high sponge biomass (Davison et al., 2019), since potential food sources as organic matter and bacteria can bind to suspended particles in the water column.*

L635 repetition.

Thank you, we deleted this sentence to avoid repetition.

L638 Show prim. production.

We did not measure primary production in our study, but we have fluorescence profiles of the ctd casts in figure S4, which is now added to figure 4.

We checked the Frajka-Williams and Rhines 2010 paper (following comments from reviewer 2) and their paper does not report primary production, but mean surface chl-a concentrations over the period 1998 to 2008. We therefore altered this sentence to the following:

*The fact that surface chl-a concentrations are comparable above the two lander station sites (see Figure 2A in Frajka-Williams and Rhines, 2010), suggests that differences in primary production alone are insufficient to explain the differences sponge biomass between regions.*

L650 Which export? I assume from the Hudson Strait. Why is the export tidal?

We mean export from surface waters to the seafloor. We mean that chl-a showed a tidal periodicity during this period, and we see that back in Figure S9B. We now added the correct cross-refence.

L677 Why would that have an influence? They produce offspring when they have enough energy but that can happen to any time of the year.

We are not sure what the reviewer means with this comment.

L689 Was not Chlorophyll-a measured here and not fluorescence?

Thank you, we mixed up sediment trap data with the fluorescence sensors. We removed "and low fluorescence".

L706 It seems that a more important factor is the horizontal distance to the food source from the Hudson Bay or Baffin Bay.

Thank you for this comment. We think that, as surface chl-a concentration is likely comparable between the two sites (Frajka-Williams et al, 2010), food sources are comparable between the two sites and the water column dynamics play a more important role. In addition, surface waters of the Northern Labrador Sea or Baffin Bay are quite distant from the study area and therefore unlikely serves as a source of organic matter. We therefore do not assume that there is a food source at Hudson Bay or Baffin Bay, but local hydrodynamics are more important.

L708-730 This paragraph is not okay in this way. There is no explanation about the estimate of the sponge biomass by the ROV or trawl method, which is a number that is not easy to obtain. Sponge biomass of a trawl cannot be referenced with personal communication. The image analysis was already published in an earlier report and also discussed there. The paragraph is based on a comparison of these results and the data from personal communication and there are no results concerning this paragraph in this manuscript. Therefore, this paragraph should be removed. The respiration potential can be discussed in one sentence when needed but it should be clear, how estimates are made or from where numbers were taken. I would recommend concentrating on the long-term measurements of environmental factors, which is the strength of this manuscript.

Thank you, we tried to give an estimate of OM utilization at the two sites, but agree it is quite speculative. We will remove this paragraph.

L731- 767 The isotope data was already discussed in Wolff et al. There are some more points in the discussion of this manuscript, but they are not connected to any new results. The author has to consider

if the isotope part should stay in the manuscript. Again, I think there is no substantial gain in presenting the same results again and they are not relevant for the time-series of the environmental data.

See our reply in major comments section.

L776 Please rephrase the sentence. Why should primary production alone be a good predictor, this was never mentioned before? Maybe better state your positive result: Primary production in an area that is connected by water currents is important for the delivery of food to the sponge area. Please reflect these results also in your title.

Following comments from the other reviewer as well, we rewrote our conclusion section.

Figure S1 and S2 Is this really necessary? Nothing was really done with this data.

Figure S1 and S2 serve as background information on the slope angles, which gives insight in the vertical velocities for instance. We removed figure S2.

Figure S3 Seems like the "HSB- ARGO" was above both areas. And the "LSB-ARGO" was mostly south of the study area.

Thank you, we chose the cut-off point of the HSB/LSB regions at the latitude of the LSB lander, as the general current direction is southward and therefore HSB-Argo locations cover the area where the three currents converge, and the LSB – Argo locations cover the area south of it, and where less sponges are present.

Figure S4F The buoyancy frequency shows a very interesting peak in ~230 m above the high sponge area. This is a bit higher than the sponge ground itself but might be very important for the food delivery. This should be mentioned.

Thank you for this observation. The peak the reviewer refers to is from profiles of the CTD casts taken on the shelf, likely related to the boundary layer. CTD transect above the HSB lander shows also a peak in BVF, but more close to the Cold-Intermediate-Layer. We added a sentence on this in the results.

Figure S5 and S6 remove à is already published data.

See our reply in major comments section.

Figure S7 y-axes are different, which makes it hard to compare. This figure is not referenced in the text. Refer to it or remove it.

Thank you for your comment, we want to compare the temperature and salinity between landers in each respective water layer (350 – 450 m depth at HSB; 550 – 650 m depth at LSB). If we would adapt the y-axes, the LSB water layer (550 – 650 m depth) would shrink considerably in size and difference between HSB and LSB would not be visible anymore. We therefore suggest to keep the scale of the y-axes different between the subpanels.

We now refer to it in section 3.2.

Reply to reviewer 2

Dear Dr. de Froe and colleagues,

Please see the below my revision of your manuscript entitled "Year-long benthic measurements of environmental conditions indicate high sponge biomass is related to strong bottom currents over the Northern Labrador shelf." The manuscript presents an important dataset and would make an interesting contribution to science, at least from a regional perspective. The science and the writing are relatively good. I do have some comments that I would like to see addressed before considering this work publishable. I tagged it here as ''major revision", but it mostly concerns the Discussion and the Conclusion.

My main concerns are about the Discussion where a lot of speculation is made. In addition, I have the feeling that the authors make a wrong use of some references and/or use references and attribute this as a finding (e.g. "We showed that ... [citation]"). I would ask you to please carefully review your Discussion. Overall, I finished my reading thinking... so what? I provide more specific comments below.

There is also a lot of Figures/panels. I would advise that if data are not presented or necessary for the findings described, they could be removed.

Dear reviewer, thank you for taking the time to review our manuscript and thank you for your feedback and comments.

Thank you for your comments and suggestions on our manuscript. As the reviewer suggest we will carefully review our discussion section, use of figures, and write a new conclusion section. Below we reply to the comments more extensively:

- L. 139: "This region is known for intense mixing and water mass transformation (Dunbar, 1951; Kollmeyer et al., 1967; Griffiths et al., 1981; Drinkwater and Jones, 1987; Yashayaev, 2007)"

-> The Yashayaev paper is not about mixing in Hudson Strait and should be removed.

Thank you for this suggestion, removed.

- L. 140: "four distinct flow components can be identified (Figure 1A...)"

-> There is no flow component in Figure 1...

Thank you for pointing this out. By mistake we uploaded an old version of the figure to the MS. We updated figure 1.

- Figure 1: There is a big void at 61N. Is it because there is no sponge or because it was not sampled? (e.g. it is a closure?). If the latter, it should be clearly stated that the sampling is not representative. Overall, it looks like there is sponge almost everywhere that was sample (very little number of red dots).

- Figure 1 (related to point above): It looks like the LSB site was just not sampled. How can you tell it has low biomass?

The sponge biomass data was retrieved from Kenchington et al., 2010. It seems that the data presented in Figure 1 does not contain all the sponge biomass data, and that an error occurred. We now updated fig 1.

- L. 186: What is the vertical resolution of ADCP bins?

Thank you for this comment, we see that our description has lead to some confusion. In this study we used two single point measurement ADCPs. We added this information in the paragraphs in section 2.2.1.

See also our reply to comment L190 by reviewer 1.

- L.286: "ADCP sensor at HSB were shifted for a small period of the deployment, implying the lander was occasionally moving a bit"

-> Like sea floor displacement? Or oscillation and coming back? I have trouble to picture what you mean and what could have caused such a displacement.

We believe the temporary disturbance of the pitch/roll/heading data was due to moving of the lander, and the lander likely moved back again in place as pitch/roll/heading data was close to identical before and after this disturbance. In addition, raw velocity data retrieved from the ADCP, is already corrected for heading/pitch/roll, and therefore we do not expect this to have an effect on our data.

We added the following sentence, and add timeseries of pitch/roll/heading data in the supplements as figure S3:

Pitch/heading/roll was almost identical before and after this disturbance. Furthermore, the ADCPs correct for pitch/roll/heading of the device when producing the raw beam data. Removing datapoints during disturbance did not change the outcome of any of the analyses, statistical tests, or descriptive statistics and therefore datapoints were retained in the HSB time series.

- L. 304: "extracted from weekly ice charts (Canadian Government, 2022)."

-> Please verify is this is the correct way to cite this document.

Thank you, we adapted this to (Canadian Ice Service, 2022).

- L. 316: "The temperature changes from cooling to warming with depth signify the Cold Intermediate Layer (CIL)."

-> This is a weird sentence. Can you expand?

Thank you, we mean that a cold intermediate layer is visible. We removed this sentence and replaced it by the following:

A cold intermediate layer was visible at all profiles between 50 – 150 m depth.

- L. 322: "The bottom oxygen concentrations at the lander stations were, for both transects, relatively depleted compared to the deep water CTD transects at similar depths."

-> 250 uM/L is still pretty high!

Thank you, we adapted this sentence to the following:

*Although oxygen concentrations were overall quite high, the bottom oxygen concentrations at the lander stations were, for both transects, relatively depleted compared to the deep water CTD transects at similar depths.*

- Figure 3 caption:

-> is it the "surface circulation"?
We think the reviewer means figure 5. Yes it is surface circulation we will add this.

-> What are the dots and the lines in panel B?

Dots are data from Argo float profiles (averaged over the water layer), the lines are smoothed fit, that shows the seasonal pattern. We adapted the caption to the following:

general surface circulation pattern in the Labrador Sea based on Argo float data spanning from 1995 - 2020. Arrows indicate mean direction, colours and length of arrow present the strength of the mean flow, the yellow arrows present the standard error of the flow over 1995 – 2020. The lander locations are indicated by the coloured dots. B & C) seasonal temperature, from Argo float profiles, of the water layer in which HSB/LSB lander was located. Dots represent individual water-layer-binned temperature measurements vs. date of the year. The lines are a smoothed fit that shows the seasonal pattern.

- Figure 12: Would it be possible to have error bars?

We can add error bars (SD) the dotted lines (the mean), not to individual measurements. Therefore we suggest to leave the figure as is.

DISCUSSION:

This is that should be revised in depth.

- L. 519: "Our findings confirm previous work which showed that Irminger Water is gradually cooled while moving southward by mixing with the Baffin Island Current (Cuny et al., 2002)"

-> How your study confirms this?

Thank you for your comment. Cuny et al, 2002 show that Irminger water is gradually cooled moving southward by misign with the Baffin Island Current. Our study does not confirm this, but we do see the same pattern. Namely warmer waters at HSB (containing IW), which cools moving southward in the Labrador current. We replace confirm by "are in line with".

*L. 521: "our Argo float profiles..."

-> Argo profiles are public. I would rephrase, these are not "yours".

Thank you, we adapted this.

- L. 523: "For example, the 350-450 m depth layer in the HSB area regularly showed presence of Irminger Water (>4.5 ºC), while Irminger Water was only sporadically measured at LSB (Figure 5B). Irminger Water might therefore be cooled and freshened in the area around HSB due to convergence and consequently mixing with the Hudson Outflow and Baffin Island Current."

-> I don't understand the reasoning. How do your discard different advection patterns? Or the fact that the moorings are at different depths?

Thank you for your comment, the advection patterns are quite well studied in this areas, and we also show advection patterns with the drifter data (Fig 5A). We take the different depths into account by showing temperature (derived from argo floats profiles) for both water layers (fig 5 B and C). LSB temperature measurements are consistently cooler than at HSB, and this implies that also Irminger water (>4.5 °) gradually cools while the current is flowing southward.

L. 528: " This time lag corresponds to an along slope velocity of 0.3 m s-1, which is close to the mean bottom current speeds measured at HSB (0.25 m s -1) and on the Labrador Slope (0.11 – 0.23 m s-1 ; Lazier and Wright, 1993). This supports earlier findings on the Labrador Shelf that found a connection between the Hudson Strait outflow strength and the southern Labrador Shelf water salinity (Sutcliffe et al., 1983; Myers et al., 1990)"

-> Again, I don't see how the former support the latter.

Thank you, we try to show that water masses between the two landers are related. The fact that our corresponding time-lag (five days) matches the residual mean current speed, indicates that the two sites are related. We know the Labrador current flows southward along the continental slope, and our data matches this. This sentence also caused confusion with the other reviewer, therefore we deleted it.

L. 536: "These observations are thought to be related to the sources of the bottom water and circulation. Thus, intermediate water flows from Baffin Bay via the Davis Strait southward along the continental slope (Curry et al., 2014). This water mass, referred to as Baffin Bay Water (BBW), contains higher nutrient concentrations (e.g., 41.6 ± 25.5 μM Si(OH)4, 18.5 ±2.6μM NO3-; Sherwood et al., 2021) due to in situ remineralization of deep water circulating in the Baffin Bay basin (Jones et al., 1984; Tremblay et al., 2002; Lehmann et al., 2019). BBW mixes with water masses on the Labrador Shelf and Slope and Hudson Strait outflow water while flowing southward along the Labrador Slope, resulting in lower nutrient concentrations at the LSB compared to the HSB (Figure S4). The absence of high nutrient concentrations at the shelf/deep CTD station at both sites supports this interpretation."

-> I am not sure if your study shows this...

We try to show that the observations done from regional oceanography (general currents) match the idea that increased bottom nutrients are coming from Baffin Bay water. Sherwood et al. 2021 shows an extensive study that investigates how bottom water from Baffin Bay flows across Davis strait and on to the Northern Labrador shelf. The nutrient concentrations measured in this study match this. See reply at L 557 on how we adapted the text.

L.557: "Namely, such an efflux from the sediments would be quickly advected away by the high bottom tidal currents, while nutrient concentrations were elevated up to 100 meters above the bottom (Figure 4 B & C)"

-> Can you rule out advection from upstream? sediment efflux at the site would be advected, but efflux from upstream can be advected at the site of the measurements... I don't think that your explanation is satisfactory."

We agree that advection upstream is likely the source of these nutrients, but we added nuance to this paragraph. We rewrote this section. We made it a separate section: 4.2 increased bottom nutrient concentrations and added text on how sediment efflux could lead to increased nutrient concentrations. We still think that large-scale circulation is more plausible, but more research would be needed to test this.

L. 563: "This study provides the first concurrent long-term measurements of hydrodynamic and environmental conditions at a high-and low

sponge-biomass site."

-> You mean specific to these sites or worldwide?

Thank you, we adapted the sentence as follows:

This study, to our knowledge, is the first to report year-long hydrodynamic- and environmental conditions measured simultaneously at a high- and low-sponge-biomass ground.

L. 578: "Temperature increased gradually from summer until December, which is measured previously on the Labrador upper slope and attributed to Irminger Water (Cuny et al., 2002)."

-> The temperature increase is attributed to the IW? Can you clarify? A larger proportion of warmer IW throughout the year?

Indeed, this is shown by Cuny et al. 2002, but as we added a paragraph on internal waves, and we mention Irminger Water in paragraph 4.1, we decided to leave this paragraph out.

L. 589: "As the lander was placed ~500 m from the shelf break (Figure S2C&D), and bottom water could be transported ~5 km in the north-easterly direction in one semidiurnal tidal cycle (Figure 9A), this means that colder bottom water is transported on to the Labrador Shelf

from beyond the shelf break to the HSB lander site."

-> I am not sure I follow this reasoning... The temperature is colder above (on the shelf) compared to the slope. You say here that colder temperature is transported on the warm shelf...

Thank you for this comment, we will remove this statement because it seems to speculative. See comment L587 of reviewer 1.

L. 593: "Therefore, although higher variability in bottom water temperature has been attributed to the presence.

of internal waves at other sponge grounds (Roberts et al., 2018; Davison et al., 2019), we attribute the variability in our study area to tidalinduced cross-slope transport of bottom water."

-> The main generation mechanisms for internal waves are the tides. So in this case do you mean the barotropic tide? If so, you would need to demonstrate that there is not internal waves generated (or propagating) at these sites.

Thank you for this comment, we added an analysis on comparing the tidal ellipses from the benthic landers with the barotropic model data. From this, we could elucidate that at HSB, tidal energy is dissipated through tide-topography interaction. HSB was also located at near-critical conditions for the M2 tide, which confirms this idea that there is tide-topography interaction. At LSB, barotropic and baroclinic tidal currents were of similar magnitude, only orientation was different, which could be attributed to the slope orientation.

L. 612-628: This paragraph is convincing (strong current = better feeding opportunities). All the rest that precedes seems highly speculative.

Thank you, we hope with the new and deleted texts in section 4.1, 4.2, and 4.3 we tackled this comment.

L. 637: "Our CTD profiles show elevated chl-a concentrations in the CIL (~150 m depth)"

-> It would be nice to have the chl-a profile in figure 3A.

We doubled checked our data, and fluorescence measurements of the CTD profiles show increased fluorescence just above the Cold-Intermediate-Layer, and slightly elevated values at HSB inside the CIL, compared to LSB at similar depth. We remove this statement as it seemed wrongly interpret by us.

We added fluorescence in figure 4. Fluorescence profiles are also given in S4.

L. 638: The fact that primary production rates are comparable above the two lander station site s (Frajka-Williams and Rhines, 2010), suggests that differences in primary production alone are insufficient to explain the differences sponge biomass between regions."

-> This statement is problematic. Frajka-Williams paper is not about these 2 lander sites but rather compared the Northern vs Central Labrador Sea.

Thank you for this comment. Indeed the Frajka-Williams paper focusses on the Northern vs Central Labrador Sea. However, in figure 2a of that paper, it shows increased mean chlorophyll concentrations of phytoplankton bloom in surface waters at both the HSB and LSB landers. Therefore, we argue that primary production alone is not sufficient to explain differences in sponge biomass between the two lander sites. See also reply on comment L 645 – 669. We adapted the text as follows:

Previous studies show that surface chl-a concentrations are comparable between the two sponge grounds (see Figure 2A in Frajka-Williams and Rhines, 2010), suggesting that differences in surface productivity alone are insufficient to explain the differences sponge biomass between regions.

L.645-669: This whole paragraph is highly speculative. The authors talk about seasonal sea ice and spring bloom dynamics as if there was no interannual changes (while they are actually quite large!). The whole paragraph needs to be better backed up by the literature review, or by new data.

In this paragraph we tried to describe and explain arrival of chl-a in bottom waters in early March, which is surprisingly early for this region. Indeed there is yearly variability in sea ice and spring bloom dynamics, but the start of spring bloom in this region and the Labrador shelf has been studied. We think we can still make a statement that chl-a arrived earlier than the timing of the onset of the spring bloom. A recent paper on spring bloom initiation on the Labrador Shelf (Cyr et al., 2023) shows that standard deviation of spring bloom initiation is around 21 days for the southern Labrador shelf. Figure 2B in Frajka-williams & Rhines 2010 shows bloom onset is around June. This gives us an indication that spring bloom starts later than chl-a arrives at the seafloor, which has likely to do with stratification.

We added two studies to the analysis (Cyr et al. 2021, and Cyr et al. ,2023), which provide sufficient information that our assumptions are reasonable.

We rewrote the whole paragraph and hope that we explained our ideas more clearly and backed our assumptions up by literature.

-> So what? how this study contributes to science? Can you highlight the main findings? The results that are currently recalled in the Conclusion looks to me things that were already known...

We see that our statements in the conclusion are to generic to pin point what our study has contributed to science. We rewrote our conclusion section.

- L. 184: mHz -> MHz

Thank you, adjusted.

- Figure 4: xlabel is cut.

Thank you, adjusted.

- L. 505: "is known" -> "are known"

Thank you, adjusted.